# How important are future marine and shipping aerosol emissions in a warming Arctic summer and autumn?

Anina Gilgen[1], Wan Ting Katty Huang[1], Luisa Ickes[1,2], David Neubauer[1], and Ulrike Lohmann[1]

[1]ETH Zürich, Institute for Atmospheric and Climate Science, Switzerland
[2]Now at Stockholm University, Department of Meteorology, Sweden

*Correspondence to:* Anina Gilgen (anina.gilgen@env.ethz.ch)

**Abstract.** Future sea ice retreat in the Arctic in summer and autumn is expected to affect both natural and anthropogenic aerosol emissions: sea ice acts as a barrier between the ocean and the atmosphere, and reducing it increases dimethyl sulphide and sea salt emissions. Additionally, a decrease in the area and thickness of sea ice could lead to enhanced Arctic ship traffic, for example due to shorter routes of cargo ships. Changes in the emissions of aerosol particles can then influence cloud properties, precipitation, surface albedo, and radiation. Next to changes in aerosol emissions, clouds will also be affected by increases in Arctic temperatures and humidities. In this study, we quantify how future aerosol radiative forcings and cloud radiative effects might change in the Arctic in late summer (July/August) and early autumn (September/October).

Simulations were conducted for the years 2004 and 2050 with the global aerosol-climate model ECHAM6-HAM2. For 2050, simulations with and without additional ship emissions in the Arctic were carried out to quantify the impact of these emissions on the Arctic climate.

In the future, sea salt as well as dimethyl sulphide emissions and burdens will increase in the Arctic. The increase in cloud condensation nuclei, which is due to changes in aerosol particles and meteorology, will enhance cloud droplet number concentrations over the Arctic Ocean ($+10\%$ in late summer and $+29\%$ in early autumn; in-cloud values averaged between $75°$ and $90°$ N). Furthermore, both liquid and total water path will increase ($+10\%$ and $+8\%$ in late summer; $+34\%$ and $+26\%$ in early autumn) since the specific humidity will be enhanced due to higher temperatures and the exposure of the ocean's surface.

Changes in both aerosol radiative forcings and cloud radiative effects at the top of the atmosphere will not be dominated by the aerosol particles and clouds themselves but by the decrease in surface albedo (and by the increase in surface temperature for the longwave cloud radiative effect in early autumn). Mainly due to the reduction in sea ice, the aerosol radiative forcing will become less positive (decreasing from $0.53\,\mathrm{W\,m^{-2}}$ to $0.36\,\mathrm{W\,m^{-2}}$ in late summer and from $0.15\,\mathrm{W\,m^{-2}}$ to $0.11\,\mathrm{W\,m^{-2}}$ in early autumn). The decrease in sea ice is also mainly responsible for changes in the net cloud radiative effect, which will become more negative in late summer (changing from $-36\,\mathrm{W\,m^{-2}}$ to $-46\,\mathrm{W\,m^{-2}}$). Therefore, the cooling component of both aerosols and clouds will gain importance in the future.

We found that future Arctic ship emissions related to transport and oil/gas extraction (Peters et al., 2011, ACP) will not have a large impact on clouds and radiation: changes in aerosol concentrations only become significant when we increase these ship emissions by a factor of ten. However, even with tenfold ship emissions, the net aerosol radiative forcing shows no significant changes. Enhanced black carbon deposition on snow leads to a locally significant but very small increase in

radiative forcing over the central Arctic Ocean in early autumn (no significant increase for average between $75°$ and $90°$ N). Furthermore, the tenfold higher ship emissions increase the optical thickness and lifetime of clouds in late summer (net cloud radiative effect changing from $-48\,\mathrm{W\,m^{-2}}$ to $-52\,\mathrm{W\,m^{-2}}$). These aerosol-cloud effects have a considerably larger influence on the radiative forcing than the direct effects of particles (both aerosol particles in the atmosphere and particles deposited on snow). In summary, future ship emissions of aerosols and their precursor gases might have a net cooling effect, which is small compared to other changes in future Arctic climate such as those caused by the decrease in surface albedo.

## 1 Introduction

In the last decades, Arctic temperatures have increased approximately twice as fast as the global average temperature, e.g. due to temperature and ice-albedo feedbacks (Pithan and Mauritsen, 2014), changes in the Atlantic Ocean thermohaline circulation (Chylek et al., 2009), and the decline in European anthropogenic $SO_2$ emissions since 1980 (Navarro et al., 2016). This temperature increase has been leading to reductions in both Arctic sea ice extent and thickness for the last decades: for the period from November 1978 (start of satellite records) to December 2012, the Northern Hemisphere sea ice extent decreased by $3.8 \pm 0.3\%$ per decade (Vaughan et al., 2013). This decrease is more pronounced in summer and autumn than in winter and spring (Vaughan et al., 2013). Since global and thus Arctic temperatures will further increase in the near future, the Arctic is expected to become ice-free in late summer within the next several decades (Collins et al., 2013; McFarquhar et al., 2011).

Sea ice concentration (SIC) refers to the percentage of an area which is covered with sea ice. Ocean areas with high SIC have a larger surface albedo and reduced exchanges of heat, momentum, and gases between the ocean and the atmosphere than areas with low SIC (Vaughan et al., 2013). With an open Arctic Ocean, natural aerosol emissions will increase because more sea salt particles and more dimethyl sulphide (DMS; a precursor for sulphate aerosol particles) will be emitted (Browse et al., 2014). Under present-day conditions, emissions from the ocean are already an important aerosol source in some Arctic regions in summer: measuring aerosol particles with radii between $0.25\,\mu m$ to $10\,\mu m$ in Svalbard (a map of the Arctic can be found in the Appendix, see Fig. A1), Deshpande and Kambra (2014) identified sea spray particles as the main source for Arctic summer aerosol particles. In a modelling study, Struthers et al. (2011) found that sea ice retreat might increase the sea salt aerosol number emissions in summer by a factor of two to three by 2100.

Presently, the contribution of Arctic shipping to aerosol radiative forcings within the Arctic is very small compared to other emissions (AMAP Assessment, 2015). However, sea ice retreat might cause an increase in shipping aerosol emissions over the Arctic Ocean, since reduced summer sea ice enables ships to cross the Arctic Ocean. Cargo ships could shorten their paths (Corbett et al., 2010; Melia et al., 2016), tourism could be expanded (Eckhardt et al., 2013), and the Arctic oil and gas production will likely be intensified (Peters et al., 2011). Compared to other regions, the present-day Arctic air is exceptionally pristine, and aerosol levels are very low. Hence, increases in both natural and anthropogenic aerosol emissions might have a strong effect on cloud properties and radiation. Furthermore, deposition of black carbon (BC) on snow and ice lowers the surface albedo (Warren and Wiscombe, 1985) and therefore has the potential to accelerate sea ice retreat (Flanner, 2013).

Aerosol particles influence clouds e.g. by acting as cloud condensation nuclei (CCN) or ice nucleating particles (INPs). Freezing processes involving INPs are called heterogeneous freezing; for a recent overview on heterogeneous freezing modes, see Kanji et al. (2017). The ability of an aerosol particle to act either as a CCN or an INP depends on its size and its chemical composition (Boucher et al., 2013). Hence, both aerosol concentration and composition influence cloud properties substantially

(Boucher et al., 2013): at a constant liquid water content (LWC), an increase in the number concentration of CCN changes the cloud droplet number concentration (CDNC); it leads to more but smaller droplets, which increases the total surface area of the cloud. Since cloud droplets must reach a certain size before they form rain, this process may delay the formation of precipitation (Albrecht, 1989). On the other hand, an increase in aerosol concentrations could also lead to enhanced precipitation due to the presence of INPs, which reduce the required supercooling and/or supersaturation for ice initiation. An earlier freezing of some

cloud droplets, followed by the Wegener-Bergeron-Findeisen process, may rapidly form cold precipitation (Lohmann, 2002). Aerosol-cloud interactions can affect cloud properties and the onset and/or intensity of precipitation further, as described e.g. in Lohmann and Feichter (2005); Jackson et al. (2012). In Arctic mixed-phase clouds, observations suggest that the number of precipitating ice particles decreases by 1-2 orders of magnitude under polluted conditions when aerosol concentrations are high (Lance et al., 2011).

However, clouds are not only affected by aerosol particles. Increasing atmospheric temperature is expected to shift the melting and the freezing levels – and thus also cloud ice – to higher altitudes. Additionally, higher temperatures will increase evaporation from the surface and, consequently, the available water vapour in the atmosphere. An open ocean further amplifies the increase in water vapour. Analysing satellite data from 2000 to 2010, Liu et al. (2012) found a negative correlation between sea ice extent and cloud cover over the Arctic Ocean, which was statistically significant and especially pronounced between

July and November. Recently, Abe et al. (2016) showed with a coupled atmosphere-ocean model that enhanced heat and moisture fluxes resulting from the reduction in sea ice cover are indeed responsible for the simulated increases in cloud cover.

Both aerosol particles and clouds impact the Earth's radiation budget. Whether an aerosol particle predominantely absorbs or scatters radiation depends on its physical and chemical characteristics. Aerosol scattering of shortwave (SW) radiation tends to cool the atmosphere, whereas absorption of SW and longwave (LW) radiation tend to warm it (Boucher et al., 2013). The

sum of scattering and absorption is called extinction. Since the aerosol extinction (normalised by the aerosol mass) is generally largest when the size of the particle is comparable to the size of the wavelength, the SW effect is more important than the LW effect for the majority of atmospheric particles (Stier et al., 2007). However, for large particles such as dust or sea salt, LW effects can become relevant (Stier et al., 2007).

Similar to aerosol particles, clouds impact the Earth's radiation budget by absorption of LW radiation (warming) and scatter-

ing of SW radiation (cooling). To a smaller extent, LW radiation is also scattered and SW radiation absorbed (Chou et al., 1999; Slingo, 1989). The absorption and emission of LW radiation is a function of the emissivity of the cloud (which depends on microphysical cloud properties and the water path), the (height-dependent) cloud temperature, and the surface temperature (Corti and Peter, 2009; Chen et al., 2006; Shupe and Intrieri, 2003). The scattering of SW radiation is a function of the microphysical cloud properties, of the cloud water path, of the solar zenith angle, and of the surface albedo (Corti and Peter, 2009; Liou,

2002; Shupe and Intrieri, 2003). Since aerosol particles influence cloud microphysics, they also impact cloud radiative effects

(CREs). With a higher CCN concentration at constant LWC, more radiation is scattered back to space and the cooling effect of clouds is enhanced. This is the so-called "Twomey effect" (Twomey, 1974, 1977), also referred to as radiative forcing due to aerosol-cloud interactions ($RF_{aci}$; Boucher et al., 2013). Furthermore, changes in cloud lifetime (e.g. delayed precipitation; "Albrecht effect"; Albrecht, 1989) also affect the CREs. Together with $RF_{aci}$, these adjustments are referred to as the effective radiative forcing due to aerosol-cloud interactions ($ERF_{aci}$; Boucher et al., 2013).

Compared with the global mean, the SW radiative effect of Arctic clouds is less important because of the large solar zenith angle and the high surface albedo (Alterskjær et al., 2010). Therefore, the LW absorption of clouds becomes more important and can dominate the total CRE depending on the specific time and location. Arctic clouds warm the planet in the annual average and show a net cooling effect only in summer (Walsh and Chapman, 1998).

How Arctic clouds and their radiative effects will change in the future is still an open question. Generally, both the SW and the LW CRE are expected to become stronger when more CCN are available (Mauritsen et al., 2011). However, compared to other temperature feedbacks, the contribution of changes in Arctic clouds might be small (Pithan and Mauritsen, 2014). Palm et al. (2010) suggested that the overall effect of enhanced aerosol concentrations is to increase the net warming effect of Arctic clouds because LW radiation dominates in the long polar winter. In contrast, a modelling study of Alterskjær et al. (2010) found that the increase in anthropogenic aerosol emissions since pre-industrial times has led to larger changes in the annual Arctic SW (-0.85 $\mathrm{W\,m^{-2}}$) than in the LW (0.55 $\mathrm{W\,m^{-2}}$) CRE at the surface. However, their simulated LW radiation effect was approximately one order of magnitude smaller than suggested by the observation-based study of Garrett and Zhao (2006). Whereas Garrett and Zhao (2006) considered measurements from a specific location (near Barrow, Alaska) and analysed strong pollution events, Alterskjær et al. (2010) simulated the effect over the whole Arctic (defined as north of $71°$ N in their study) under all conditions. Other explanations for the different results include model uncertainties, especially regarding cloud cover and thin cloud frequency (Alterskjær et al., 2010). For the Arctic summer, Mauritsen et al. (2011) showed that an increase in the number of aerosol particles can either decrease or increase the net CRE depending on the background aerosol concentration.

Therefore, the future increase in both natural and anthropogenic aerosol emissions due to sea ice decline is expected to influence radiation both directly and indirectly. The following studies investigated the impact of future changes in either natural or anthropogenic aerosol emissions: Struthers et al. (2011, using the global aerosol-climate model CAM-Oslo) and Browse et al. (2014, using the global aerosol microphysics model GLOMAP) analysed the influence of enhanced natural aerosol emissions on Arctic clouds in the future; we will discuss their findings in the comparison with our results. The impact of Arctic shipping on black carbon deposition on snow and ice by 2050 was studied by Browse et al. (2013), who found only a small contribution of BC from ships. Dalsøren et al. (2013) used the chemical climate model OsloCTM2 to study the impact of enhanced global and Arctic shipping in 2030. In their high growth scenario, $O_3$ had the largest impact on radiative forcing in autumn (August to October).

In this study, we aim to quantify changes in future Arctic aerosol particles from both natural and anthropogenic sources enabled by sea ice reductions. Furthermore, we analyse changes in clouds and radiation, which are partly caused by these changes in aerosol emissions. We use the state-of-the-art global aerosol-climate model ECHAM6-HAM2, which allows us to study changes in Arctic aerosols and their impact on climate.

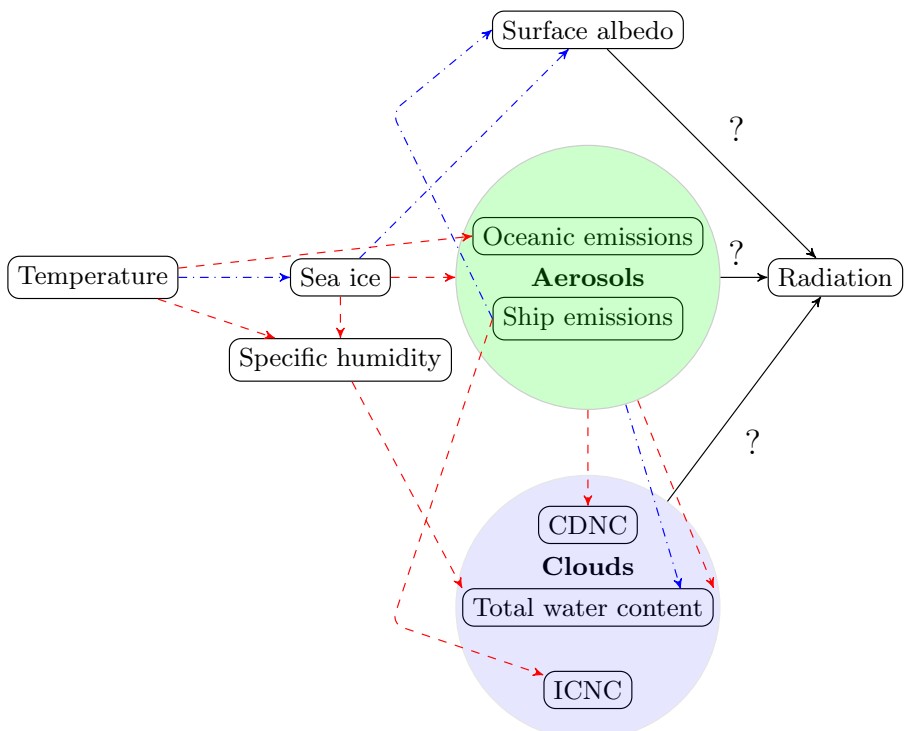

**Figure 1.** Simplified sketch showing how different variables (may) vary as a result of enhanced Arctic temperatures. Red dashed arrows denote expected increases, blue dashdotted arrows expected decreases. Black solid arrows show which components impact radiation. CDNC and ICNC stand for cloud droplet and ice crystal number concentration, respectively. Note that an increase in aerosol concentrations can either increase or decrease precipitation and thus the total water content, as mentioned in Sect. 1.

Figure 1 provides a simplified overview of how the increase in Arctic temperature can affect radiation. The most important interactions between atmospheric variables, aerosols, clouds, and surface properties are included. The figure shows that the increase in temperature directly affects sea ice, specific humidity, and aerosols. Changes in these variables can then directly or indirectly impact clouds and radiation.

5     The model and the simulations, the boundary conditions, the emissions, and the used statistical method are described in Sect. 2. In the results and discussion section (Sect. 3), we focus on the months July to October, when both the decrease in SIC and the increase in shipping through the Arctic Ocean will be most pronounced. In the conclusions (Sect. 4), our key findings are summarised.

## 2 Methodology

### 2.1 ECHAM6-HAM2

#### 2.1.1 General information about ECHAM6-HAM2

ECHAM6-HAM2 is the combination of the general circulation model ECHAM6 (Stevens et al., 2013) with the two-moment

cloud microphysics scheme by Lohmann et al. (2007) and the aerosol model HAM2 (Stier et al., 2005; Zhang et al., 2012). ECHAM6 solves prognostic equations for vorticity, divergence, surface pressure, and surface temperature and uses a flux form semi-Lagrangian transport scheme to advect water vapour, cloud liquid water, cloud ice, and trace components.

In HAM2, the aerosol components $SO_4$ (sulphate), BC, organic carbon (OC), sea salt, and mineral dust are considered (Zhang et al., 2012). The size distribution of the aerosol particles is described by four size ranges: the nucleation mode

($r_m < 5\,nm$; $r_m$ is the mode radius of the aerosol particles), the Aitken mode ($5\,nm < r_m < 50\,nm$), the accumulation mode ($50\,nm < r_m < 500\,nm$), and the coarse mode ($r_m > 500\,nm$). Only a soluble mode exists for the nucleation mode, whereas a soluble/internally mixed and an insoluble mode exist for the other three size modes. Therefore, seven aerosol modes are considered in total, each described by a log-normal size distribution. Coagulation and condensation can shift aerosol particles to larger modes and/or from insoluble to internally mixed modes. Removal processes of aerosol particles in ECHAM6-HAM2

comprise wet deposition, dry deposition, and sedimentation. To link the simulated aerosol population with the CDNC and the ice crystal number concentration (ICNC), parameterisations for cloud droplet activation and ice nucleation are implemented (Abdul-Razzak and Ghan, 2000; Lohmann and Diehl, 2006; Lohmann et al., 2008).

Regarding the sulphur chemistry, DMS is oxidised to $SO_2$ (sulphur dioxide), which can form sulphuric acid in the aqueous phase or in the gas phase. Gas-phase sulphuric acid in the atmosphere can either nucleate, i.e. form new small, soluble particles,

or condense onto pre-existing aerosol particles. Condensation can be limited by the available surface area of aerosol particles, by the available gas-phase sulphuric acid, or by the diffusion of the gas-phase sulphuric acid to the particle surface. If any gas-phase sulphuric acid is left after condensation, the sulphuric acid nucleates and forms new sulphate particles. Besides the available concentration of sulphuric acid, nucleation depends on temperature and relative humidity.

In the standard ECHAM6-HAM2 setup, a minimum CDNC of $40\,cm^{-3}$ is implemented. This ensures that the global CDNC

is not unrealistically low due to missing aerosol species in the model such as nitrate or due to the simplistic model description of organics (no explicit treatment of secondary organic aerosols; neglection of marine organics). Without a lower threshold for CDNC, the model might underestimate the CDNC also in the Arctic, where organic aerosol particles are emitted from the sea surface microlayer (Hawkins and Russell, 2010; Bigg et al., 2004; Leck and Bigg, 2005; Chang et al., 2011). However, since the Arctic is a remote environment with low aerosol concentrations, observations show that the value of $40\,cm^{-3}$ is

often undershot in this region: between July 15[th] and September 23[th], Bigg and Leck (2001) measured daily median CCN concentrations between $15\,cm^{-3}$ and $50\,cm^{-3}$ at a supersaturation of $0.25\%$. In July 2014, Leaitch et al. (2016) found a median CDNC of $10\,cm^{-3}$ for low-altitude clouds (cloud top below $200\,m$) and of $101\,cm^{-3}$ at higher altitudes. In October 2004, McFarquhar et al. (2007) conducted aircraft measurements in single-layer stratus clouds and found averaged cloud

droplet number concentrations of $43.6 \pm 30.5\,\mathrm{cm}^{-3}$. Applying the standard CDNC threshold of $40\,\mathrm{cm}^{-3}$ would drastically reduce the influence of changes in the CCN concentration and therefore impede aerosol-cloud interactions. Thus, we decided to use $10\,\mathrm{cm}^{-3}$ as a lower threshold for the CDNC everywhere and retuned this new model version. The studies by Bigg and Leck (2001) and Leaitch et al. (2016) indicate that values even below this lower threshold can occur. While these measurements are representative for a specific point, our model represents average values over a larger area ($1.875° \times 1.875°$), which should be less variable than a point measurement. Nevertheless, we acknowledge that the threshold of $10\,\mathrm{cm}^{-3}$ could still be too high under certain conditions. In the Arctic, this threshold is hit 11% (weighted with liquid water content) averaged from July to October under present-day conditions. Without a lower threshold for CDNC, the underestimation of the Arctic aerosol concentrations in our model (see also Section 3) would locally lead to unrealistically high precipitation formation rates and a too strong effect of increased aerosol emissions on clouds.

### 2.1.2 Aerosol emissions

Emissions of sea salt, dust, and oceanic DMS are calculated online and depend on the $10\,\mathrm{m}$ horizontal wind speed ($u_{10}$). Marine organic aerosol emissions are not considered in this study. Sea salt emissions follow Long et al. (2011) with sea surface temperature (SST) corrections according to Sofiev et al. (2011). The correction is applied because SST affects sea salt emissions by influencing bubble rising velocities, the gas exchange between the bubbles and the water, the bubble bursting behaviour, and maybe also the coverage of oceanic whitecaps (Lewis and Schwartz, 2004). Dust emissions are calculated as stated in Tegen et al. (2002), with some modifications based on Cheng et al. (2008). The monthly mean DMS seawater concentrations are prescribed according to Kettle and Andreae (2000), and the flux from the ocean to the atmosphere is calculated following Nightingale et al. (2000). Changes in oceanic DMS concentrations are not straightforward to project: taking primary production or SST as a proxy seems not justified since Arctic oceanic DMS concentrations also depend on taxonomic differences in phytoplanktonic assemblages (Becagli et al., 2016). Using a coupled ocean-atmosphere model (with ECHAM5-HAM as atmospheric component), the study by Kloster et al. (2007) explicitly simulates DMS but only reports changes between the time periods 2061-2090 and 1861-1890, which are not directly comparable to the time periods we are interested in. Thus, we decided to leave the oceanic DMS concentrations unchanged.

Besides dust, sea salt, and oceanic DMS, the emissions of all other aerosol components or sulphate precursors are prescribed, mainly from the ACCMIP emission inventory (Lamarque et al., 2010). For ship emissions, we used the inventories by Dalsøren et al. (2009) and Peters et al. (2011), which are described in the next paragraphs. Ship emissions are put into the second lowest model layer ($\approx 150\,\mathrm{m}$). While OC and BC particles from ships are exclusively emitted into the insoluble Aitken mode, the sulphate mass is equally distributed between the accumulation and the coarse modes. It is assumed that $2.5\,\%$ of $SO_2$ from ships is emitted as primary sulphate (Dentener et al., 2006).

Our ship emissions are based on the inventories by Dalsøren et al. (2009) and Peters et al. (2011), which include the species $SO_2$, BC, and OC. The shipping emissions for the year 2004 follow Dalsøren et al. (2009), who combined the observational data sets COADS (Comprehensive Ocean-Atmosphere Data Set) and AMVER[1] considering ships above 100 gross tons. For

---

[1]http://www.amver.com/

the global ship emissions in the year 2050, we use the Dalsøren et al. (2009) ship emission inventory and apply the same reduction in emission factors for 2050 as in the study by Peters et al. (2011) (80% for $SO_2$ and 20% for OC), which are based on the Amendments to MARPOL Annex VI adopted by the International Maritime Organization in 2007.

For additional ship emissions in the Arctic in 2050, we take the ship emissions by Peters et al. (2011). They used the 2004-inventory by Dalsøren et al. (2009) as a "background" for calculating future Arctic ship emissions in the year 2050 for transit shipping and for shipping that is related to oil and gas production. Changes in ship emissions from the sectors tourism, fishery, and local/national transport are not considered. For the year 2004, no transit shipping was assumed, and the oil and gas shipping was estimated based on oil tankers operating in the Arctic region. The expected increase in these two sectors is related to SIC: less sea ice will faciliate the passage through the Arctic ocean and expose new areas to oil and gas production. Peters et al. (2011) assumed that emission factors of $SO_2$ and OC will decrease due to regulations and improved technology but that everything else (other aerosol emission factors; shipping routes) will remain constant.

We increased Arctic ship emissions by a factor of ten to detect a significant signal in aerosol particles. This is in agreement with the results of Peters et al. (2014), who studied the effect of ship emissions on tropical warm clouds with ECHAM5-HAM. In the following, we show how realistic these tenfold emissions are in the context of other studies and recent findings.

Compared with other estimates of future Arctic transit shipping, the results from Peters et al. (2011) lie between those from Paxian et al. (2010) and Corbett et al. (2010): the fuel consumption by Paxian et al. (2010) is 1.4 to 2.4 times smaller than the values reported by Peters et al. (2011). Depending on the scenario, the estimated $CO_2$ emissions by Corbett et al. (2010) are 2 to 4.6 times higher in 2050 than the values reported by Peters et al. (2011).

Recently, McKuin and Campbell (2016) pointed out that both global and Arctic ship emission inventories might underestimate BC ship emissions because too low BC emission factors were used. While the ship emission inventory by Peters et al. (2011) used a BC emission factor of 0.35, McKuin and Campbell (2016) found – depending on the averaging method and the area – factors between 0.79 and 0.92. These differences in BC emission factors suggest that $\approx 2.5$ times higher BC ship emissions might be more appropriate for future transit and oil-gas-related shipping than the original estimate from Peters et al. (2011). However, note that McKuin and Campbell (2016) also point out that small fishing vessels ($< 100$ gross tonnage), which are not included in the analysis by Peters et al. (2011), contribute substantially to ship emissions. Neglecting these emissions from fishing activity likely leads to an underestimation of background ship emissions. This is important because higher background emissions might lead to a smaller impact of future transit and oil/gas related shipping (i.e. smaller relative increase of total aerosol emissions).

Tenfold ship emissions are achieved if we consider that i) transit shipping (which contributes most to the ship emissions by Peters et al. (2011) over the pristine Arctic Ocean between July and October) might be up to 4.6 times higher according to Corbett et al. (2010) and ii) the BC emission factor used by Peters et al. (2011) is likely underestimated by a factor of $\approx 2.5$. Increasing the additional ship emissions (both transit shipping and oil/gas related shipping) from Peters et al. (2011) by a factor of ten is an upper estimate and is probably too high to represent conditions in 2050.

### 2.1.3 Heterogeneous freezing of mixed-phase clouds in ECHAM6-HAM2

In ECHAM6-HAM2, dust and BC particles (also those emitted by ships) can act as INPs in the immersion mode when transferred to the internally mixed mode. Heterogeneous freezing in ECHAM5-HAM is thoroughly described in the study of Hoose et al. (2008). The only differences in ECHAM6-HAM2 are that i) contact freezing is limited to montmorillonite dust because contact freezing of BC is controversial and that ii) only particles in the accumulation and coarse modes can induce freezing. The freezing rate is defined as the number of cloud droplets that freeze per time and volume of air. Among other factors such as temperature, the contact freezing rate depends on the volume-mean droplet radius as well as the CDNC, while the immersion freezing rate depends on the cloud water mixing ratio.

### 2.1.4 Calculation of aerosol radiative forcings and cloud radiative effects

Both aerosol radiative forcings and CREs are calculated online by calling the radiation scheme once with and once without considering aerosol particles or clouds; the difference between the two radiation calls is called radiative forcing due to aerosol-radiation interactions ($RF_{ari}$) for aerosols and CRE for clouds. While $RF_{ari}$ is normally used for the forcing by anthropogenic emissions being the only external forcing to the system, a double radiation call with zero aerosols as the reference provides the sum of the natural and anthropogenic radiative forcing. For SW radiation, aerosol radiative forcings and CREs both depend on the surface albedo. For example, an aerosol particle that scatters SW radiation can either have a cooling or a warming effect depending on whether the underlying surface has a lower or a higher surface albedo, respectively. Since the surface albedo decreases in our future simulations due to melting of sea ice, changes in $RF_{ari}$ and CRE can either be caused by changes in aerosol/cloud properties or changes in surface albedo. For clouds, we can distinguish the two causes by applying the cloud radiative kernel method described in the study of Zelinka et al. (2012), which is independent of changes in surface albedo. With this method, we can furthermore disentangle changes in LW CRE caused by changes in clouds from those caused by surface temperature changes (see also Shell et al., 2008). A higher surface temperature enhances the outgoing LW radiation from the surface. Thus, more LW radiation can be absorbed by clouds and the LW CRE increases. In addition, the cloud radiative kernel method allows for diagnosis of how different cloud types (low and free-tropospheric clouds; Zelinka et al., 2016) and changes in different cloud properties (cloud cover/amount, cloud optical thickness, and cloud top altitude) contribute to the total changes in CREs. Note that with this method, gridboxes without incoming radiation are set to missing values for both SW and LW CRE. While this is not an issue for July, August, and September, most values between $85°$ and $90°$ N are missing in October. For the SW CRE, we set these missing values to zero; for the LW CRE, September values instead of the mean over September and October are shown for these gridboxes.

In our model, the reduction of snow albedo due to deposited BC is determined through interpolations of a lookup table based on a single-layer application of the SNICAR model (Flanner et al., 2007). The BC concentration in the top $2\,cm$ of snow is considered (Engels, 2016). The concentration depends on the surface influx of snowfall as well as the influx of BC removed from the atmosphere through dry deposition, wet deposition, and sedimentation. Both BC scavenged by hydrometeors through in-cloud (Croft et al., 2010) and below-cloud (Croft et al., 2009) wet deposition is assumed to reach the surface within one

timestep (if hydrometeors do not evaporate in subsaturated regions below clouds). Given that both the spatial and the temporal resolutions of our model are low ($1.875° \times 1.875°$; $7.5\,\mathrm{min}$), this assumption seems justified. The concentration of BC in snow can be further modified through scavenging by snow melt and glacier runoff. Since the scavenging ratios are low (0.2 for BC particles in the internally mixed mode and 0.03 for those in the externally mixed mode; Flanner et al., 2007), the BC concentration in snow increases after snow melt. Lastly, while albedo reductions of snow on land and on sea ice are considered, the impact of BC deposition on bare sea ice is not. This is due to the different characteristics of the sea ice albedo concerning its interaction with the deposited BC, which would only lie on top of the ice instead of being mixed-in. However, as the spatial coverage of bare sea ice without any snow cover is small in the model, the impact of omitting this darkening is expected to be negligible.

## 2.2 Model simulations

A summary of the model simulations can be found in Table 1. ECHAM6-HAM2 is an atmosphere-only model, i.e. SIC and SST need to be prescribed (see Section 2.3). To estimate the impact of future Arctic warming and sea ice retreat on aerosol particles and clouds, we conducted simulations under present-day (year 2004) and future (year 2050) conditions. The following simulations were performed with a resolution of T63L31 (corresponding to $\approx 1.875\,° \times 1.875\,°$ with 31 vertical levels):

- **arctic_2004**: Global greenhouse gas concentrations, SIC, SST, and prescribed aerosol emissions (including ships) from the year 2004 are used.

- **arctic_2050_EM2004**: The global greenhouse gas concentrations in the year 2050 follow IPCC's Representative Concentration Pathway RCP8.5 (Collins et al., 2013). To prescribe future SIC and SST, we used results from an Earth System Model (ESM; see Section 2.3) simulation. The same prescribed aerosol emissions are used as in 2004. Therefore, all anthropogenic aerosol emissions between **arctic_2004** and **arctic_2050_EM2004** are identical.

- **arctic_2050**: The same as **arctic_2050_EM2004** but the prescribed aerosol emissions are representative for 2050 (RCP8.5). The emission factors of $SO_2$ and OC ship emissions are smaller than in **arctic_2050_EM2004** since regulations and technological improvements are taken into account. Additional Arctic ship emissions are not accounted for.

- **arctic_2050_shipping**: The same as **arctic_2050** but with additional ship emissions in the Arctic. These emissions are estimated from Peters et al. (2011, see Section 2.1.2) based on future transport and oil/gas extraction. Since these additional Arctic ship emissions induced no significant changes in our test simulations (not shown), we increased the emissions by a factor of ten (mass flux). By comparing **arctic_2050** with **arctic_2050_shipping**, we can estimate the impact of future Arctic ship emissions enabled by the smaller SIC.

Each simulation is run for 20 years with the same forcing for each year, therefore yielding 20 ensemble members.

**Table 1.** An overview of the different model simulations.

| Model simulation | Greenhouse gas concentrations | SIC/SST | Ship emissions | Other anthropogenic aerosol emissions |
|---|---|---|---|---|
| **arctic_2004** | Year 2004 | Year 2004 (AMIP) | Year 2004 (Dalsøren et al., 2009) | Year 2004 (ACCMIP) |
| **arctic_2050_EM2004** | Year 2050 (RCP8.5) | Year 2050 (MPI-ESM RCP8.5) | Year 2004 (Dalsøren et al., 2009) | Year 2004 (ACCMIP) |
| **arctic_2050** | Year 2050 (RCP8.5) | Year 2050 (MPI-ESM RCP8.5) | Dalsøren et al. (2009) with emission factors for 2050 | Year 2050 (ACCMIP RCP8.5) |
| **arctic_2050_shipping** | Year 2050 (RCP8.5) | Year 2050 (MPI-ESM RCP8.5) | Dalsøren et al. (2009) with emission factors for 2050 and additional ship emissions by Peters et al. (2011) | Year 2050 (ACCMIP RCP8.5) |

## 2.3 Boundary conditions

Both SIC and SST are prescribed in ECHAM6-HAM2. For future conditions, we used model results from the Earth System Model MPI-ESM as input (simulation for the climate model intercomparison project phase 5 (CMIP5), RCP8.5; Giorgetta et al., 2013). We chose MPI-ESM because its atmospheric component is ECHAM and the simulated future sea ice retreat is

close to the model median of CMIP5. An inconsistency in our simulations is that we did not apply the SST and SIC mid-month correction to the MPI-ESM data as recommended by Taylor et al. (2000), which is applied for the AMIP data that we used for the year 2004 (Taylor et al., 2000). Therefore, the seasonal variability in SIC and SST is somewhat underestimated in 2050. However, compared to the large differences in SIC and SST between 2004 and 2050, we do not expect this to affect the main conclusions of our study.

As mentioned previously, future greenhouse gas emissions follow the RCP8.5 scenario, which shows a similar $CO_2$ emission increase as the A2 scenario that Peters et al. (2011) assumed in their analysis. From 2004 to 2050, the global greenhouse gas volume mixing ratios change as follows: from 377 ppm to 541 ppm for $CO_2$, from 1.76 ppm to 2.74 ppm for $CH_4$, from 319 ppb to 367 ppb for $N_2O$, from 256 ppt to 107 ppt for CFC-11, and from 540 ppt to 345 ppt for CFC-12 (CFCs are chlorofluorocarbons). Also most prescribed aerosol emissions (excluding DMS terrestrial emissions, biogenic organic carbon

emissions, and ship emissions) follow RCP8.5, which decline in most industrial sectors from 2004 to 2050.

We refrained from averaging SIC and SST over several years (e.g. 2000-2010) to avoid having spurious regions with intermediate SIC and SST. However, the interannual variability in SIC is pronounced, therefore we performed test simulations using SIC and SST from: i) the years 2003 and 2004 from AMIP and ii) the first and the second ensemble members from the MPI-ESM CMIP5 simulation for the year 2050. Overall, the Arctic SIC in 2003 was somewhat smaller than in 2004, and the

SIC in the first ensemble member from MPI-ESM was smaller than in the second ensemble member. We found that the basic results and main conclusions do not depend on these differences in SIC but looking at only two years for both present-day and future might not be sufficient to confirm that all our results are robust. In the following, we will always refer to the simulations using SIC and SST from 2004 and future SIC and SST from the first ensemble member of MPI-ESM.

To verify consistency between future shipping routes and sea ice extent, we further compared the sea ice conditions used to calculate future ship emissions with the sea ice conditions employed in our simulations (Appendix B).

## 2.4 Statistical test

Wilks (2016) recently pointed out that the approach to accept alternative hypotheses at any gridpoint where locally significant results occur (which is commonly used in atmospheric sciences) leads to overstatements of scientific results: with this so-called

"naive stippling approach", a significance test is calculated for every gridpoint and all gridboxes are stippled where the p-value is smaller than $5\%$ (for a significance level of $\alpha = 0.05$). This approach has two main limitations: 1. Assuming that the spatial correlation is zero, $5\%$ of the gridboxes show on average stippling *just by chance*. 2. Spatial autocorrelation – often large when analysing gridded climate data – increases the false discovery rate (FDR) for the "naive stippling approach", i.e. the null hypothesis is often rejected although it is true. As suggested by Wilks (2016), we circumvent the problem by controlling

the FDR instead. The advantages of this approach are the elimination of many spurious signals and the robustness concerning spatial correlation. In this method, a threshold p-value is calculated below which the result is supposed to be signal, not noise. We assume that the spatial correlation is moderate or large for the variables we are looking at. Therefore, we set $\alpha_{FDR}$ to $2 \cdot \alpha$ (see Wilks, 2016, for explanation). For the individual gridpoints, p-values are calculated using the Wilcoxon-Mann-Whitney test instead of the often used Welch's test since the latter is only valid if the samples are normally distributed (a condition which

was sometimes not met in our results). The only exception where we used the Welch's test is for testing the significance of the results from the cloud radiative kernel method (see Appendix C): we could not apply the Wilcoxon-Mann-Whitney test to the cloud radiative kernel results because they are given as differences instead of absolute values. Throughout this paper, the term "significant" is interchangeable with "statistically significant".

## 3 Results and Discussion

First, the changes in natural aerosol populations, clouds, and their radiative forcings/effects in a warming Arctic are assessed (Sect. 3.1). Second, we determine the influence of additional Arctic shipping activity related to transit shipping and petroleum activites on climate (Sect. 3.2).

Most figures show the mean over the twenty ensemble members for the reference simulation on the left and differences between the perturbed ensemble mean and the reference ensemble mean on the right. As mentioned previously, we analyse

the months July to October. Since the conditions change considerably from July to October, averaging over these four months might hide significant changes occurring in only one or two months. Therefore, we decided to average the results from July to August (late summer) and from September to October (early autumn). If the season is not specified in the text, results refer

to both late summer and early autumn. Most of the figures show results for early autumn, except for changes in clouds and $RF_{ari}$ associated with enhanced Arctic shipping, which refer to late summer. When we compare our results to other studies, we average over the same time and area as the authors of the corresponding study did for a meaningful comparison.

Each simulation consists of twenty ensemble members to account for the high variability in Arctic climate. However, uncertainties associated with the used climate model can of course not be captured with this approach. It is well known that different global climate models deviate considerably, e.g. when simulating aerosol-cloud interactions. Furthermore, models of different resolutions generally have problems to reproduce the structure of mixed-phase clouds prevalent in the Arctic (Morrison et al., 2009; Klein et al., 2009; Fan et al., 2011; Morrison et al., 2011; Possner et al., 2017), and the future sea ice extent as well as the prescribed aerosol emissions are highly uncertain (Collins et al., 2013). To gain a better understanding of the robustness of our results, we compare them with other studies, both concerning relative and absolute changes. In addition, we provide in the Supplementary Information a comparison of the simulation **arctic_2004** with Arctic observations. While the simulated ice water path (IWP) and the aerosol optical thickness (AOT; at least in some Arctic regions) have a low bias, the surface concentrations of BC and sulphate, the liquid water path (LWP), the cloud cover, and the SW, LW, and net CREs at the surface and the TOA agree well with the observations. The underestimation of AOT in our model is probably a combination of several causes, including e.g. missing local aerosol sources in the model (e.g. marine organics or gas flaring emissions; Hawkins and Russell, 2010; Chang et al., 2011; Stohl et al., 2013), an underestimation of aerosol transport from midlatitudes to the Arctic (Bourgeois and Bey, 2011), uncertainties in the optical properties and emissions of aerosols (e.g. for BC, see Bond and Bergstrom, 2006; Bond et al., 2013), and the neglection of spume drops in the sea salt parameterisation by Long et al. (2011). In general, it is very likely that our model underestimates the total aerosol concentrations in the Arctic.

## 3.1 Changes due to warming and sea ice retreat

In the following, we analyse how a future temperature increase in the Arctic affects natural aerosol particles, clouds, and radiation. For that, simulation **arctic_2050_EM2004** is compared with **arctic_2004**. The Arctic sea ice area decreases from $6.1 \cdot 10^6 \, \mathrm{km}^2$ to $3.4 \cdot 10^6 \, \mathrm{km}^2$ and from $5.7 \cdot 10^6 \, \mathrm{km}^2$ to $2.3 \cdot 10^6 \, \mathrm{km}^2$ in late summer and early autumn, respectively. To gain some insight into the importance of this retreat in sea ice, we averaged some vertically integrated variables such as AOT or CDNC burden over four different regions north of $60°$ N (see Tables 2 and 3 for late summer and early autumn, respectively): i) the whole region north of $60°$ N, ii) regions with open ocean in both 2004 and 2050 (SIC $< 0.5$), iii) regions with sea ice coverage in both 2004 and 2050 (SIC $> 0.5$), and iv) regions that are covered with sea ice in 2004 (SIC $> 0.5$), but not anymore in 2050 (SIC $< 0.5$). This analysis is only qualitative since advection can hide significant changes related to the sea ice retreat, the SIC values used for the calculations are monthly means, and the threshold of $0.5$ for SIC to differentiate open ocean and sea ice is somewhat arbitrary.

### 3.1.1 Aerosol particles

Over the central Arctic Ocean, the decrease in SIC (Fig. 2) enables emission fluxes of DMS and sea salt, which significantly increase their burdens (Supplementary Fig. 4; Tables 2, 3). As a second-order effect, significant increases in $u_{10}$ (Supplementary

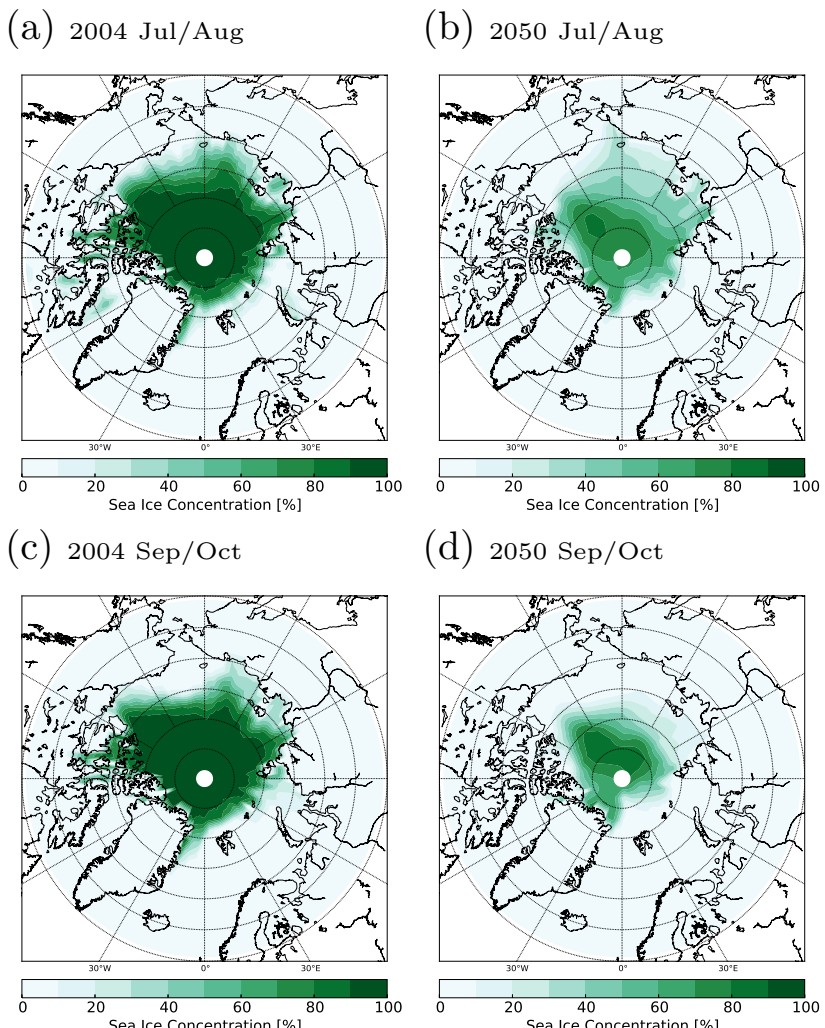

**Figure 2.** SIC in 2004 and 2050 for late summer (Jul/Aug) and for early autumn (Sep/Oct).

Fig. 5) over the central Arctic Ocean in early autumn increase sea salt and DMS emissions. In regions where the SIC does not change, both changes in $u_{10}$ (insignificant) and changes in SST (Supplementary Fig. 6) affect DMS and sea salt emissions, and thus their burdens. For example, the decrease in the sea salt burden over the Bering Strait is due to decreases in SST (caused by a model bias in the MPI-ESM sea surface temperature compared to AMIP) and $u_{10}$.

5    Despite the pronounced increases in DMS burden, the sulphate burden shows no large changes since it is dominated by other emissions (e.g. anthropogenic $SO_2$ emissions; not shown). Also the aerosol size distributions at $950\,\mathrm{hPa}$ (corresponding

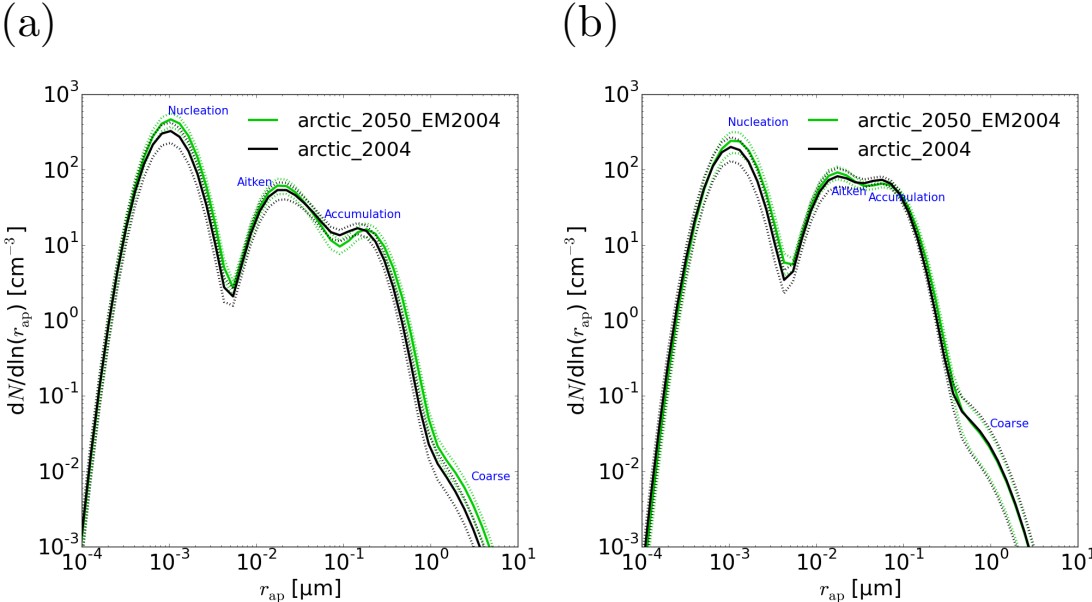

**Figure 3.** Aerosol number size distributions in 2004 (**arctic_2004**) and 2050 (**arctic_2050_EM2004**); $N$ stands for the number concentration (assuming that air density $\rho_{air} \approx 1\,kg\,m^3$), $r_{ap}$ for the radius of the aerosol particles. The size distributions are shown for early autumn (Sep/Oct) at $950\,hPa$ (a) and $800\,hPa$ (b), averaged between $75°$ and $90°\,N$. The solid lines denote ensemble means, the dotted lines the subtracted/added standard deviations. Different colors (black, green) stand for different simulations (see legend).

to $\approx 540\,m$; Fig. 3a) and $800\,hPa$ (corresponding to $\approx 1950\,m$; Fig. 3b) show only small, non-significant changes from 2004 to 2050 (shown for early autumn; averaged between $75°$ and $90°\,N$). The number concentration slightly increases in the nucleation mode in both seasons, which we attribute to the enhanced DMS emissions. DMS is oxidised via $SO_2$ to sulphuric acid, which can form new particles. In late summer, the number concentration in the Aitken mode increases to some extent. In

5 early autumn, the number concentration decreases at $r_{ap} \approx 0.1\,\mu m$ ($r_{ap}$ is the radius of the aerosol particles), which might be caused by decreases in BC and OC burdens (not shown), but increases in the coarse mode. The smaller BC and OC burdens can be explained by the increase in precipitation, which leads to enhanced wet deposition (the BC and OC emissions are identical between the two simulations). The increased number in the coarse mode can be explained by the increase in sea salt emissions.

Struthers et al. (2011) compared sea salt emissions for a nearly ice-free summer (2100) with present-day conditions (2000)

10 and found an increase in mass emissions by a factor of $\approx 4$ (present-day value $7.1\,\mu g\,m^{-2}\,s^{-1}$); this is an average over JJA (June, July, August) and $70°$ to $90°\,N$. Note that we chose 2050 for our simulations due to the availability of Arctic ship emissions for this year. In the same region, Browse et al. (2014) found that sea salt emissions increased by a factor of 10 (present-day value $6.9 \cdot 10^{-3}\,\mu g\,m^{-2}\,s^{-1}$) in August when comparing a hypothetically ice-free ocean with present-day conditions (2000). In

**Table 2.** Absolute values for the year 2004 and differences between 2050 and 2004 (i.e. between simulations **arctic_2050_EM2004** and **arctic_2004**) for sea salt burden, DMS burden, AOT, LWP, IWP, cloud cover ("CC"), in-cloud CDNC burden, and $T_{surf}$ for late summer (Jul/Aug). The numbers are averaged over four regions between $60°$ and $90°$ N: i) the whole region, ii) gridboxes which are ocean in both 2004 and 2050 (SIC < 0.5; "Ocean"), iii) gridboxes which are covered by sea ice in both 2004 and 2050 (SIC > 0.5; "Sea ice"), and iv) gridboxes which have sea ice in 2004 (SIC > 0.5) but not in 2050 (SIC < 0.5; "Transition"). Significant changes are marked with a star. Note that the SST is prescribed, i.e. shows no interannual variability.

| | Total region | | Ocean | | Sea ice | | Transition | |
|---|---|---|---|---|---|---|---|---|
| | 2004 | 2050 − 2004 | 2004 | 2050 − 2004 | 2004 | 2050 − 2004 | 2004 | 2050 − 2004 |
| Sea salt ($10^{-7}\,\mathrm{kg\,m^{-2}}$) | 1.2 | 0.18* | 3.0 | 0.36* | 0.18 | 0.12* | 0.29 | 0.28* |
| DMS ($10^{-7}\,\mathrm{kg\,m^{-2}}$) | 1.5 | 0.27* | 3.2 | 0.39* | 0.66 | 0.42* | 0.92 | 0.73* |
| AOT ($10^{-2}$) | 3.6 | 0.26* | 3.9 | 0.19 | 1.3 | 0.19* | 1.6 | 0.19* |
| LWP ($\mathrm{g\,m^{-2}}$) | 96 | 8.0* | 108 | 7.3* | 65 | 5.1* | 67 | 7.9* |
| IWP ($\mathrm{g\,m^{-2}}$) | 17 | 0.00 | 15 | 0.00 | 12 | 0.09 | 14 | -0.06 |
| CC (%) | 77 | 0.08 | 81 | 0.85 | 88 | -0.35 | 82 | 0.50 |
| CDNC ($10^{10}\,\mathrm{m^{-2}}$) | 6.0 | 0.47* | 5.1 | 0.30* | 1.9 | 0.22* | 2.4 | 0.34* |
| $T_{surf}$ (K) | 281 | 0.98* | 278 | 1.6* | 273 | -0.36* | 272 | 0.23* |

**Table 3.** As Table 2 but for early autumn (Sep/Oct).

| | Total region | | Ocean | | Sea ice | | Transition | |
|---|---|---|---|---|---|---|---|---|
| | 2004 | 2050 − 2004 | 2004 | 2050 − 2004 | 2004 | 2050 − 2004 | 2004 | 2050 − 2004 |
| Sea salt ($10^{-7}\,\mathrm{kg\,m^{-2}}$) | 2.7 | 0.84* | 6.6 | 2.0* | 0.29 | 0.27* | 0.45 | 0.53* |
| DMS ($10^{-7}\,\mathrm{kg\,m^{-2}}$) | 0.62 | 0.12* | 1.1 | 0.18* | 0.32 | 0.24* | 0.47 | 0.34* |
| AOT ($10^{-2}$) | 3.2 | 0.28* | 3.5 | 0.32 | 1.3 | 0.03 | 1.4 | 0.21* |
| LWP ($\mathrm{g\,m^{-2}}$) | 72 | 5.3* | 92 | 2.0 | 24 | 14* | 37 | 19* |
| IWP ($\mathrm{g\,m^{-2}}$) | 21 | 0.59* | 21 | 0.79* | 12 | 0.17 | 14 | 0.76 |
| CC (%) | 87 | 0.05 | 89 | -0.70* | 92 | 1.3* | 92 | 2.3* |
| CDNC ($10^{10}\,\mathrm{m^{-2}}$) | 4.0 | 0.31* | 4.3 | 0.30 | 0.96 | 0.28* | 1.3 | 0.47* |
| $T_{surf}$ (K) | 271 | 2.8* | 277 | 1.8* | 258 | 7.9* | 264 | 7.4* |

our simulations ($70°$ to $90°$ N), sea salt emissions increase by a factor of 1.8 and 1.7 in JJA and August by 2050, respectively, compared to the present-day values of $1.52 \cdot 10^{-3}$ and $2.42 \cdot 10^{-3}\,\mu\mathrm{g\,m^{-2}\,s^{-1}}$. The relative increase in emissions is largest in the study by Browse et al. (2014), where the absolute decrease in SIC is largest, and smallest in our study, where the absolute decrease in SIC is smallest. Present-day emissions are a factor of $\approx 3$ lower in our simulations compared with Browse et al.

(2014), which results from the differences in the two parameterisations (Gong, 2003; Long et al., 2011, with SST corrections) as shown in the study of Long et al. (2011). The absolute present-day emissions reported by Struthers et al. (2011) are at least three orders of magnitudes higher than in our simulations. This might again be caused by the parameterisations used since differences in $u_{10}$ and SST are too small to explain the large disagreement. Struthers et al. (2011) used a modification of the Mårtensson parameterisation combined with the Monahan parameterisation for particles $> 1.4\,\mu\mathrm{m}$ (Mårtensson et al., 2003;

Monahan et al., 1986). However, neither Long et al. (2011) using the Mårtensson parameterisation nor us using the Monahan parameterisation for particles $r_{\mathrm{dry}} < 4\,\mu\mathrm{m}$ (in earlier simulations with ECHAM-HAM; not shown) found emissions as high as Struthers et al. (2011). Therefore, we expect that differences in the number fluxes of large particles ($> 4\,\mu\mathrm{m}$), which contribute most to mass emissions (Long et al., 2011), are responsible for the large discrepancy. When we compare our simulated mass emissions in the Arctic ($60°$ to $90°$ N) from July to October with the 11 CMIP5 models that provide sea salt mass emission

fluxes, our model shows the lowest sea salt emissions: we arrive at a value of $5.9 \cdot 10^{-3}\,\mu\mathrm{g\,m^{-2}\,s^{-1}}$ under present-day conditions, while the CMIP5 models emit $\approx 4 \cdot 10^{-2}\,\mu\mathrm{g\,m^{-2}\,s^{-1}}$ (GISS-E2-H, GISS-E2-R, MIROC-ESM, MIROC-ESM-CHEM), 7 to $9 \cdot 10^{-2}\,\mu\mathrm{g\,m^{-2}\,s^{-1}}$ (MIROC5, NorESM1-M, NorESM-ME), and 1 to $2 \cdot 10^{-1}\,\mu\mathrm{g\,m^{-2}\,s^{-1}}$ (GFDL-CM3, MIROC4h, MRI-CGCM3, MRI-ESM1). Our simulated absolute increases in sea salt mass emissions are therefore likely underestimated because our parameterisation does not account for the contributions from spume drops (Long et al., 2011) and thus results in

small emission fluxes of large (i.e. supermicron) aerosol particles. However, these large aerosol particles have a comparatively low impact on climate due to their low number concentrations. Since the total sea salt number emissions of the parameterisation by Long et al. (2011) are not generally lower than in other parameterisations, we do not expect that our simulated impact on CCN and radiation is completely different compared with other sea salt parameterisations. To confirm this, we conducted an additional simulation similar to **arctic_2004**, but with the old standard sea salt parameterisation of ECHAM-HAM (i.e. the

parameterisation by Guelle et al., 2001). This parameterisation results in considerably higher sea salt mass emissions than the parameterisation by Long et al. (2011) ($9.8 \cdot 10^{-2}\,\mu\mathrm{g\,m^{-2}\,s^{-1}}$ averaged from $60°$ to $90°$ N and from July to October). Nevertheless, the resulting AOT and CDNC are quite comparable: using the parameterisation by Guelle et al. (2001), the AOT is somewhat higher in the Arctic than with the parameterisation by Long et al. (2011) (0.039 compared to 0.034; averaged from July to October), while the CDNC burden is slightly lower ($4.8 \cdot 10^{10}\,\mathrm{m^{-2}}$ compared to $5.0 \cdot 10^{10}\,\mathrm{m^{-2}}$; in-cloud values).

### 3.1.2  Clouds

Except for cloud cover, LWP, and IWP, the averages of cloud properties (such as LWC or CDNCs) refer to in-cloud values, i.e. by averaging only over periods and locations when and where clouds are present.

In general, the number of aerosol particles acting as CCN increases in the future, which leads to enhanced CDNCs (Fig. 4d). The increase in the number of CCN is not only caused by the increases in oceanic aerosol emissions but also by changes

in meteorology: the updrafts available for activation increase in the boundary layer between $75°$ and $90°$ N in early autumn (Supplementary Fig. 7), which supports the formation of cloud droplets in this region. Averaged between $75°$ and $90°$ N, the CDNC burden increases by 10% and 29% in late summer and early autumn, respectively. Relative changes are largest in regions where sea ice melted (Tables 2, 3). Also LWC increases (see Fig. 4b) because both the open ocean and higher air temperatures increase the specific humidity. The increase in LWC can be ascribed to both higher CDNCs and larger cloud droplets (not shown). Averaged between $75°$ and $90°$ N, LWP increases by 10% in late summer and by 34% in early autumn. Precipitation shows significant increases in early autumn (Supplementary Fig. 8). In late summer, changes are only significant when averaged between $60°$ and $90°$ N and smaller than in early autumn (+4% compared to +9%).

We also obtain increased CDNCs (which we attribute to increased CCN concentrations) when averaging over all-sky conditions. In contrast, Browse et al. (2014) found small decreases in CCN concentrations (also averaged over all-sky conditions) over the Arctic Ocean. In their simulations, the liquid clouds over the ocean suppressed new particle formation via aqueous phase oxidation of $SO_2$ (a process also considered in ECHAM6-HAM2). Instead, particles grew to larger sizes and were efficiently scavenged by drizzle. The different responses when compared to our simulations could e.g. be caused by different oxidant concentrations ($H_2O_2$, $O_3$) or by the different handling of drizzle and precipitation: Browse et al. (2014) derived drizzle rates from Arctic observations of cloud altitude and droplet concentrations and scaled them by the low-cloud fraction. On the other hand, cloud microphysical processes (e.g. diffusional growth, coagulation) are explicitly calculated in our simulations and coupled with aerosol particles via Köhler theory and freezing parameterisations. Drizzle is not considered as a separate size class in our simulations; however, Sant et al. (2015) showed that the impact of drizzle on the CDNC burden is rather small in the Arctic in ECHAM5-HAM.

As expected, the higher temperatures in 2050 influence the occurrence of cloud ice (both cirrus and mixed-phase) in our simulations by shifting the isotherms and thus also cloud ice towards higher altitudes. Changes in ice water content (IWC) (Fig. 5b) can be caused by changes in the ICNC (Fig. 5d) and/or the effective ice crystal radius (Fig. 5f). Both changes in the ICNC and radius have a considerable influence at altitudes below $500\,\mathrm{hPa}$, whereas changes in radius dominate at higher altitudes. The increase of ICNC near the surface is mainly caused by enhanced convection, which leads to small but numerous ice crystals following the temperature-dependent empirical parameterisation of Boudala et al. (2002).

Compared to the pronounced increases in LWP, changes in the IWP are small and only significant over the whole Arctic region and over the ocean in early autumn (slight increases; see Tables 2, 3). This can be explained by two opposing effects: on the one hand, the total water path increases due to the higher specific humidity. On the other hand, the temperature increase leads to a higher fraction of liquid water to the total water path. In our simulations, the first effect slightly dominates in early autumn. The absolute changes might be underestimated since our model in general underestimates the ice water content of clouds.

Especially in early autumn, significant changes in cloud cover occur (see Fig. 6). Cloud cover decreases where convective precipitation is most enhanced (e.g. near Svalbard; see Supplementary Fig. 9) but increases where sea ice vanished, e.g. over the East Siberian Sea and the Beaufort Sea (Fig. A1 shows a map of the Arctic Ocean where the regional seas are labelled). When averaged over the open ocean area, cloud cover shows rather small but significant decreases in early autumn, whereas

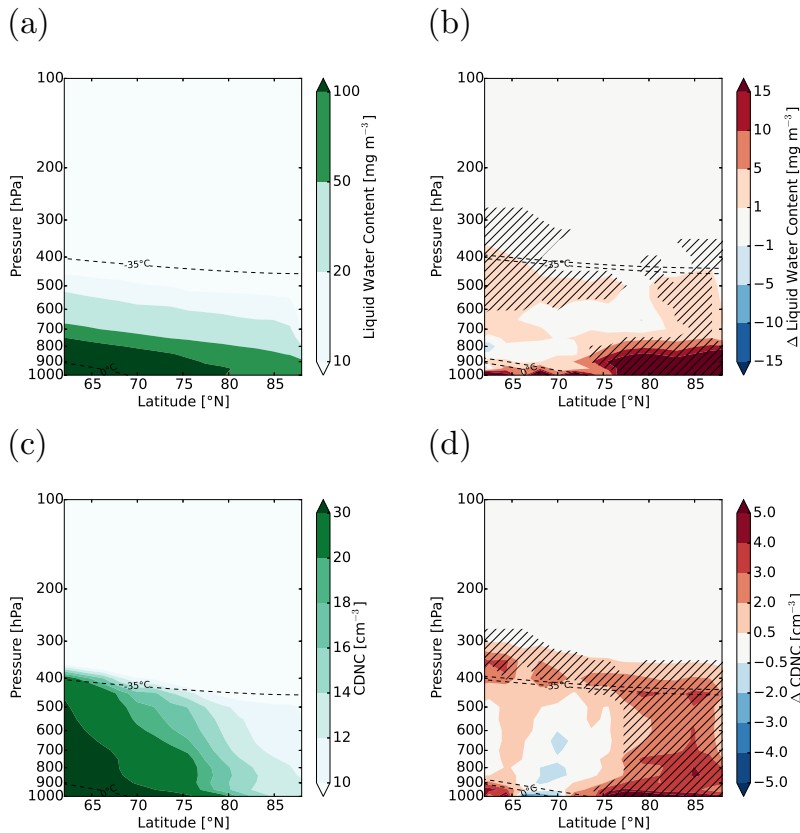

**Figure 4.** LWC and CDNC in 2004 in (a)/(c) and differences between 2050 and 2004 (i.e. between simulations **arctic_2050_EM2004** and **arctic_2004**) in (b)/(d) (in-cloud values) in early autumn (Sep/Oct). Hatched areas are significant at the 95% confidence level. The dashed lines show the $0°C$ and the $-35°C$ isotherms.

it increases significantly and pronouncedly where sea ice melted (Table 3). The latter is consistent with the findings from Abe et al. (2016), who found increases in the October cloud cover caused by sea ice reduction, which leads to an enhanced moisture flux to the atmosphere. Also in our simulations, the surface fluxes increase significantly over regions where sea ice melted (not shown).

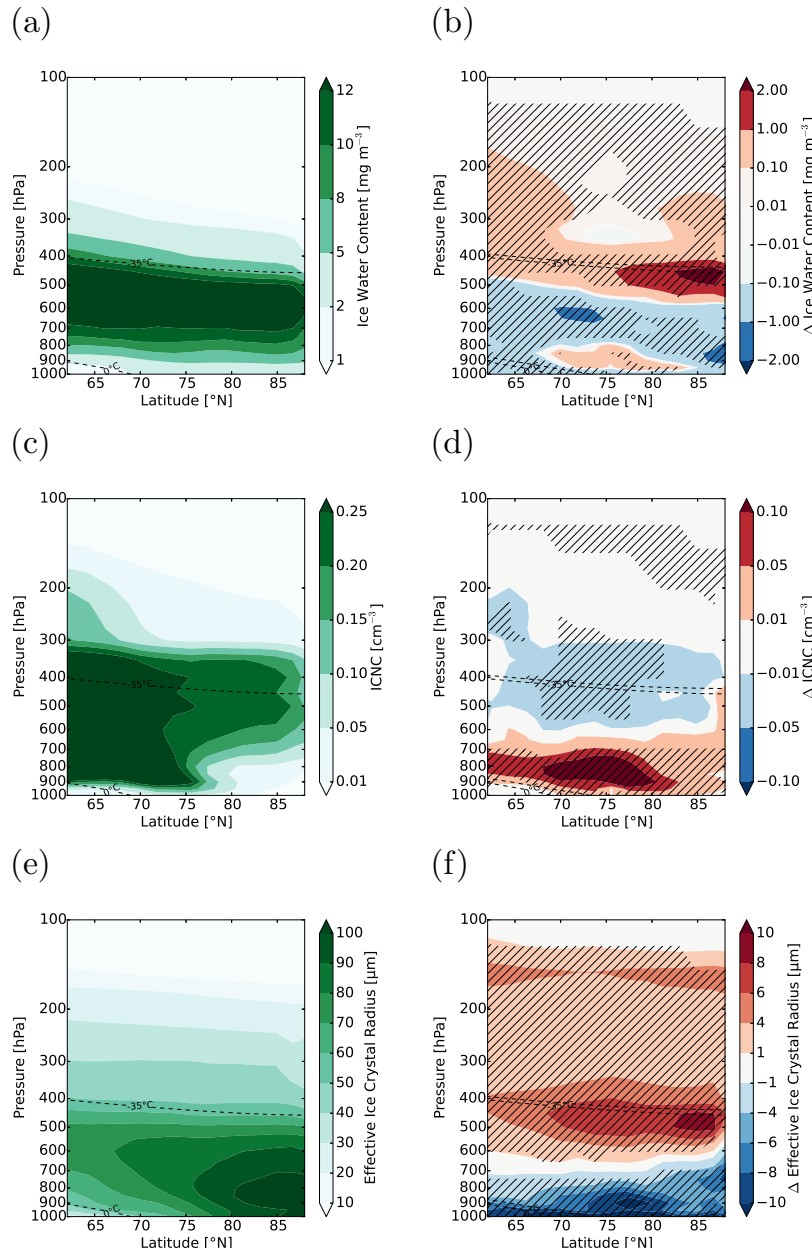

**Figure 5.** IWC, ICNC, and effective ice crystal radius in 2004 in (a)/(c)/(e) and differences between 2050 and 2004 (i.e. between simulations **arctic_2050_EM2004** and **arctic_2004**) in (b)/(d)/(f) (in-cloud values) in early autumn (Sep/Oct). Hatched areas are significant at the 95% confidence level. The dashed lines show the $0°C$ and the $-35°C$ isotherms. Note that they are zonally and temporally averaged, hence ice can exist at altitudes below the $0°C$ isotherm.

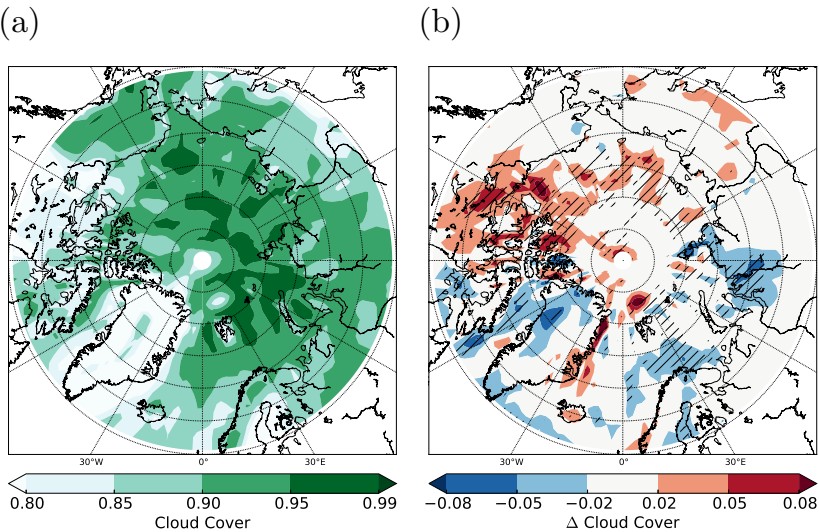

**Figure 6.** (a) Cloud cover in 2004 and (b) differences between 2050 and 2004 (i.e. between simulations **arctic_2050_EM2004** and **arctic_2004**) in early autumn (Sep/Oct). Hatched areas are significant at the 95% confidence level.

### 3.1.3 Aerosol radiative forcings

Unless otherwise stated, all aerosol radiative forcings and cloud radiative effects refer to those at the top of the atmosphere (TOA). As mentioned previously, $RF_{ari}$ refers to the instantaneous effect of all aerosols on radiation. In 2004, aerosol particles have a negative $RF_{ari}$ and thus cool the Arctic under clear-sky conditions (i.e. absence of clouds; see Fig. 7c), except over

sea ice and Greenland, where the surface albedo is high (see Fig. 7a). If the presence of clouds is considered, aerosol particles warm the atmosphere also over Alaska and northeast Siberia (late summer) and over the whole northern Russia (early autumn; shown in Fig. 7e). Part of this warming might be caused by BC and dust aerosols above clouds (Supplementary Fig. 10): the clouds reflect more SW radiation than the snow/ice-free surface and part of the scattered SW radiation can also be absorbed by aerosol particles causing an increase in aerosol absorption as compared to clear-sky conditions (see e.g. Myhre et al., 1998).

Moreover, the scattering of aerosol particles could become less important in the presence of clouds, which increases the relative importance of aerosol absorption to extinction. Averaged over the whole Arctic region, aerosol particles have a cooling effect under clear-sky conditions in 2004 ($-1.23\,\mathrm{W\,m^{-2}}$ for late summer and $-0.65\,\mathrm{W\,m^{-2}}$ for early autumn) but a warming effect if clouds are considered ($0.12\,\mathrm{W\,m^{-2}}$ for late summer and $0.09\,\mathrm{W\,m^{-2}}$ for early autumn). Note that changes at the surface are of opposite sign, i.e. the aerosol particles cool the surface under all-sky conditions. The simulated AOT has a low bias in

the Arctic, which can affect these estimates of the aerosol radiative effect. Depending on whether the aerosol absorption or the scattering is underestimated, the aerosol radiative effect is either under- or overestimated. It is also possible that both effects cancel each other. In our simulations, both the cooling and the warming are more pronounced in late summer than in early autumn due to the higher solar zenith angle in late summer. Increases in the DMS and sea salt burdens increase the AOT in

2050 (significant changes from $1.6 \cdot 10^{-2}$ to $1.8 \cdot 10^{-2}$ in late summer and from $1.5 \cdot 10^{-2}$ to $1.7 \cdot 10^{-2}$ in early autumn; averaged over 75°-90° N). While the AOT does not significantly change over open ocean, it significantly increases over regions where sea ice melted (Tables 2, 3). The absorption aerosol optical thickness significantly decreases in early autumn ($1.16 \cdot 10^{-3}$ to $1.05 \cdot 10^{-3}$, averaged over 75°-90° N), which can be explained by the decrease in BC burden. In both late summer and

early autumn, $\mathrm{RF_{ari}}$ shows significant decreases under both clear-sky (Fig. 7d; shown for early autumn) and all-sky (Fig. 7f) conditions, especially in regions where the surface albedo decreased (compare Fig. 7b). We cannot distinguish between the $\mathrm{RF_{ari}}$ induced by surface albedo changes and that induced by changes in aerosols, but we expect that the increase in natural aerosol emissions decreases $\mathrm{RF_{ari}}$ since sea salt and sulphate particles are nearly pure scatterers.

The radiative forcing due to BC deposition on snow decreases significantly (see Supplementary Fig. 11) because less snow-

covered sea ice and less snow on land exist. However, the radiative forcing due to deposited BC as well as its absolute changes are small compared to other radiative forcings and CREs. This is also displayed in Tables 4 and 5, which show the area-averaged absolute differences in radiation, radiative forcings, and radiative effects north of 60° N and north of 75° N, respectively.

### 3.1.4   Cloud radiative effects

Not only the aerosol radiative forcing but also CREs change significantly. Using the radiative kernel (RK) method, we first

assess how CREs change only as a function of cloud properties (i.e. independent of changes in surface albedo or surface temperature). In this case, both the SW and the LW CRE (RK) become stronger in late summer (Tables 4, 5), for example by $-2.2\,\mathrm{W\,m^{-2}}$ for SW and $+0.88\,\mathrm{W\,m^{-2}}$ for LW when averaged between 75° and 90° N. In early autumn, changes in CREs (RK) are significant when averaged over latitudes between 75° and 90° N (but not over the whole Arctic; see Tables 4, 5), where the SW and LW CREs (RK) change by $-0.36\,\mathrm{W\,m^{-2}}$ and $-0.96\,\mathrm{W\,m^{-2}}$, respectively. These decreases in the SW CRE

(RK) north of 75° N in early autumn (see also Fig. 8c) can be attributed to increases in the cloud optical thickness and low cloud cover (cloud top altitudes below $680\,\mathrm{hPa}$; not shown). In contrast, the negative changes in LW CRE (RK) north of 75° N (see also Fig. 8f) are due to decreases in the free-tropospheric cloud cover (cloud top altitudes above $680\,\mathrm{hPa}$; not shown).

If we use the standard method for calculating CREs, which considers also impacts due to changes in surface albedo and surface temperature, changes in both SW and LW CRE are much more pronounced over the central Arctic Ocean in early

autumn than with the radiative kernel method (Fig. 8b, e). Similarly to $\mathrm{RF_{ari}}$, the large changes in SW CRE are mainly caused by the smaller surface albedo (i.e. larger changes in Fig. 8b than in Fig. 8c). In contrast, increases in LW CRE primarily result from increases in surface temperature (Supplementary Fig. 6). The significant decrease of LW CRE over the Bering Sea (which only occurs in Fig. 8e and not in Fig. 8f) can also be explained by changes in surface temperature (a decrease in this case). Decreases in surface albedo are highly correlated with increases in surface temperature over the Arctic Ocean because the

surface temperature of ice (which can be much lower than $270\,\mathrm{K}$ in early autumn, e.g. due to the ice-albedo feedback) changes to the temperature of sea water (minimum temperature of $271.38\,\mathrm{K}$). Furthermore, changes in cloud cover and thickness affect both SW and LW CRE. Changes in SW and LW CRE thus mostly occur at the same locations. Since they are of opposite sign and on the same order of magnitude, they cancel to a large degree (Tables 4, 5). While regionally significant decreases and increases occur in the net CRE in early autumn, it shows no significant changes when averaged between 60°/75° and 90° N.

In late summer, the net CRE decreases significantly from 2004 to 2050 (by $-10\,\mathrm{W\,m^{-2}}$, averaged between $75°$ and $90°$ N), i.e. the cooling effect of clouds increases, even though changes in surface albedo are smaller than in early autumn (-0.12 compared to -0.21; averaged between $75°$ and $90°$ N). This is because i) the SW component dominates in these months due to the higher zonal zenith angle and ii) the surface temperature over the central Arctic Ocean does not show pronounced increases like in early autumn (Table 2), therefore not enhancing the LW CRE. The surface temperature even decreases in some regions because melt ponds on ice can have temperatures higher than $271.38\,\mathrm{K}$ (but below $273.16\,\mathrm{K}$) in late summer, while the SST is $271.38\,\mathrm{K}$ in gridboxes with $0 < \mathrm{SIC} < 1$ (equilibrium conditions, i.e. heat changes lead to changes in SIC, not SST).

Compared with the results by Struthers et al. (2011), our changes in the SW CRE are rather small: averaged between $70°$ and $90°$ N (JJA), the radiative effect increases from $-63.7$ to $-107.7\,\mathrm{Wm^{-2}}$ (i.e. change by $-44\,\mathrm{Wm^{-2}}$) and from $-47.1$ to $-55.4\,\mathrm{Wm^{-2}}$ (i.e. change by $-8.3\,\mathrm{Wm^{-2}}$) in their and our simulations, respectively. The larger relative change reported by Struthers et al. (2011) is likely caused by the larger decrease in SIC (and, thus, albedo): while still considerable parts of the Arctic Ocean are covered by sea ice in our simulations in 2050 (especially in June and July), only little sea ice is left in the simuations by Struthers et al. (2011) in 2100. For present-day, the absolute estimates of SW CRE by the two models are similarily close to the satellite-derived value by the Clouds and the Earth's Radiant Energy System (CERES), which is $-56.8\,\mathrm{Wm^{-2}}$ averaged over the same months and latitudes for the period July 2005 to June 2015 (Loeb et al., 2018).

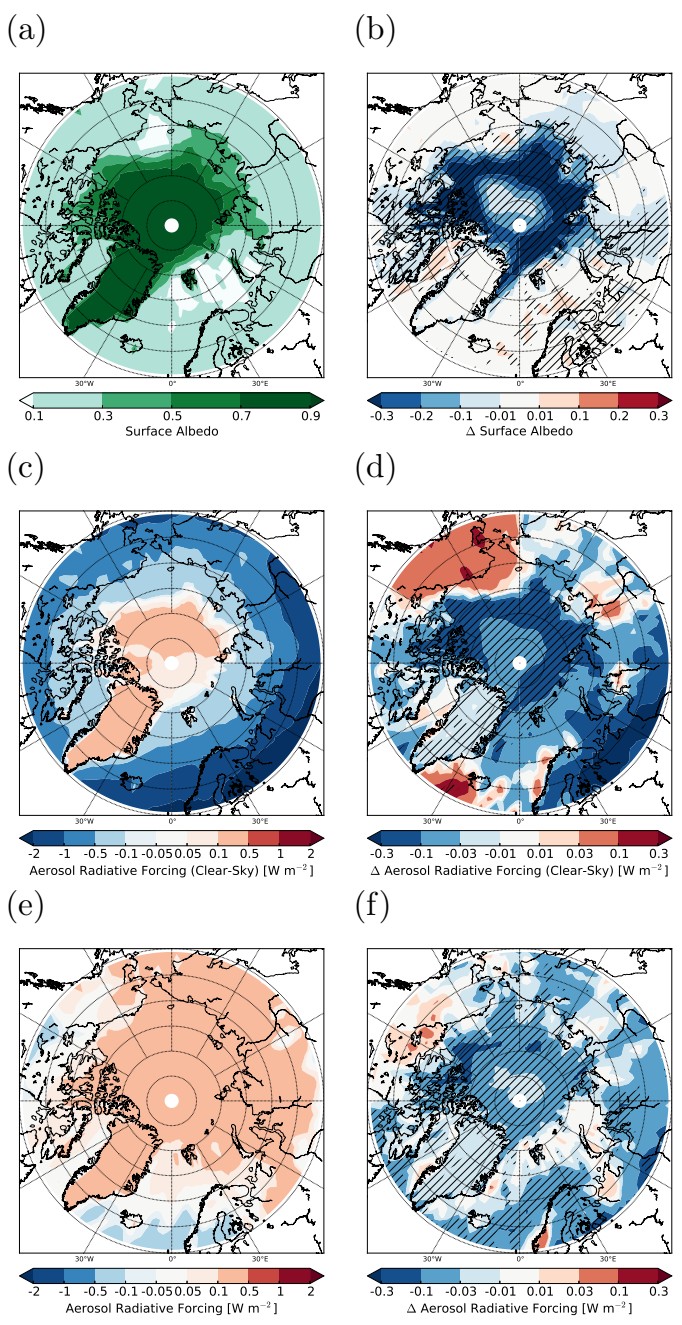

**Figure 7.** Surface albedo, aerosol net radiative forcing (clear-sky), and aerosol net radiative forcing (all-sky) in 2004 in (a)/(c)/(e) and differences between 2050 and 2004 (i.e. between simulations **arctic_2050_EM2004** and **arctic_2004**) in (b)/(d)/(f) in early autumn (Sep/Oct). Hatched areas are significant at the 95% confidence level.

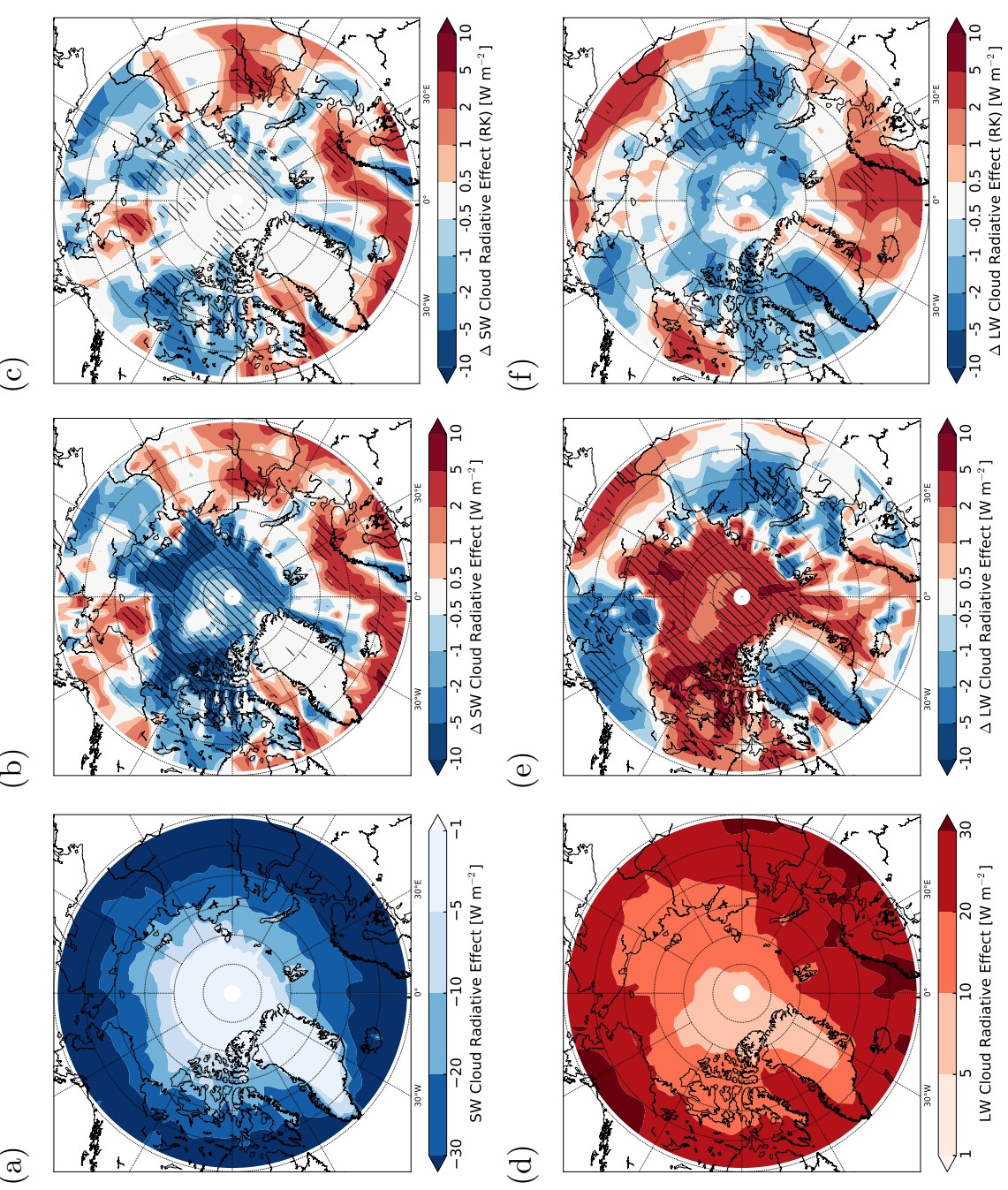

**Figure 8.** SW and LW CRE in 2004 in (a)/(d) and differences between 2050 and 2004 (i.e. between simulations **arctic_2050_EM2004** and **arctic_2004**) in (b)/(c)/(e)/(f) in early autumn (Sep/Oct). In (b) and (e), the changes in CREs were calculated online from two radiation calls (once with, once without clouds). In (c) and (f), the changes in CREs were calculated with the radiative kernel (RK) method (see text for more details). Hatched areas are significant at the 95% confidence level.

**Table 4.** Absolute values for the year 2004 and differences between 2050 and 2004 (i.e. **arctic_2050_EM2004**−**arctic_2004**) in radiation, radiative forcings, and CREs (in $\mathrm{W\,m^{-2}}$) averaged over all latitudes north of $60° \mathrm{N}$ in late summer (Jul/Aug) and early autumn (Sep/Oct). **arctic_2050_EM2004**−**arctic_2004** accounts for changes between 2050 and 2004 associated with a warmer climate, which leads to a reduction in SIC and therefore increased natural aerosol emissions. RK stands for radiative kernel method (see text for details). The star (*) denotes changes that are significant at $\alpha = 5\,\%$.

| | Late summer (2004) | Late summer (2050 − 2004) | Early autumn (2004) | Early autumn (2050 − 2004) |
|---|---|---|---|---|
| Net SW radiation | 233 | 3.4* | 67 | 0.70* |
| Net LW radiation | -231 | 0.60 | -202 | -1.7* |
| $RF_{ari}$ | $12 \cdot 10^{-2}$ | $-9.5 \cdot 10^{-2}$* | $9.4 \cdot 10^{-2}$ | $-3.4 \cdot 10^{-2}$* |
| BC deposition | $13 \cdot 10^{-2}$ | $0.02 \cdot 10^{-2}$ | $1.9 \cdot 10^{-2}$ | $0.49 \cdot 10^{-2}$ |
| SW CRE | -67 | -4.0* | -26 | -0.45* |
| LW CRE | 18 | -0.04 | 21 | 0.55* |
| SW CRE (RK) | -67 | -2.0* | -26 | -0.00 |
| LW CRE (RK) | 18 | 0.92* | 21 | -0.07 |

**Table 5.** The same as Table 4 but averaged over all latitudes north of $75° \mathrm{N}$.

| | Late summer (2004) | Late summer (2050 − 2004) | Early autumn (2004) | Early autumn (2050 − 2004) |
|---|---|---|---|---|
| Net SW radiation | 201 | 12* | 29 | 2.5* |
| Net LW radiation | -228 | 0.77* | -196 | -4.4* |
| $RF_{ari}$ | $53 \cdot 10^{-2}$ | $-17 \cdot 10^{-2}$* | $15 \cdot 10^{-2}$ | $-4.1 \cdot 10^{-2}$* |
| BC deposition | $21 \cdot 10^{-2}$ | $0.32 \cdot 10^{-2}$ | $2.0 \cdot 10^{-2}$ | $0.99 \cdot 10^{-2}$ |
| SW CRE | -45 | -10* | -7.8 | -2.2* |
| LW CRE | 9.3 | -0.06 | 13 | 2.0* |
| SW CRE (RK) | -45 | -2.2* | -7.8 | -0.36* |
| LW CRE (RK) | 9.3 | 0.88* | 13 | -0.96* |

## 3.2 Impact of additional ship emissions

Future sea ice retreat will enable ships to cross the Arctic Ocean, thus likely leading to enhanced shipping activity in late summer and early autumn. In this section, we will study the influence of these anthropogenic aerosol emissions on aerosol populations, clouds, and their radiative forcings/effects by comparing the simulation **arctic_2050_shipping** with **arctic_2050**.

### 3.2.1 Aerosol particles

Due to the increase in Arctic ship emissions (tenfold increase of the ship emissions by Peters et al. (2011) in 2050), the burdens of BC and sulphate are significantly enhanced in late summer (not shown). In early autumn, rises in ship-related aerosol burdens are more pronounced and also significant for OC (not shown). The maximum increases in aerosol burdens (see Fig. 9b) occur at the same locations as the ship emissions, but significant increases can spread over a large part of the Arctic (see Fig. 9c), as shown for the example of BC. The largest absolute changes in BC concentration occur near the surface, although significant changes reach altitudes as high as $400\,\mathrm{hPa}$ in early autumn (Supplementary Fig. 12d). While the changes in natural aerosol emissions (2050 versus 2004) only have a minor influence on the number size distribution (Fig. 3), the impact of increased ship emissions is considerably larger. Figure 10 shows the aerosol number size distributions averaged between $75°$ and $90°\,\mathrm{N}$, at both $950\,\mathrm{hPa}$ (corresponding to $\approx 540\,\mathrm{m}$; Fig. 10a) and $800\,\mathrm{hPa}$ (corresponding to $\approx 1950\,\mathrm{m}$; Fig. 10b) for early autumn. At $950\,\mathrm{hPa}$, the number of particles in the nucleation mode largely decreases in both seasons (Fig. 10a). For the Aitken mode, a small decrease and a distinct increase occur in late summer (not shown) and early autumn, respectively. The number concentration in the accumulation mode increases to some extent in both late summer and early autumn. At $800\,\mathrm{hPa}$ (Fig. 10b), the effect of ship emissions on the aerosol size distribution is smaller than at $950\,\mathrm{hPa}$.

The additional aerosol particles emitted by ships provide additional surfaces for the condensation of gaseous sulphuric acid. Thus, the vertically integrated condensation rate of sulphate increases where the ship emissions occur (not significant; Supplementary Fig. 13b). The vertically integrated nucleation rate of sulphate shows neither a clear decrease nor a clear increase along the shipping paths (Supplementary Fig. 13d); if the increase in condensation suppressed nucleation (as Fig. 10a suggests), we would expect a decrease in the nucleation rate. However, the vertical cross section of aerosol particles in the nucleation mode shows that the number concentration indeed decreases significantly near the surface (Supplementary Fig. 13f).

The number concentrations in the accumulation mode (and the Aitken mode in early autumn) increase both by direct emissions and by shifting aerosol particles to larger sizes due to coagulation and condensation. Since ship emissions occur near the surface, the influence at $800\,\mathrm{hPa}$ is much smaller than at $950\,\mathrm{hPa}$.

### 3.2.2 Clouds

Although ship emissions have a larger effect on aerosol burdens and size distributions in early autumn than in late summer, significant aerosol-induced changes in clouds predominantely occur in late summer. In the following, we will therefore only discuss results for late summer. The CDNC increases (Fig. 11b; increase in CDNC burden by 33% averaged between $75°$ and $90°\,\mathrm{N}$) and the effective radius decreases with additional ship emissions (Fig. 11d), consistent with the $\mathrm{RF_{aci}}$. Overall, the increase in CDNC dominates over the decrease in cloud droplet radius, leading to an enhanced LWC (Fig. 11f). We attribute this increase in LWC to a slower collision-coalescence process (cloud adjustments).

Using satellite data, Christensen et al. (2014) studied the effect of ship tracks on both mixed-phase and liquid clouds. In the late summer of 2050, the clouds that are impacted by ships in our simulations are mostly liquid. Therefore, we restrict our comparison to the influence of ships on liquid clouds. Consistent with the observations by Christensen et al. (2014),

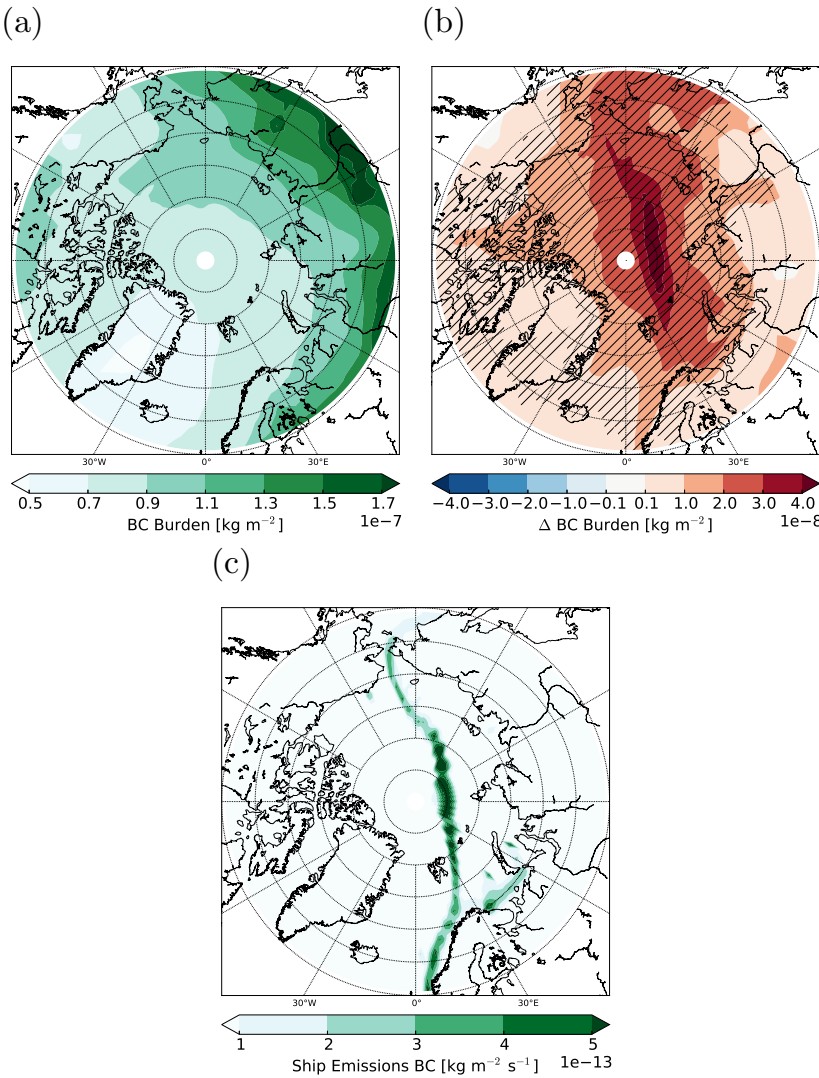

**Figure 9.** Panel (a) shows the BC burden in 2050 without considering enhanced Arctic ship emissions. Panel (b) shows the difference between a simulation with additional Arctic ship emissions and a simulation without these emissions in 2050 (difference between **arctic_2050_shipping** and **arctic_2050**). Hatched areas are significant at the 95% confidence level. Panel (c) shows tenfold higher (transit and petroleum-related) ship emissions of BC in 2050 based on the emission inventory by Peters et al. (2011). All values are for early autumn (Sep/Oct).

we also found decreases in the effective radius and increases in cloud optical thickness. The relative changes in effective radius are larger in their observations (-20% at cloud top altitude) than in our simulations (-2% to -4% at altitudes below $500\,\mathrm{hPa}$; averaged between $75°$ and $90°N$), whereas changes in cloud optical thickness compare well (+20% in both studies,

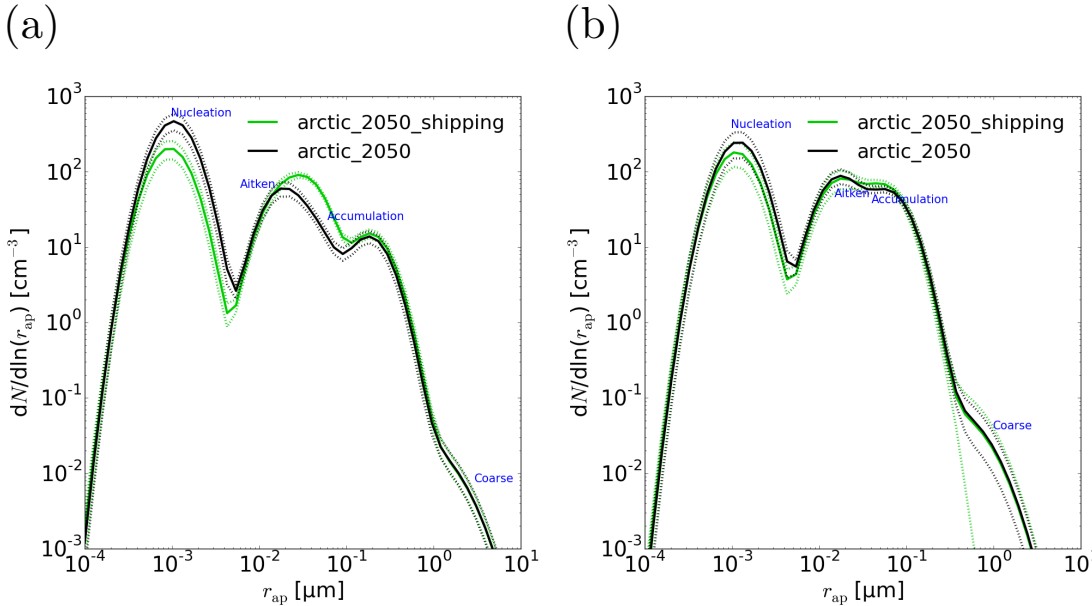

**Figure 10.** The impact of additional future ship emissions (**arctic_2050_shipping** versus **arctic_2050**) on aerosol number size distributions; $N$ stands for the number concentration (assuming that $1\,\mathrm{kg_{air}} \approx 1\,\mathrm{m^3}$), $r_{\mathrm{ap}}$ for the radius of the aerosol particles. The size distributions are shown for early autumn (Sep/Oct) at $950\,\mathrm{hPa}$ (a) and $800\,\mathrm{hPa}$ (b), averaged between $75°$ and $90°$ N. The solid lines denote ensemble means, the dotted lines the subtracted/added standard deviations. Different colors (black, green) stand for different simulations (see legend).

**Table 6.** As Table 4 but for **arctic_2050** (absolute values) and **arctic_2050_shipping**−**arctic_2050** (differences) averaged over all latitudes north of $60°$ N in late summer (Jul/Aug) and early autumn (Sep/Oct). **arctic_2050_shipping**−**arctic_2050** considers the impact of an increase in future Arctic ship emissions in 2050.

| | Late summer (2050) | Late summer ($2050_{\mathrm{ship}} - 2050$) | Early autumn (2050) | Early autumn ($2050_{\mathrm{ship}} - 2050$) |
|---|---|---|---|---|
| Net SW radiation | 238 | -3.0* | 68 | -0.46* |
| Net LW radiation | -231 | -0.01 | -204 | 0.32 |
| RF$_{\mathrm{ari}}$ | $11 \cdot 10^{-2}$ | $0.79 \cdot 10^{-2}$ | $4.1 \cdot 10^{-2}$ | $1.1 \cdot 10^{-2}$ |
| BC deposition | $12 \cdot 10^{-2}$ | $-0.26 \cdot 10^{-2}$ | $2.1 \cdot 10^{-2}$ | $0.15 \cdot 10^{-2}$ |
| SW CRE | -69 | -2.9* | -26 | -0.46* |
| LW CRE | 18 | -0.04 | 21 | 0.35 |
| SW CRE (RK) | -69 | -3.4* | -26 | -0.46* |
| LW CRE (RK) | 18 | 0.20 | 21 | 0.26 |

**Table 7.** The same as Table 6 (impact of additional Arctic shipping) averaged over all latitudes north of $75°$ N.

| | Late summer (2050) | Late summer ($2050_{ship} - 2050$) | Early autumn (2050) | Early autumn ($2050_{ship} - 2050$) |
|---|---|---|---|---|
| Net SW radiation | 213 | -3.9* | 32 | -0.45* |
| Net LW radiation | -227 | -0.47 | -200 | -0.75 |
| $RF_{ari}$ | $41 \cdot 10^{-2}$ | $1.3 \cdot 10^{-2}$ | $11 \cdot 10^{-2}$ | $0.52 \cdot 10^{-2}$ |
| BC deposition | $19 \cdot 10^{-2}$ | $0.64 \cdot 10^{-2}$ | $2.5 \cdot 10^{-2}$ | $-0.02 \cdot 10^{-2}$ |
| SW CRE | -57 | -3.7* | -9.9 | -0.38* |
| LW CRE | 9.1 | -0.23 | 15 | 0.61 |
| SW CRE (RK) | -57 | -4.4* | -9.9 | -0.35* |
| LW CRE (RK) | 9.1 | 0.18 | 15 | 0.46 |

averaged between $75°$ and $90°$N). The LWP slightly decreases in their analysis (-1%; in-cloud); in contrast, it increases in our simulations (+17%; all-sky, averaged between $75°$ and $90°$N). While our simulated precipitation shows no clear trend, the results by Christensen et al. (2014) suggest that ship emissions delay precipitation by enhancing cloud lifetime. The different results could be explained by the location of the ship tracks analysed by Christensen et al. (2014): the majority of their samples lie between $45°$ S and $45°$ N, and only very few datapoints are from the Arctic. Precipitation formation at high latitudes differs considerably from that at low latitudes since e.g. convection is usually much more important at low latitudes.

While liquid clouds are significantly impacted by ships in our simulations, this is not the case for cloud ice, neither in late summer nor in early autumn. Theoretically, ship emissions could influence heterogeneous freezing in ECHAM6-HAM2 by several processes, for example:

- The increase in BC emissions could lead to enhanced immersion freezing by BC.

- The increased $SO_2$ emissions could shift some dust particles from the insoluble to the internally mixed mode, which shifts contact freezing to immersion freezing, i.e. to colder temperatures, as found by e.g. Hoose et al. (2008).

- Decreases in the droplet radius would decrease the contact freezing rate.

- Increases in the CDNC would increase the contact freezing rate.

The last two effects might partly cancel each other since a larger number concentration of CCN is expected to simultaneously decrease the droplet radius and increase the CDNC. However, also the first two points seem to be irrelevant as ship emissions have no significant impact on cloud ice in our simulations. To better understand why and gain some insights into the importance of the different heterogeneous freezing processes, we calculated the number of ice crystals that freeze via each of these processes (Fig. 12a, c, e). Immersion freezing by dust is the dominant freezing process in the Arctic in late summer (Fig. 12c). However, contact freezing by dust is more important near the surface since it can induce freezing at higher temperatures than immersion freezing (Fig. 12a). With additional ship emissions, the number of ice crystals formed by contact freezing decreases

near the surface and increases at higher altitudes (Fig. 12b). Since the relative changes in CDNC are larger than the relative changes in droplet radius (which would increase the contact freezing rate), we suspect that contact freezing near the surface is reduced by shifting more dust particles to the internally mixed modes. This is consistent with the slight (non-significant) increase in immersion freezing occurring near the surface (Fig. 12d).

Compared to dust, BC initiates freezing only in very few cloud droplets (Fig. 12e) because its influence is mainly restricted to high altitudes where temperatures are sufficiently low to initiate freezing. However, BC particles from ships are emitted near the surface. Therefore, the largest increases in BC concentrations also occur near the surface (Supplementary Fig. 12b). As a consequence, BC immersion freezing is slightly enhanced near the surface (Fig. 12f), but absolute changes are orders of magnitudes smaller than the decreases in contact freezing of dust. These findings lead to the conclusions that i) BC immersion

freezing is largely not affected because of the low altitude of ship emissions, ii) even if it were, it would hardly matter because dust is by far the dominant INP, and iii) $SO_2$ emissions from ships lead to a slight shift from contact to immersion freezing near the surface, thus rather leading to a non-significant decrease in cloud ice at low altitudes.

Heterogeneous freezing is still an active field of research, and contradictory evidence exists concerning the ability of combustion aerosols to act as INPs (Kanji et al., 2017). Laboratory results suggest that soot starts initiating freezing at temperatures

$\leq -30\,°C$ (Kanji et al., 2017, Fig. 1-7). On the other hand, Thomson et al. (2018) found an increase in INP concentrations in ship tracks at higher temperatures. The increases were small at temperatures around $-20°C$, moderate at $-25°C$ ($\approx +0.5\,L^{-1}$; saturation ratio of 1.22), and quite pronounced at $-30°C$ ($\approx +2\,L^{-1}$; saturation ratio of 1.32). The ship plumes were measured near the port of Gothenburg ($57.7°$ N, $11.8°$ E) in 2013 and 2014, and the meteorology in general represented climate conditions of the late-autumn maritime North. If ship exhaust (not necessarily the BC particles) can indeed induce freezing at higher

temperatures than in the laboratory-based BC-parameterisation used in our model, the impact on cloud ice could be larger than in our simulations, especially in early autumn when temperatures are colder.

### 3.2.3  Aerosol radiative forcings

The higher aerosol burdens due to ship emissions lead to enhanced AOTs (significant increase from $1.4 \cdot 10^{-2}$ to $2.0 \cdot 10^{-2}$ in late summer and insignificant increase from $1.4 \cdot 10^{-2}$ to $1.5 \cdot 10^{-2}$ in early autumn; averaged between $75°$ and $90°$N). Changes

induced by additional ship emissions are on the same order of magnitude as the changes caused by additional sea salt and DMS emissions from 2004 to 2050 ($\approx +0.2 \cdot 10^{-2}$). In contrast to the changes in aerosol absorption from 2004 to 2050 (no significant changes in late summer; decrease in early autumn), ship emissions lead to pronounced and significant increases in the aerosol absorption optical thickness (from $1.12 \cdot 10^{-3}$ to $1.19 \cdot 10^{-3}$ in late summer and from $0.83 \cdot 10^{-3}$ to $1.00 \cdot 10^{-3}$ in early autumn; averaged between $75°$ and $90°$N). This is not surprising since OC and predominantely BC are important absorbers of sunlight.

In late summer, the SW component clearly dominates changes in the net $RF_{ari}$ (e.g. $+13\,mW\,m^{-2}$ in SW compared to $+0.40\,mW\,m^{-2}$ in LW under all-sky conditions; averaged between $75°$ and $90°$N). Under clear-sky conditions, the ship emissions induce a pronounced cooling (i.e. $RF_{ari}$ decreases; see Fig. 13b). This cooling reverses to a non-significant warming under all-sky conditions (Fig. 13d). Again, this shows that the scattering of aerosol particles becomes less important when the scattering of clouds is considered as well, and that the aerosol absorption can be enhanced in the presence of clouds.

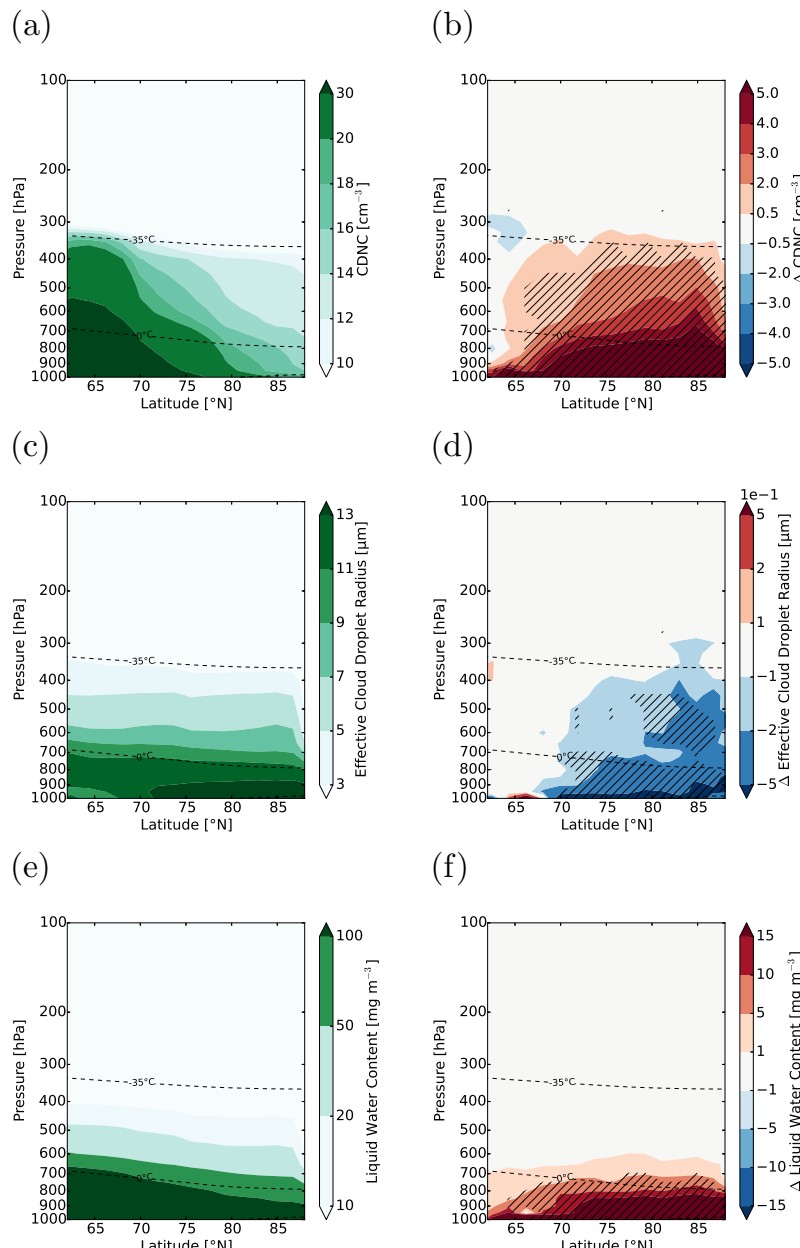

**Figure 11.** CDNC, effective cloud droplet radius, and LWC in late summer (Jul/Aug; in-cloud values): (a)/(c)/(e) show the absolute values for 2050 (reference), (b)/(d)/(f) the difference between a simulation with enhanced ship emissions and the reference simulation (difference between **arctic_2050_shipping** and **arctic_2050**). Hatched areas are significant at the 95% confidence level. The dashed lines show the $0°C$ and the $-35°C$ isotherms.

In early autumn, changes in the SW component still dominate changes in the $RF_{ari}$ in the region of shipping activity (e.g. $+8\,\mathrm{mW\,m^{-2}}$ in SW compared to $+3\mathrm{mW\,m^{-2}}$ in LW under all-sky conditions; averaged between $75°$ and $90°$N). Under clear-sky conditions, the ship emissions lead to locally significant decreases in $RF_{ari}$ (see Supplementary Fig. 14b). Under all-sky conditions, changes in net $RF_{ari}$ are not significant (Table 7).

In early autumn, the BC deposition on snow leads to a small but significant warming over part of the Arctic Ocean (see Fig. 14f). Although these changes are pronounced in relative terms, they are more than one order of magnitude lower in absolute terms compared to the enhanced cooling by clouds, which is discussed in the next section: averaged between $60°$ and $90°\,\mathrm{N}$, the radiative forcing of deposited BC insignificantly increases by $1.5 \cdot 10^{-3}\,\mathrm{W\,m^{-2}}$ in early autumn, while the SW CRE is significantly enhanced by $-2.9\,\mathrm{W\,m^{-2}}$ in late summer (Table 6).

Based on the future Arctic ship emissions by Corbett et al. (2010), Dalsøren et al. (2013) estimated how short-lived atmospheric pollutants might change by 2030. Meteorology, sea ice extent, and emissions not related to ships were not changed between 2004 and 2030 in their simulations. Therefore, we compare our simulated changes which are only due to shipping (change from **arctic_2050** to **arctic_2050_ship**) with their results. In their high emission scenario, BC and OC annual ship emissions increase in the Arctic by 2030 (BC by a factor of $\approx 5$ and OC by a factor of $\approx 2$), whereas $SO_2$ emissions slightly decrease (by $\approx 4\%$). In our simulations, annual Arctic BC, OC, and $SO_2$ ship emissions increase by factors of 11, 10, and 7, respectively. Averaged between $60°$ and $90°\,\mathrm{N}$ and over August, September, and October, Dalsøren et al. (2013) find that the radiative forcing of aerosols overall increases: $+5\,\mathrm{mW\,m^{-2}}$ for sulphate, $+5$ to $+6\,\mathrm{mW\,m^{-2}}$ for BC, and nearly no changes for OC. The sum of these values is larger than the value that we find ($+5.7\,\mathrm{mW\,m^{-2}}$ averaged over the same time period and area) although our increases in ship emissions are higher. It is possible that the radiative forcing of all aerosols is more positive in the study by Dalsøren et al. (2013) because of the different $SO_2$ emissions: in our simulations, the $SO_2$ emissions increase, which leads to cooling. In contrast, the $SO_2$ emissions in the study by Dalsøren et al. (2009) slightly decrease, which leads to a small positive forcing. Furthermore, the effect of clouds on $RF_{ari}$ might differ between the simulations by Dalsøren et al. (2009) and our simulations. The changes induced by deposited BC are $\approx 1\,\mathrm{mW\,m^{-2}}$ in both the study by Dalsøren et al. (2009) and in our simulations. While the increase in BC emissions is much larger in our simulations, less snow is available in 2050 compared to 2004.

### 3.2.4 Cloud radiative effects

In late summer, aerosol particles from ships lead to more but smaller cloud droplets and an enhanced LWC ($ERF_{aci}$), which increases the reflection of solar radiation. Thus, we see an enhanced cooling effect of clouds in most areas where the CDNC burden increases (Fig. 14b, d), i.e. the SW CRE becomes significantly more negative ($\approx -3.7\,\mathrm{W\,m^{-2}}$, averaged between $75°$ and $90°\,\mathrm{N}$). Changes in the LW CRE are smaller in terms of absolute amount, not consistently spatially correlated with ship emissions, and not significant (not shown). We additionally analysed the different contributions to the changes in CREs from cloud cover, cloud top altitude, and cloud thickness (see Fig. 15). The residuals in Fig. 15g and 15h show what can be attributed to neither cloud cover, nor cloud top altitude, nor cloud thickness; it should ideally be zero. While the changes in CRE caused by changes in cloud cover and cloud top altitude are not significant (Fig. 15a-d), the increase in cloud optical

thickness leads to significant decreases and increases in the SW and LW CRE, respectively (Fig. 15e, f). Averaged between $75°$ and $90°$ N, the increased optical thickness changes the SW CRE by $-4.6\,\mathrm{W\,m^{-2}}$ and LW CRE by $0.52\,\mathrm{W\,m^{-2}}$ in late summer (significant). When we partition the contributions from low and free-tropospheric clouds (defined as clouds with a cloud top altitude below/above the altitude of $680\,\mathrm{hPa}$), we find that 74% of the changes in SW cloud optical thickness occur in low clouds. This is not surprising considering that the ship emissions occur near the surface.

Possner et al. (2016) studied the influence of model resolution on ship-induced aerosol-cloud interactions and CREs (marine stratocumuli). They found that the changes in SW CRE were overestimated by a factor of 2.6 with the coarser model resolution ($\Delta x = 50\,\mathrm{km}$, $\Delta t = 180\,\mathrm{s}$) compared with the higher model resolution ($\Delta x = 1\,\mathrm{km}$, $\Delta t = 20\,\mathrm{s}$). In case this finding is generally applicable to numerical models, it could imply that the SW CRE is also overestimated in our simulations.

In the study by Dalsøren et al. (2013), aerosol-cloud interactions lead to much smaller changes in radiative forcing ($-2\,\mathrm{mW\,m^{-2}}$; averaged between $60°$ and $90°$ N and over August, September, and October) than in our simulations ($-0.85\,\mathrm{W\,m^{-2}}$; averaged over the same period and space). This is expected because our changes in future Arctic aerosol ship emissions are considerably larger than in Dalsøren et al. (2009). Furthermore, it should be noted that Dalsøren et al. (2009) calculate $\mathrm{RF_{aci}}$ using an empirical relationship that estimates CDNC from aerosol concentrations. In our case, CCN are calculated based on Köhler theory and we consider fast adjustments, i.e. report $\mathrm{ERF_{aci}}$ instead.

To summarise, ship emissions lead to a locally significant but very weak positive radiative forcing over the central Arctic Ocean in early autumn caused by absorption of deposited BC on snow. In contrast, the direct impact of aerosol particles on the net radiation ($\mathrm{RF_{ari}}$) is not significant. The changes in CREs are significant and show that aerosol particles enhance the cooling effect of clouds in late summer. When we partition CRE into its different components, we find no significant radiative changes induced by changing cloud top altitude or cloud cover, but the cloud optical thickness increases and is responsible for the significant net cooling. Since the cooling induced by aerosol-cloud interactions exceeds the warming of deposited BC by at least one order of magnitude, ship emissions of aerosols and their precursor gases overall induce a cooling in our simulations.

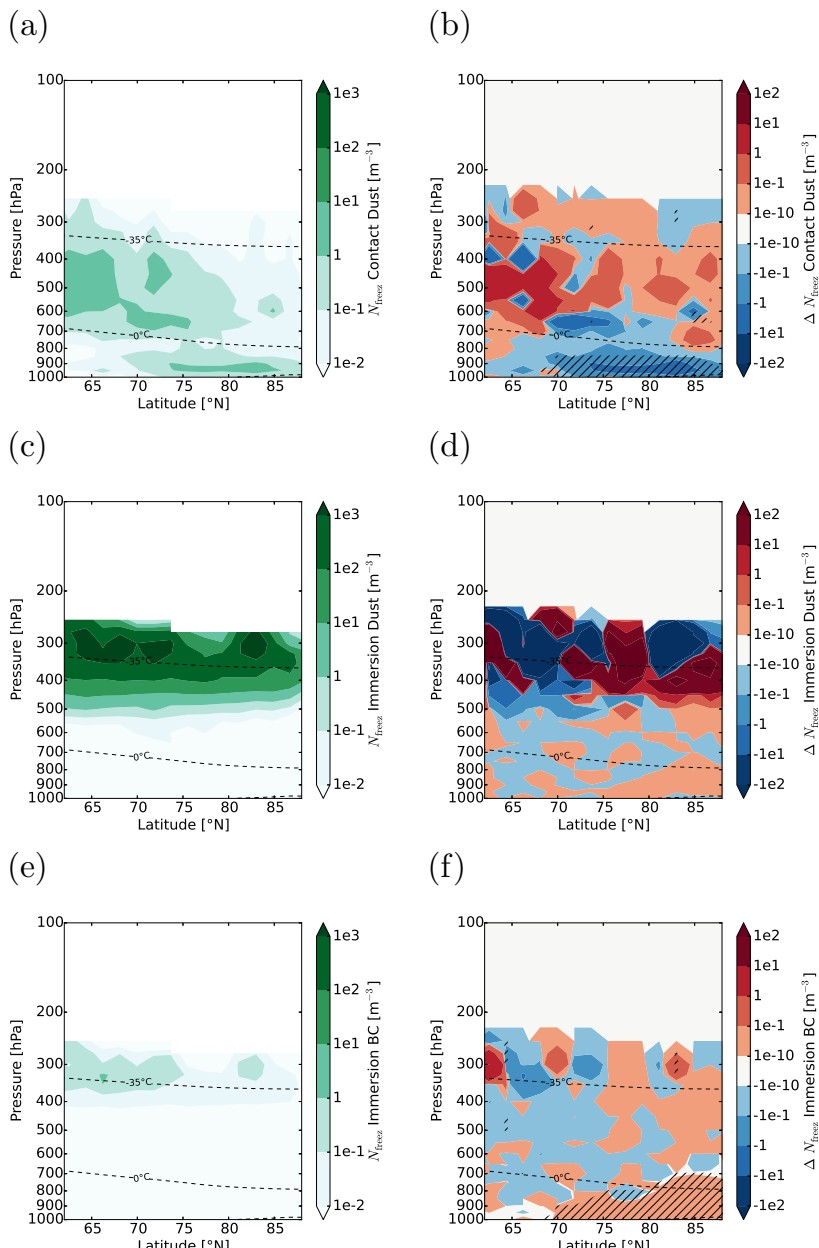

**Figure 12.** Number of cloud droplets that freeze heterogeneously per timestep ($N_{\text{freez}}$) in 2050: (a)/(b) contact freezing by dust, (c)/(d) immersion freezing by dust, (e)/(f) immersion freezing by black carbon in late summer (Jul/Aug). On the left side, absolute values for 2050 (reference) are shown. On the right side, the difference between a simulation with enhanced ship emissions and the reference simulation is displayed (difference between **arctic_2050_shipping** and **arctic_2050**). Note that the scale is logarithmic and that the lowest bin had to be decreased to $10^{-10}$ to display statistically significant increases in immersion freezing by BC. Hatched areas are significant at the 95% confidence level. The dashed lines show the $0°$C and the $-35°$C isotherms.

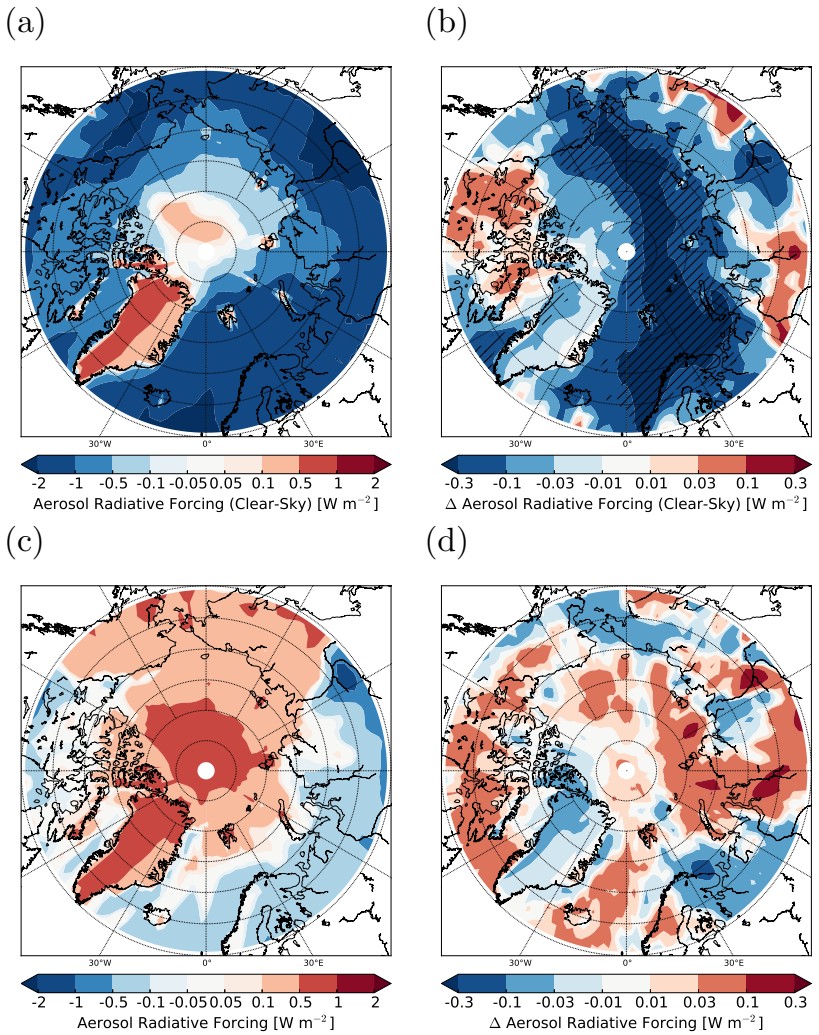

**Figure 13.** Aerosol radiative forcing in late summer (Jul/Aug) 2050: (a)/(b) under clear-sky and (c)/(d) under all-sky conditions. On the left side, absolute values for 2050 (reference) are shown. On the right side, the difference between a simulation with enhanced ship emissions and the reference simulation is displayed (difference between **arctic_2050_shipping** and **arctic_2050**). Hatched areas are significant at the 95% confidence level.

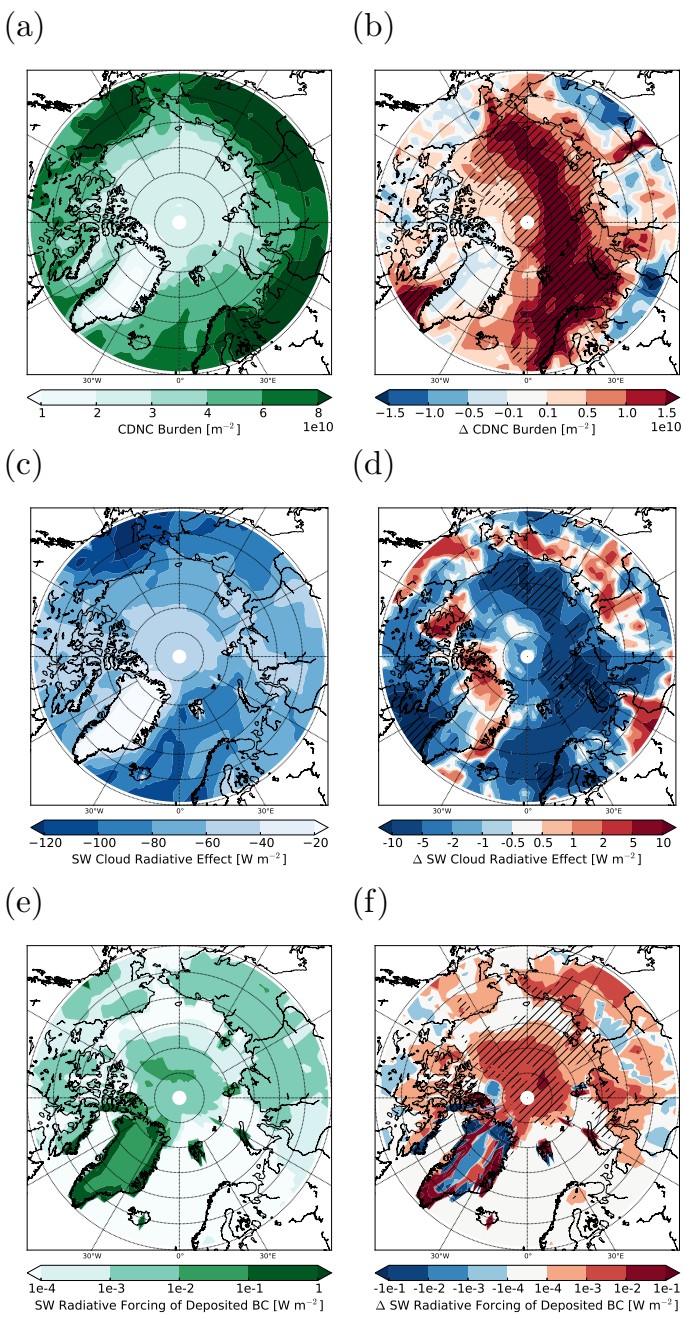

**Figure 14.** The impact of additional ship emissions in the Arctic on: (b) in-cloud CDNC burden, (d) SW CRE, and (f) radiative forcing due to BC deposition on snow. In (a), (c), and (e), the reference without additional ship emissions is shown (**arctic_2050**). Hatched areas are significant at the 95% confidence level. (a) to (d) are shown for late summer (Jul/Aug), (e) and (f) for early autumn (Sep/Oct). Note that the scale in e) and f) is logarithmic.

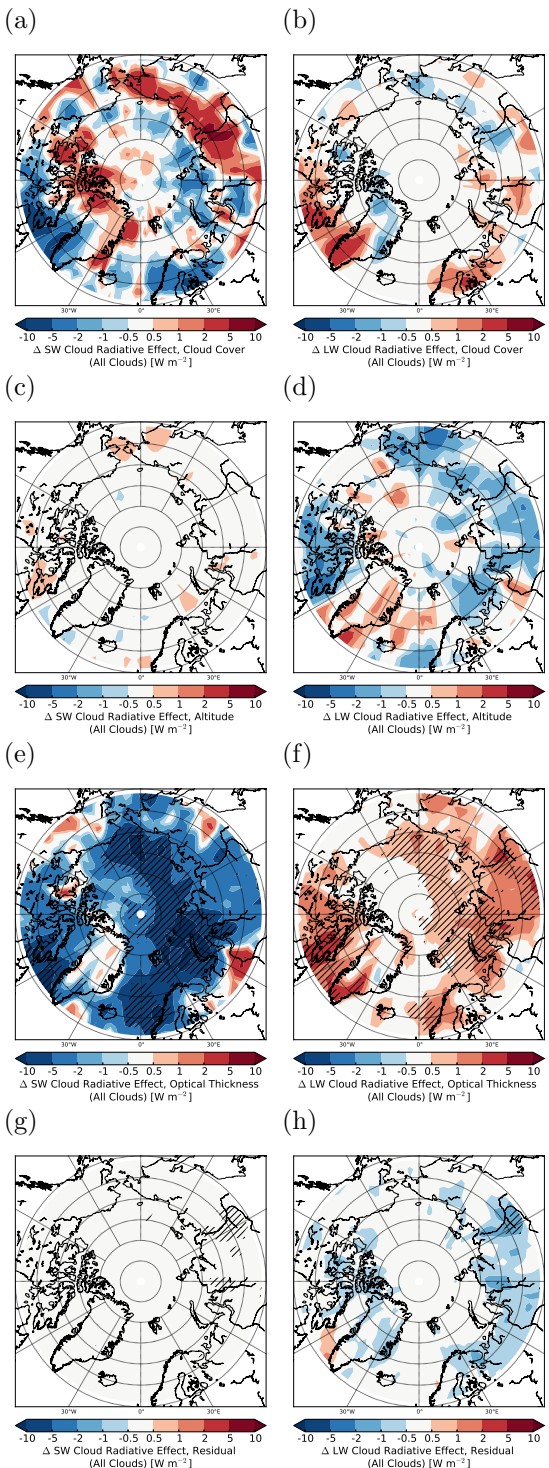

**Figure 15.** Different contributions to the changes in SW (left) and LW (right) CREs in late summer (Jul/Aug) caused by enhanced shipping: contribution from changes in (a)/(b) cloud cover, (c)/(d) cloud top altitude, and (e)/(f) cloud optical thickness. In (g)/(h), the residual is shown. Hatched areas are significant at the 95% confidence level.

## 4 Summary and conclusions

The main goal of this work was to analyse aerosol-cloud, aerosol-radiation, and cloud-radiation interactions in a warming Arctic when sea ice extent diminishes in late summer and early autumn. Simulations with ECHAM6-HAM2 were conducted for the years 2004 and 2050. We also estimated the impact of enhanced future Arctic shipping activity on climate.

Our results suggest that the future decrease in summer Arctic SIC will significantly increase sea salt and DMS burdens in the Arctic due to enhanced emissions. Both changes in aerosols and meteorology will lead to enhanced CDNCs. Furthermore, not only the number concentration but also the size of cloud droplets will generally increase because of higher specific humidities leading to thicker clouds. In late summer, the net CRE at the TOA will become more negative mainly because of the decrease in surface albedo associated with melting of sea ice. Also $RF_{ari}$ will decrease in late summer and early autumn mainly as a consequence of sea ice melting. The decrease in both net CRE and $RF_{ari}$ might delay Arctic warming to some extent.

The simulated LWP, cloud cover, CREs, and surface concentrations of BC and sulphate under present-day conditions compare well with Arctic observations. However, our model has a low bias in AOT and cloud ice, which could impact the simulated absolute changes in the radiative forcings and the CREs. Missing aerosol sources such as nitrate and SOA most likely contribute the simulated underestimation of AOT. In future work, nitrate as well as a state-of-the-art SOA scheme will therefore be incorporated in our model. Furthermore, intermodel differences in sea salt emissions are large (de Leeuw et al., 2011), and so are the differences between our results and other modelling studies that investigated changes in natural aerosols with declining sea ice. This highlights that the results from this study – as from any climate model study projecting the future – are uncertain.

Arctic ship emissions related to transport and oil/gas extraction have a negligible impact on clouds and radiation in our simulations. Only when we increase the ship emissions of Peters et al. (2011) by a factor of ten is the signal-to-noise ratio sufficiently large to detect ship-induced changes. Considering that our model probably underestimates the background aerosol concentrations in the Arctic, the simulated impact of the (tenfold) ship emissions could be overestimated. With tenfold ship emissions, the AOT significantly increases by the same order of magnitude as natural AOT changes from 2004 to 2050. $RF_{ari}$ shows only minor, insignificant changes in the presence of clouds, though. An increase in BC deposition on snow leads to a very small local warming in early autumn. Meanwhile, $ERF_{aci}$ induces a cooling in late summer. The magnitude of changes in $ERF_{aci}$ are considerably larger than those induced by the deposition of BC on snow, implying that ship emissions might overall induce a cooling. In our simulations, only liquid clouds show significant changes with increased ship emissions, while cloud ice is unaffected. Considering the large uncertainty of heterogeneous freezing processes, this result needs to be regarded with caution.

Compared to other changes (such as the decrease in surface albedo or the increase in natural aerosol emissions), ship emissions of aerosols and their precursor gases seem to have a small effect on climate considering that we scaled the emissions up by one order of magnitude. However, even though this study suggests that Arctic ship emissions of aerosols and their precursor gases might have a negligible or slightly beneficial impact on climate, they will also increase air pollution and might disturb local flora and fauna. Furthermore, this study does not account for ship-induced changes in greenhouse gases (e.g. $O_3$),

which are also important forcers (Dalsøren et al., 2013; AMAP Assessment, 2015). More studies are required to confirm or refute the findings of this work as well as to explore further ship-related environmental impacts.

## 5 Code availability

The code is available upon request.

5 ## 6 Data availability

The data is available upon request.

**Appendix A: Map of Arctic Seas**

As a help for readers not familiar with the Arctic Ocean, Fig. A1 shows its most important regional seas. Furthermore, some land masses are labelled for better orientation.

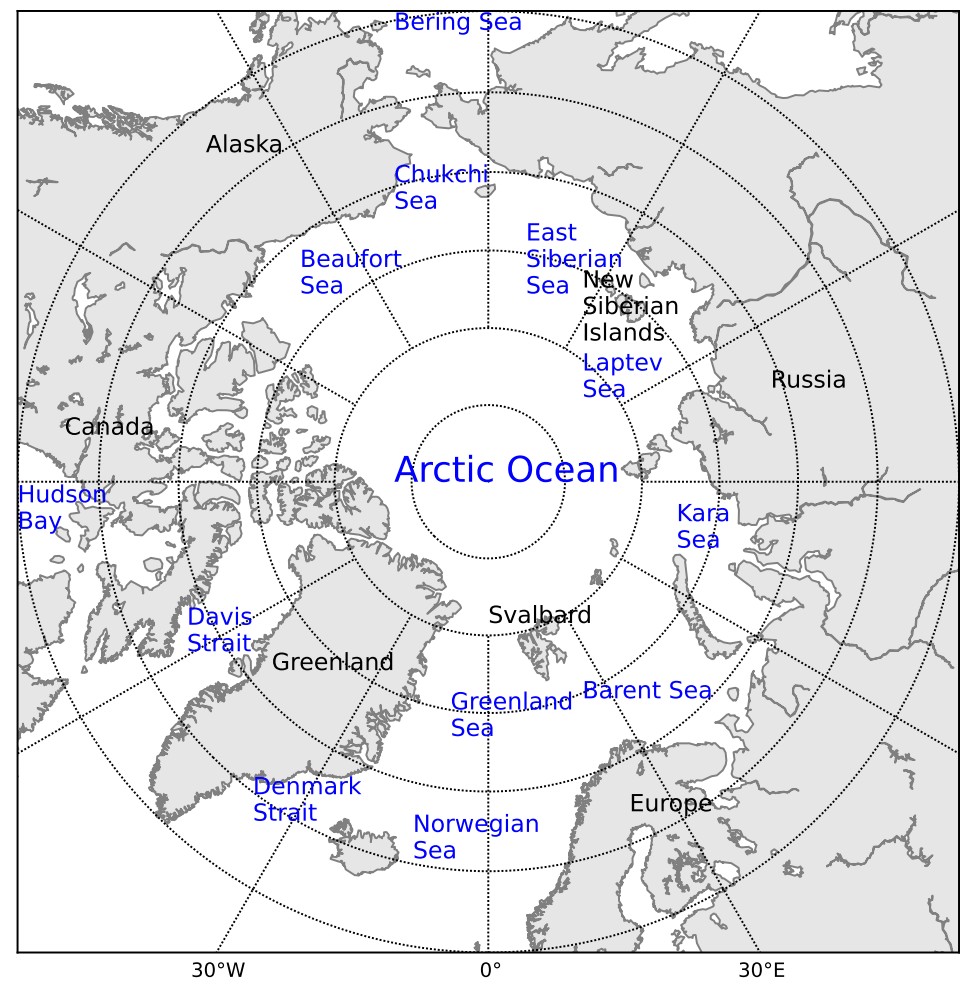

**Figure A1.** The Arctic Ocean and regional seas are labelled in blue, land masses in black.

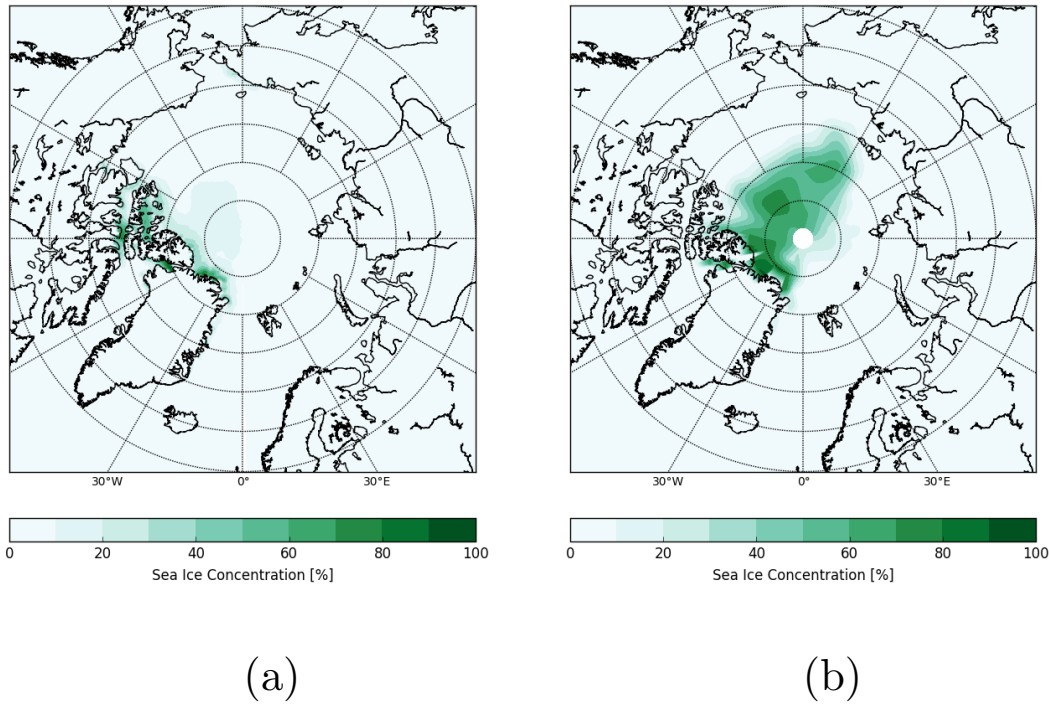

Figure A2. SIC in 2050 for (a) NCAR-CCSM3 in September (average over 5 ensemble members) and (b) MPI-ESM in October (the ensemble member used in this study).

## Appendix B: Comparison of sea ice between MPI-ESM and NCAR-CCSM3

Here we compare the sea ice used as input for the study of Peters et al. (2011) with our prescribed sea ice from MPI-ESM. With that we want to ensure that the ship emissions – which explicitly depend on the sea ice thickness and concentration – are compatible with the sea ice used in our study. Peters et al. (2011) used a 5-year running average of the NCAR-CCSM3 model to
5  calculate future sea ice conditions (scenario A2). Instead of averaging over years, we calculated the mean over the 5 ensemble members of NCAR-CCSM3 from CMIP3 for our comparison, which should give similar results. For their calculations, Peters et al. (2011) chose the months March, June, September, and December to represent each season. In our model, we prescribe the sea ice monthly because this is more realistic. Therefore, we will compare the sea ice in July from MPI-ESM with the sea ice in June from NCAR-CCSM3 (used in the calculation for ship emissions in July) and the sea ice from August to October
10  from MPI-ESM with the sea ice in September from NCAR-CCSM3. For this comparison, we will focus on the regions where most Arctic ship emissions are projected to occur in the future.

The sea ice thickness is generally thinner in MPI-ESM than in NCAR-CCSM3. The opposite is the case for the sea ice extent, which is larger in MPI-ESM than in NCAR-CCSM3. In August and October, the SIC in MPI-ESM is higher than the

NCAR-CCSM3 September value (used by Peters et al. (2011) for August, September, and October). At the locations of the ship tracks, differences are most pronounced north of the New Siberian Islands, where the SIC reaches up to 60-70% in MPI-ESM, whereas basically no sea ice is left in NCAR-CCSM3 (see Fig. A2). However, with an extended use of ice breakers, ships can pass despite the higher SIC. Furthermore, the area where this larger SIC for MPI-ESM occurs is rather small, as the SIC in MPI-ESM rapidly decreases towards the New Siberian Islands and the Russian coast. By slightly changing the shipping routes, most of the additional expenses linked to SIC (i.e. to ice breakers, which are included in the cost-benefit analysis of Peters et al., 2011) would be saved. We therefore expect that costs associated with breaking and/or bypassing sea ice are small and should not considerably change the ship emissions derived by Peters et al. (2011).

## Appendix C: Significance test for cloud feedback

The cloud feedback is calculated using radiative kernels. These kernels are calculated as *differences* of two simulations, here represented by the vectors $\boldsymbol{a} = \begin{pmatrix} a_1 \\ a_2 \\ a_3 \\ ... \\ a_n \end{pmatrix}$ and $\boldsymbol{b} = \begin{pmatrix} b_1 \\ b_2 \\ b_3 \\ ... \\ b_n \end{pmatrix}$, where $n$ is the number of samples. In our case, we could not simply use a one sample t-test upon the differences $\boldsymbol{a} - \boldsymbol{b}$ because the differences are calculated from 20 *independent* samples (i.e. years) with different standard deviations for the different simulations. Instead, we reconstructed from the following differences the standard deviation of $\boldsymbol{a}$, the standard deviation of $\boldsymbol{b}$, and the difference between the means of $\boldsymbol{b}$ and $\boldsymbol{a}$:

– For standard deviation of $\boldsymbol{a}$: calculate standard devation of $\begin{pmatrix} b_1 - a_1 \\ b_1 - a_2 \\ ... \\ b_1 - a_n \end{pmatrix}$.

– For standard deviation of $\boldsymbol{b}$: calculate standard devation of $\begin{pmatrix} a_1 - b_1 \\ a_1 - b_2 \\ ... \\ a_1 - b_n \end{pmatrix}$.

– For difference between the means of $\boldsymbol{b}$ and $\boldsymbol{a}$: $\frac{b_1+b_2+...+b_n}{n} - \frac{(a_1+a_2+...+a_n)}{n} = \frac{b_1-a_1+b_2-a_2+...+b_n-a_n}{n}$, i.e. we calculated the kernels between $b_1$ and $a_1$, ..., $b_n$ and $a_n$ and calculated the average of these differences.

With this information, we could calculate the p-values using the Welch's test for each gridpoint and control the FDR as described in Sect. 2.4.

*Competing interests.* The authors confirm that they have no conflict of interest.

*Acknowledgements.* The research leading to these results has received funding from the European Union's Seventh Framework Programme (FP7/2007-2013) project BACCHUS under grant agreement no. 603445. This work was also supported by a grant from the Swiss National Supercomputing Centre (CSCS) under project ID s652. We are very grateful to Stig B. Dalsøren and Glenn P. Peters, who kindly gave access to their ship emission inventories. Furthermore, we thank Sylvaine Ferrachat, who set up the simulations for the (unpublished) precursor study of this paper. We acknowledge the international modeling groups for providing their data for analysis, and the Program for Climate Model Diagnosis and Intercomparison (PCMDI) for collecting and archiving the model data. In this context, we also thank Jan Sedlacek for his effort to process the CMIP5 data set. The ECHAM-HAMMOZ model is developed by a consortium composed of ETH Zürich, Max Planck Institut für Meteorologie, Forschungszentrum Jülich, University of Oxford, the Finnish Meteorological Institute, and the Leibniz Institute for Tropospheric Research, and managed by the Center for Climate Systems Modeling (C2SM) at ETH Zürich. Concerning the model development, special thanks go to the "Fire in the Earth System" Group of Silvia Kloster (part of the Emmy Noether Junior Research Group; MPI) for implementing the BC deposition over land in ECHAM. We are also grateful to Ina Tegen and Stefan Barthel, who implemented the used sea salt parameterisation into ECHAM-HAM. Furthermore, we acknowledge the SHEBA data provided by NCAR/EOL under the sponsorship of the National Science Foundation (https://data.eol.ucar.edu/). The Clouds and the Earth's Radiant Energy System (CERES) Energy Balanced and Filled (EBAF) Top-of-Atmosphere (TOA) data was obtained from the NASA Langley Research Center CERES ordering tool at http://ceres.larc.nasa.gov/. Last but not least, we thank the three anonymous referees for their valuable comments.

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
