# Peer review of "How important are future marine and shipping aerosol emissions in a warming Arctic summer and autumn?"

_Atmospheric Chemistry and Physics, 2017_

## Referee Comment (RC1) · Anonymous Referee #1 · 6 Dec 2017

General comments: The topic of the paper is timely and of interest to both the research community, to policy makers and to the general public as we move towards sea ice free Arctic summers/autumns. The methods applied hold high scientific quality. However, the overall impression of the work is that it is a bit unfinished and rough. The discussions lack precision and results are presented in a somewhat confusing manner. The research quality is good, but modifications to how it is presented is needed before publication.

Specific comments:

- In general, a lot of the discussion concerning specific results does not include the

numbers in question and not references to where to find these numbers in the text (if present). This makes the discussion only qualitative. This is a shame, when the numbers are clearly available from the model results. Also, the reader ends up flipping back and forth looking for the numbers to back the discussion. Below, I will list some places in the text where this should be addressed.

- The reader lacks some of the basic information about the set up and the control simulation to be able to understand the results. There should, for instance, be a plot of the sea ice concentration at annual minimum or averaged over each season available for both periods, at least in the supplementary.

- A lot of the changes that occur between 2004 and 2050 are discussed, but not shown. This goes for example for temperature and precipitation. Make sure to label when the results you are referring to are not shown (see comments below) and please consider to show more of the changes that you use in your explanations, at least in the supplementary.

- For some figures, you average from 70-90N, for the tables you use either 60-90N or 75-90N. It would be more consistent if your figures and tables matched and one could follow the impacts of interest from figure to tables etc. Please consider changing this.

- Parts of the text is very oral and parts are too elaborate. Below I make both comments on things that should be changed and comments about how the text itself can be improved.

o P2, L17: Remove "where some (. . .) are labelled"

o P2,L20: Replace "until" with "before"

o P2, L22: Remove "(cruise ships)"

o P2, L24: "Nowadays" is very oral. Please rewrite. Also include " (. . .) pristine compared to other regions (. . .)" The use of depleted here make it sound like the aerosols have been removed. Please rewrite.

o P2, L28: Remove "and can (. . .) (Vali, 1985)" It is of no relevance here and only distracts the reader.

o P2, L32: Make part of the previous paragraph.

o P2, L32-P3,L8: May be a bit hard to follow because the indirect effects are described before the general radiative effects of clouds. Consider a rewrite to change the order.

o P3, L1: Perhaps mention why smaller droplets increase the cooling effect of clouds?

o P3, L18: "Re-emission of SW"? Please rewrite!

o P3, L23-27: Suggest to use the same terms for SW and LW description. For LW, the emissivity includes the water path and the temperature is height dependent. I suggest to make the definitions a bit more tidy.

o P3, L31: Suggest new paragraph before "How Arctic clouds (. . .)"

o P4, L11: a bit confusing to have figure references in a listing of the main goals of the paper. Suggest to move this.

o Figure 1: A bit confusing. Why not use red for increase and blue for decrease?

o P4, L22: Suggest to remove: "HAM2 (. . .) modes." and move "To link (. . .) implemented (. . .)" to the end of the next paragraph, after "(. . .) sedimentation".

o P5, L12: Does lowering the CDNC threshold affect the global radiative balance?

o P6, L13: Suggested rewrite: "(..), we used an inventory described in (. . .)"

o P6,L19: Remove "more equations can be found therein"

o P8,L10: Define the abbreviations COADS and AMVER.

o P8,L15: Remove "in addition". It is a bit confusing

o P8, L19: Please replace the word "exploit"

o P8, L21: Consider removing the sentence "We processed (. . .)". Too detailed information in my opinion.

o P9, L10: Suggest to remove "(non- (. . .) control)".

o P9, paragraph starting on line 18: I find this very confusing and suggest a clarification of what you mean in this paragraph. Do not see the link between these two sentences. The reasoning therefore fails.

o P9, L25: Please rewrite to "(. . .) can be considered as a realistic (. . .)"

o P9 General comment to the justification of a tenfold increase of the ship emissions. I suggest a rewrite of this discussion. You increase emissions so as to see a signal and try to justify it afterwards, while at the very end of the paragraph state that the emissions are now "probably too high". Increasing your emissions to get a signal is fine. To discuss that emission estimates may be too low is fine. Your emissions may well be an upper estimate. However, the discussion is long, a bit vague and a bit on the defensive side and makes the reader question whether the authors question their own reasoning here.

o P9, L30: Please define your abbreviations and give references to the models used. Did you use RCp8.5 for future simulations here? Please specify.

o P10, L9: The results would have appeared more robust if the sic and SSTs used were an average taken over eg a ten year time period centered at 2004 and 2050. Using one year (2003) and one ensemble (2050) to test the robustness of your choice of sic and sst is too week.

o P11, L3: suggest to remove "(eg. SW radiation, temperature)". It is not necessary.

o P11,L3: Rewrite to "change considerably"

o P11,L9: Rewrite to "deviate considerably"

o P11,L10: reference needed after statement "Furthermore, most models (. . .) prevalent in the Arctic".

o P11, L17: Rewrite "in the vicinity". Suggest "Over the arctic ocean"

o P11,L17: When you say "significantly" here is there a statistically significant change? Please specify.

o P11,L18: How much does the SIC decrease? Please specify.

o P12,L3: "(. . .) modified (. . .)" What does modified mean here? Consider removing.

o P13,L3: Please insert the "(. . .) increases in the future (. . .)"

o P13,L5: Is it only the change in CCN concentration that affects CDNC? Not moisture availability? At what supersaturation do you calculate your CCN concentration? Does the average supersaturation change between the runs?

o P13,L8: "averaging over cloudy and non-cloudy conditions". This is a bit confusing. Are you writing about allsky conditions? Also, please replace increases with increased.

o P13, L22: Please explain why the increase in ICNC near the surface is due to the increase in CDNC.

o P13,L25: insert (not shown) after (near Svalbard).

o P16,L2: ((. . .) except over the Arctic Ocean (. . .)" This is over the sea ice?! This should most definitely be specified.

o P16,L6: "since the clouds (. . .), more SW radiation can be absorbed (. . .)". Consider rewriting this for clarity.

o P16,L10: cooling effect vs warming effect. Please specify the actual numbers here.

o P16;L13: What does the optical thickness change from and to? If you do not give the numbers you need to specify that it is not shown.

o P16,L14: Consider replacing disentangle by "distinguish between"

[Figure]

o P17,24: Please insert "(not shown)" after "temperature".

o Table 2: Please include the values from the control run to get the relative importance.

o P22,L7-8: Please insert "(not shown)" after "late summer" and after "significant for OC".

o P22,L12: Please remove "using the hypsometric equation". Redundant.

o P22,14: Please move your reference to figure 9b to directly after "800 hpa" to avoid confusion.

o P22, section 3.2.1 general comment: Your hypothesis is very likely, but can you verify in your model that this is the case? Perhaps you could perform some sensitivity test? Right now this section is not very strong as it just lists model results without any proper discussion. It would also be good to include a vertical cross section of the aerosol change. This would be beneficial for the next section as well when discussing freezing.

o P22,L29: How much does the Liquid water path /mass increase?

o P23,L2: Please include the numbers you are describing in the text.

o P25,L21: How much does the optical thickness change. You should strengthen your discussion by describing the actual model results.

o P25,L27: "(. . .) under clear-sky conditions." Insert "(not shown)" here.

o P25,L32: The significant areas are not large and looking at the figures it looks like the numbers you are giving here are averaged over the whole region. If so, please make it clear that this number is not only including significant changes.

o P25,L32: The figure reference should be to figure 13(f)?

o P27,L4: please consider changing the numbers here to (-2 to -20 Wm-2)

o P27,L5: please insert "(not shown)" after "correlated with ship emissions".

o P27,L8: "While the CRE (. . .)" is it not the change in CRE? Same goes for line 7 and the figure text to figure 14.

o P27,L9: The change in optical thickness is significant in very small regions. This should be mentioned. Also when the average numbers are given in the following sentence, make clear that these numbers are not significant.

o P27, L14: Are these numbers significant?

o P27,L16: Please insert "very": "(. . .),ship emissions lead to a significant, but very weak (. . .)"

o P27,L20: Suggest to add "in limited regions" or something similar after "(. . .) and lead to significant net cooling"

o P32,L16: Insert "a very small" in front of "local warming".

Technical corrections: - Polarstereographic maps: Please insert a few more latitude lines, perhaps at the boarders for averages that you use: 70N and 75N.

- All figures: Move (a), (b) etc above figures.

- Figures using a blue to red color scale: The lightest colors are impossible to separate in printed figures. You need to improve this color scale. At the same time, consider to use a white color surrounding zero so that values at this separation (zero values) do not come out in color.

- Figure 7: Suggest to use different color scales for positive and negative values in (a) and (d).

---

## Referee Comment (RC2) · Anonymous Referee #2 · 28 Dec 2017

This paper investigates the impacts of changes in natural and anthropogenic aerosol emissions in a warmer future climate with less Arctic sea ice, delineating the contributions from aerosol radiative forcing, aerosol-cloud interactions, and cloud radiative effects. The topic is timely and results are interesting, both for the broader scientific community, as well as from a mitigation perspective. The quality is generally good, but some improvements are needed before publication. See comments below.

Comments:

On several occasions, results are presented and discussed without actual numbers, sometimes also without showing figures. Adding values would improve the quality and

readability. Furthermore, more information about the baseline state of the conditions in the model would be helpful. See also specific comments below.

Several different spatial and temporal regions are used (e.g., JJA/August, early fall, Sept/Oct, 70-90N, 75-90N, 60-90N). This makes it a bit hard to follow and the authors should consider if it is possible to be more consistent.

I'm missing a bit of broader context; e.g., do the results from this study point to future research needs, how do the impact of shipping emissions from the present study compare with previous work, what are the most important caveats/limitations.

Abstract: consider adding some quantitative information.

Page 1: cite Melia et al. 2016? (Melia, N., K. Haines, and E. Hawkins (2016), Sea ice decline and 21st century trans-Arctic shipping routes, Geophys. Res. Lett., 43, 9720–9728, doi:10.1002/2016GL069315.)

Page 3, line 18: "re-emission" – suggest rephrasing.

Page 4, line 8-15: I think the motivation and objective for this study needs a couple of additional lines, e.g., to summarize what the bulk of the literature described above show about the importance of combining all the processes and what is new/unique about the present study.

Page 4, line 11: I don't see that this goal is sufficiently addressed in the paper. The model is run with fixed SSTs and no quantification of temperature responses. As far as I can tell, Fig.1 shows arrows only from temperature changes to radiative changes. If you want to maintain this as a main objective, you need to come back to it later in the manuscript in a better way. However, I think that disentangling the aerosol-radiation-cloud interactions is a sufficient objective in itself.

Fig. 1: I like the figure, but find the colors a bit confusing. E.g., use of blue from less sea ice to more aerosols. Perhaps use red for increases and blue for decreases? Or add colors.

Page 5, line 8: "simplistic treatment": please specify.

Page 5, line 12: changing CDNC - does that affect the global radiative balance?

Page 5, line 12: is this based on observational constraints?

Section 2.1.2: to make the methods section easier to follow, I recommend combining all description of emissions into one paragraph. This will also reduce the need to refer to following paragraphs, which makes this section a bit hard to follow. Furthermore, are marine organic aerosol emissions included?

Page 6, line 18: all BC particles? Only hydrophobic? And only ship, or also other anthropogenic particles? Please specify

Section 2.2: Arctic_2050 vs Arctic_2050_shipping: the difference is a bit unclear. Does the former have Peters et al. 2050 ship emissions, but without the x10? Shipping emission factors are described as being lower due to regulations, which is why I wonder. If so, comparing these two does not give the total effect of changes in ship emissions, but the effect the x10 increase? Please clarify.

Table 1: would be useful to add references for the emissions as well.

Page 8, lines 20-22: I think these two sentences are excessively detailed.

Page 9, line 26: first you justify the increase, then you say it is probably too high? Consider revising for clarity.

Page 9, line 27: is it possible to add a reference?

Page 10, line 14: "naïve stipling approach" is not good language. Does this refer to a standard student's t-test? Please clarify/change.

Page 11, line 1: figures show, not "will show". Consider changing the language.

Page 11, line 17: consider providing numbers or showing results in a supplementary material.

[Figure]

Page 11, line 18: perhaps I misunderstand the language, but isn't the increase in wind speed following the reduced SIC the main reason for the increased DMS and sea salt emissions, and hence for the burden increase? Or are there other mechanisms, related to e.g., scavenging due to lower SIC that dominate the burden change? Please clarify.

Page 11, line 26: caused by what?

Page 11, line 32: JJA/August – do you consider a different period here? Please clarify.

Page 12, line 1: at some point it would be good to show/describe in detail the changes in variables such as SIC between 2004 and 2050. Could be added in a supplementary material.

Page 12, line 4: absolute emissions in 2004 or absolute emission changes? Please clarify.

Section 3.1.2: are the same general features seen during summer?

Page 16, line 13: again, it would be helpful to have the actual numbers.

Page 17, line 4: perhaps instead say "a strengthening of the direct aerosol effect" since it is in fact much stronger in 2100?

Page 17, line 5-7: I'm not convinced it makes sense to compare these numbers since the foundation and model experiments are so different. Unless you're able to disentangle effects of experimental differences in more detail, I don't see that this section add much information of value and it could be left out.

Figure 5: very hard to distinguish statistically significant areas.

Page 17, line 8: please add numbers or relative change.

Page 17, line 9: if I follow correctly, these results are still without any changes in anthropogenic aerosol emissions, so a small effect due to changes in BC deposition is to be expected, unless there are large changes in the scavenging. Could be useful to

remind the readers of this. In fact, even under RCP8.5, anthropogenic aerosol emissions decline strongly through the century, which could perhaps reduce this forcing altogether.

Page 22, line 7: A comparison with previous work using the Peters et al. inventory (without the x10 enhancement) could be useful, e.g., Ødemark et al. 2012; Dalsøren et al. 2013.

Dalsøren, S. B., Samset, B. H., Myhre, G., Corbett, J. J., Minjares, R., Lack, D., and Fuglestvedt, J. S.: Environmental impacts of shipping in 2030 with a particular focus on the Arctic region, Atmos. Chem. Phys., 13, 1941-1955, https://doi.org/10.5194/acp-13-1941-2013, 2013. Ødemark, K., Dalsøren, S. B., Samset, B. H., Berntsen, T. K., Fuglestvedt, J. S., and Myhre, G.: Short-lived climate forcers from current shipping and petroleum activities in the Arctic, Atmos. Chem. Phys., 12, 1979-1993, https://doi.org/10.5194/acp-12-1979-2012, 2012.

Page 22, line 8: the maximum changes occur at the same location as the emissions; however, there are statistically significant increases over much larger areas. Should be specified.

Page 22, line 11-17: are these shifts large enough to have notable implications, e.g., for forcing? Possible to discuss to add some context?

Page 25, line 18-19: actual magnitudes would be useful.

Section 3.2.3: this section is missing a discussion of and connection to studies of the radiative forcing of shipping, both in the Arctic and overall to global impacts. This is important given that main conclusion of the study concern the negligible impact of shipping aerosol emissions. In particular, a discussion of the impact of shipping found in studies that do include explicit treatment of aerosol-cloud interactions and/or offline radiative transfer calculations could be important.

Page 32, line 20-25: be careful about the phrasing of this conclusion, as it does not

cover other effects of shipping emissions, such as NOx-induced ozone changes and CO2.

---

## Referee Comment (RC3) · Anonymous Referee #3 · 30 Dec 2017

Gilgen et al present a set of sensitivity studies with the atmospheric GCM ECHAM-HAM. The control simulation is driven by conditions approximately representative for present-day, three sets of differing boundary conditions are then computed: (a) increased sea surface temperatures and decreased sea ice cover, (b), in addition, changed aerosol emissions, and (c) in addition, further ship emissions. Each of the simulations is run for a short period of ten years. A large set of results is presented.

The study is to a large extent based solely on the results of the one model and thus the hypotheses developed are strongly dependent on the chosen parameterisations. Very little comparison to data (for the control simulation) is presented. In one paragraph, the

cloud radiative effects are compared to SHEBA data – from this it seems that the model has a very large bias. I believe it would be necessary for an improved paper to at least show some evidence that the model performs satisfactorily in the Arctic in comparisons to observations, before the results from the sensitivity studies can be considered meaningful. At two instances, the results are compared to previously-published results for similar scenarios. It is astonishing how different the results are. A key hypothesis is that sea salt emissions may substantially increase with decreasing sea ice coverage. Fundamentally, this is no surprise, so the question is how large this could be quantitatively. Unfortunately the two other model studies reported are much more different from the model presented here than the change due to sea ice retreat (one model has a factor of 3 more, the other, a factor of 1000 less emission flux in present-day conditions). Also the radiative forcing due to aerosol-radiation interactions is very different between models – the model presented here has a substantially positive forcing, the other model, a negative one. Since such results are easily available from multi-model ensembles (CMIP5 or AEROCOM), it would be easy to put the model the authors use into context, much beyond the two studies cited.

When it comes to the interpretation of the results, much is left for speculation. If the authors choose to have a pure modelling study, why don't they at least precisely clarify the processes that change? Why not budgets for changes of CCN, INP? A table that lists all relevant numbers (e.g. for the entire region, and split for open ocean and sea ice surfaces) as simulated for the different scenarios would be useful (emission fluxes, CCN, INP, cloud particle concentrations, LWP, IWP).

Specific comments:

p3 l30 – this is not "generally" true, e.g. not in summer (as the following sentence correctly acknowledges)

p3 l32 – but it is likely a small effect (Pithan and Mauritsen). What is the reference for the following sentence ("generally...")?

[Figure]

P5 l12 – it would be good to report already here whether this threshold is hit, and, if so, how often. It would further be good to analyse whether indeed the lack of nitrate or organics is a major problem of this model for the Arctic.

P6 l1 – it would be good to comment on the results of Eckhardt et al. (ACP 2015)

p8 l6 – 10 years seem very little for small forcings

p12 top paragraph – what do these discrepancies by a factor of about 3000 imply for the fidelity of the results in terms of sea salt emission changes?

P13 l8 increased p13 l22 – i.e. homogeneous freezing of droplets? P13 l23 – at constant ICNC? L13 l27 – indeed surface fluxes? Or rather simply moist adiabat changes?

P16 l2 – it would be important to clarify whether this section refers to the radiative forcing by aerosol-radiation interactions only, or to the effective radiative forcing due to aerosol-radiation interactions, or whether it includes aerosol-cloud interactions. P16 l4 – what are the absorbing components, and why is the positive forcing so large? P16 l6 – it would be useful to demonstrate this at least in the supplementary material (since the authors write "not shown" it seems hey have the analysis at hand)

p17 l5 – how is the coincidence of approximately the same reduction by 0.2 Wm-2 explained? Is the same thing happening in both models?

P18 l13 – i.e. the effect is twice as large as observed? The authors should report this analysis as a table or similar.

---

## Author Comment (AC1) · 11 May 2018

**Response to Referee 1**

We thank the referee very much for the many relevant comments. They helped to improve the paper considerably.

Note that we conducted new simulations because the ship emissions were shifted by two weeks in the old simulations. This especially affected the impact of transit ship emissions in the Arctic in late summer 2050. Furthermore, we increased the number of simulated years from 10 to 20 for better statistics, as suggested by referee 3. In some cases, the results have thus changed; as an example, the SW CRE now increases significantly with additional tenfold Arctic ship emissions in 2050.

Referee's comments in blue, our replies in *grey and italic*.

- In general, a lot of the discussion concerning specific results does not include the numbers in question and not references to where to find these numbers in the text (if present). This makes the discussion only qualitative. This is a shame, when the numbers are clearly available from the model results. Also, the reader ends up flipping back and forth looking for the numbers to back the discussion. Below, I will list some places in the text where this should be addressed.
*Thank you very much for this comment. In the new version, we include now more quantitative information, namely numbers in brackets, additional Supplementary figures (Supplementary Figs. 4-14), and Tables 2 and 3.*

- The reader lacks some of the basic information about the set up and the control simulation to be able to understand the results. There should, for instance, be a plot of the sea ice concentration at annual minimum or averaged over each season available for both periods, at least in the supplementary.
*We include now plots averaged over each season for 2004 and 2050 in the paper (Fig. 2).*

- A lot of the changes that occur between 2004 and 2050 are discussed, but not shown. This goes for example for temperature and precipitation. Make sure to label when the results you are referring to are not shown (see comments below) and please consider to show more of the changes that you use in your explanations, at least in the supplementary.
*We included additional figures in the Supplementary Material (Supplementary Figs. 4-14) and now mention when something is not shown.*

- For some figures, you average from 70-90N, for the tables you use either 60-90N or 75-90N. It would be more consistent if your figures and tables matched and one could follow the impacts of interest from figure to tables etc. Please consider changing this.
*We now average the figures of concern (aerosol size distributions) between 75° and 90°N instead of 70° and 90°N to be consistent with the averages displayed in Tables 5 and 7, and with most averages given in brackets.*

- Parts of the text is very oral and parts are too elaborate. Below I make both comments on things that should be changed and comments about how the text itself can be improved.

o P2, L24: "Nowadays" is very oral. Please rewrite. Also include " (. . .) pristine compared to other regions (. . .)" The use of depleted here make it sound like the aerosols have been removed. Please rewrite.
*Changed to:*
*"Compared to other regions, the present-day Arctic air is exceptionally pristine, and aerosol levels are very low."*

o P2, L32-P3,L8: May be a bit hard to follow because the indirect effects are described before the general radiative effects of clouds. Consider a rewrite to change the order.

*We shifted the effects of aerosol-cloud interactions on radiation to the paragraph where the CREs are described.*

o P3, L18: "Re-emission of SW"? Please rewrite!
*Changed to:*
*"Aerosol scattering of shortwave (SW) radiation tends to cool the atmosphere, whereas absorption of SW and longwave (LW) radiation tend to warm it (Boucher et al., 2013)."*

o P3, L23-27: Suggest to use the same terms for SW and LW description. For LW, the emissivity includes the water path and the temperature is height dependent. I suggest to make the definitions a bit more tidy.
*Changed to:*
*"Similar to aerosol particles, clouds impact the Earth's radiation budget by absorption and emission of LW radiation (warming) and scattering of SW radiation (cooling). To a smaller extent, LW radiation is also scattered and SW radiation absorbed (Chou et al., 1999; Slingo 1989). The absorption and emission of LW radiation is a function of the emissivity of the cloud (which depends on microphysical cloud properties and the water path), the (height-dependent) cloud temperature, and the surface temperature (Corti and Peter 2009; Chen et al., 2006; Alterskjaer et al., 2010; Shupe and Intrieri, 2003). The scattering of SW radiation is a function of the microphysical cloud properties, of the cloud water path, of the solar zenith angle, and of the surface albedo (Corti and Peter, 2009; Liou et al., 2002; Shupe and Intrieri, 2003)."*

o P4, L11: a bit confusing to have figure references in a listing of the main goals of the paper. Suggest to move this.
*We moved the figure reference after the goals and changed the text to:*
*"Figure 1 provides a simplified overview of how the increase in Arctic temperature can affect radiation. The most important interactions between atmospheric variables, aerosols, clouds, and surface properties are included. The figure shows that the increase in temperature directly affects sea ice, specific humidity, and aerosols. Changes in these variables can then directly or indirectly impact clouds and radiation."*

o Figure 1: A bit confusing. Why not use red for increase and blue for decrease?
*We adapted the figure following your suggestions.*

o P5, L12: Does lowering the CDNC threshold affect the global radiative balance?
*We used a model version that was retuned, which is now explicitly mentioned in the text.*
*Changes to:*
*"Thus, we decided to use 10 cm$^{-3}$ as a lower threshold for the CDNC everywhere and retuned this new model version."*

o P8,L10: Define the abbreviations COADS and AMVER.
*We did this for COADS. For AMVER, the acronym is better known than what it stands for, and the full name of AMVER has changed several times. Therefore, we rather refer to the homepage with a footnote.*

o P9, paragraph starting on line 18: I find this very confusing and suggest a clarification of what you mean in this paragraph. Do not see the link between these two sentences. The reasoning therefore fails.
o P9, L25: Please rewrite to "(. . .) can be considered as a realistic (. . .)"
o P9 General comment to the justification of a tenfold increase of the ship emissions. I suggest a rewrite of this discussion. You increase emissions so as to see a signal and try to justify it afterwards, while at the very end of the paragraph state that the emissions are now "probably too high". Increasing your emissions to get a signal is fine. To discuss that emission estimates may be too low is fine. Your emissions may well be an upper estimate. However, the discussion is long, a bit vague and a bit on the defensive side and makes the reader question whether the authors question their own reasoning here.

*We rewrote the text to make it clearer (page 8 in the new document). We tried to account for all of your suggestions.*

o P9, L30: Please define your abbreviations and give references to the models used. Did you use RCp8.5 for future simulations here? Please specify.
*Changed to:*
*"Both SIC and SST are prescribed in ECHAM6-HAM2. For future conditions, we used model results from the Earth System Model MPI-ESM as input (simulation for the climate model intercomparison project phase 5 (CMIP5), RCP8.5; Giorgetta et al., 2013)."*

o P10, L9: The results would have appeared more robust if the sic and SSTs used were an average taken over eg a ten year time period centered at 2004 and 2050. Using one year (2003) and one ensemble (2050) to test the robustness of your choice of sic and sst is too week.
*We agree that the interannual variability in SIC and SST is large and only looking at one other ensemble might not be sufficient to confirm that our results are robust. However, we refrained from averaging SST and SIC over ten years since we wanted to conduct simulations with a realistic state of the Arctic Ocean. By averaging over SIC, less regions are either ice-free or totally covered with ice; instead, more regions with intermediate sea ice coverage exist, which can have impacts on aerosol emissions and clouds. Furthermore, we are not sure how much sense it makes to average SST in regions which are sometimes ice-free and sometimes covered by sea ice.*

*Changed to:*
*"We refrained from averaging SIC and SST over several years (e.g. 2000-2010) to avoid having spurious regions with intermediate SIC and SST. However, the interannual variability in SIC is pronounced, therefore we performed test simulations using SIC and SST from: i) the years 2003 and 2004 from AMIP and ii) the first and the second ensemble members from the MPI-ESM CMIP5 simulation for the year 2050. Overall, the Arctic SIC in 2003 was somewhat smaller than in 2004, and the SIC in the first ensemble member from MPI-ESM was smaller than in the second ensemble member. We found that the basic results and main conclusions do not depend on these differences in SIC but looking at only two years for both present-day and future might not be sufficient to confirm that all our results are robust. In the following, we will always refer to the simulations using SIC and SST from 2004 and future SIC and SST from the first ensemble member of MPI-ESM."*

o P11,L10: reference needed after statement "Furthermore, most models (. . .) prevalent in the Arctic".
*Changed to:*
*"Furthermore, models of different types generally have problems to reproduce the structure of mixed-phase clouds prevalent in the Arctic (Morrison et al., 2009; Klein et al., 2009; Fan et al., 2011; Morrison et al., 2011; Possner et al., 2017), and the future sea ice extent as well as the prescribed aerosol emissions are highly uncertain (Collins et al., 2013)."*

o P11,L17: When you say "significantly" here is there a statistically significant change? Please specify.
*Yes. Supplementary Fig. 4 shows the changes. Moreover, we added the following sentence at the end of 2.4:*
*"Throughout this paper, the term "significant" is interchangeable with "statistically significant"."*

o P11,L18: How much does the SIC decrease? Please specify.
*We added the figures showing SIC (Fig. 2).*

o P12,L3: "(. . .) modified (. . .)" What does modified mean here? Consider removing.
*It refers to the fact that the sea salt parameterisation by Long et al. 2011 has been extended with corrections for SST, as mentioned in Section 2.1.2.*
*Changed to:*

*"Present-day emissions are a factor of ≈ 3 lower in our simulations compared with Browse et al. (2014), which results from the differences in the two parameterisations (Gong, 2003; Long et al., 2011, with SST corrections) as shown in the study of Long et al. (2011).*

o P13,L5: Is it only the change in CCN concentration that affects CDNC? Not moisture availability? At what supersaturation do you calculate your CCN concentration? Does the average supersaturation change between the runs?

*Thank you very much for pointing this out. We checked and found indeed that not only the changes in aerosol particles, but also changes in meteorology are responsible for the increases in CDNC. In our model, the CDNC depends on the calculated CCN concentration and the lower threshold of 10 cm$^{-3}$. The CCN concentration is calculated interactively following Köhler theory (parameterisation of Abdul-Razzah and Ghan, 2000) based on the aerosol size distribution and the maximum supersaturation. The maximum supersaturation depends e.g. on the updrafts, the temperature, and the CCN number concentration. We found that the updrafts available for activation increase between 75° and 90°N below ~750 hPa in early autumn (Supplementary Fig. 7), which contributes to the enhancement in CCN concentration.*
*Changed to:*
*"In general, the number of aerosol particles acting as CCN increases in the future, which leads to enhanced CDNCs (Fig. 4d). The increase in the number of CCN is not only caused by the increases in oceanic aerosol emissions, but also by changes in meteorology: the updrafts available for activation increase in the boundary layer between 75° and 90° N in early autumn (Supplementary Fig. 7), which supports the formation of cloud droplets in this region."*

o P13, L22: Please explain why the increase in ICNC near the surface is due to the increase in CDNC.

*This explanation was wrong. We thought that the increase in CDNC (as well as the increase in droplet radius) increases the contact freezing rate, but this is only important in limited areas far north. The simulated increases in ICNC are due to enhanced convection.*

*Changed to:*
*"The increase of ICNC near the surface is mainly caused by enhanced convection, which leads to small but numerous simulated ice crystals following the temperature-dependent empirical parameterisation of Boudala et al. (2002)."*

o P13,L25: insert (not shown) after (near Svalbard).

*We show this now in Supplementary Fig. 9. Furthermore, we correct "where precipitation is most enhanced" to "where convective precipitation is most enhanced" (which dries out the atmosphere and thus decreases cloud cover).*

o P16,L2: ((. . .) except over the Arctic Ocean (. . .)" This is over the sea ice?! This should most definitely be specified.

*Correct. Changed to: "except over sea ice"*

o P16,L6: "since the clouds (. . .), more SW radiation can be absorbed (. . .)". Consider rewriting this for clarity.

*Changed to:*
*"Part of this warming might be caused by BC and dust aerosols above clouds (Supplementary Fig. 10): the clouds reflect more SW radiation than the snow/ice-free surface and part of the scattered SW radiation can also be absorbed by aerosol particles causing an increase in aerosol absorption as compared to clear-sky conditions (see e.g. Myrhe et al., 1998)."*

o P16;L13: What does the optical thickness change from and to? If you do not give the numbers you need to specify that it is not shown.

*We added the numbers in brackets and in Tables 2 and 3.*

o P17,24: Please insert "(not shown)" after "temperature".

*We added the figure for surface temperature changes in early autumn to the Supplementary Material. Furthermore, the surface temperature is included in Tables 2 and 3 and now further discussed in the text. The text is changed to:*
*"This is because i) the SW component dominates in these months due to the higher zonal zenith angle and ii) the surface temperature over the central Arctic Ocean does not show pronounced increases like in early autumn (Table 2), therefore not enhancing the LW CRE. The surface temperature even decreases in some regions because melt ponds on ice can have temperatures higher than 271.38 K (but below 273.16 K) in late summer, while the SST is 271.38 K in gridboxes with 0<SIC<1 (equilibrium conditions, i.e. heat changes lead to changes in SIC, not SST)."*

o Table 2: Please include the values from the control run to get the relative importance.
*We now include the values from the control runs. Moreover, we splitted Table 2 and Table 3 (in the original document) for better readability into two tables each, one describing differences due to natural changes (Tables 4 and 5 in the new document) and one describing changes due to ship emissions (Tables 6 and 7 in the new document).*

o P22, section 3.2.1 general comment: Your hypothesis is very likely, but can you verify in your model that this is the case? Perhaps you could perform some sensitivity test? Right now this section is not very strong as it just lists model results without any proper discussion. It would also be good to include a vertical cross section of the aerosol change. This would be beneficial for the next section as well when discussing freezing.
*We added Supplementary Fig. 13 to strengthen our argumentation and discuss it in the text. Changed to:*
*"The additional aerosol particles emitted by ships provide additional surfaces for the condensation of gaseous sulphuric acid. Thus, the vertically integrated condensation rate of sulphate increases where the ship emissions occur (not significant; Supplementary Fig. 13b). The vertically integrated nucleation rate of sulphate shows neither a clear decrease nor a clear increase along the shipping paths (Supplementary Fig. 13d); if the increase in condensation suppressed nucleation, we would expect a decrease in the nucleation rate. However, the vertical cross section of aerosol particles in the nucleation mode shows that the number concentration indeed decreases significantly near the surface (Supplementary Fig. 13f)."*

o P22,L29: How much does the Liquid water path /mass increase?
*We now include the LWC in Fig. 11 (subfigures e and f).*

o P23,L2: Please include the numbers you are describing in the text.
*Changed to:*
*"Using satellite data, Christensen et al. (2014) studied the effect of ship tracks on both mixed-phase and liquid clouds. In the late summer of 2050, the clouds that are impacted by ships in our simulations are mostly liquid. Therefore, we restrict our comparison to the influence of ships on liquid clouds. Consistent with the observations by Christensen et al. (2014), we also found decreases in the effective radius and increases in cloud optical thickness. The relative changes in effective radius are larger in their observations (-20% at cloud top height) than in our simulations (-2% to -4% at altitudes below 500 hPa; averaged between 75° and 90° N, whereas changes in cloud optical thickness compare well (+20% in both studies, averaged between 75° and 90° N). The LWP slightly decreases in their analysis (-1%; in-cloud); in contrast, it increases in our simulations (+17%; all-sky, averaged between 75° and 90° N). While our simulated precipitation shows no clear trend, the results by Christensen et al. (2014) suggest that ship emissions delay precipitation by enhancing cloud lifetime. The different results could be explained by the location of the ship tracks analysed by Christensen et al. (2014): the majority of their samples lie between 45° S and 45° N, and only very few datapoints are from the Arctic. Precipitation formation at high latitudes differs considerably from that at low latitudes since e.g. convection is usually much more important at low latitudes."*

o P25,L32: The significant areas are not large and looking at the figures it looks like the numbers you are giving here are averaged over the whole region. If so, please make it clear that this number is not only including significant changes.

*We wanted to highlight with the given numbers that the local significant changes are small in absolute amount. We removed the numbers since they confused more than they helped. In the next sentence, it is mentioned that the changes in radiative forcing of BC deposition are much smaller than the changes in CREs.*

o P25,L32: The figure reference should be to figure 13(f)?
*Yes, indeed!*

o P27,L8: "While the CRE (. . .)" is it not the change in CRE? Same goes for line 7 and the figure text to figure 14.
*Yes, thank you.*

o P27,L9: The change in optical thickness is significant in very small regions. This should be mentioned. Also when the average numbers are given in the following sentence, make clear that these numbers are not significant.
*The changes are now significant and more widespread in the new simulations.*
*Changed to:*
*"While the changes in CRE caused by changes in cloud cover and cloud top altitude are not significant (Fig. 15a-d), the increase in cloud optical thickness leads to significant decreases and increases in the SW and LW CRE, respectively (Fig. 15e, f). Averaged between 75° and 90° N, the increased optical thickness changes the SW CRE by -4.6 W m$^{-2}$ and LW CRE by 0.52 W m$^{-2}$ in late summer (significant)."*

o P27, L14: Are these numbers significant?
*Yes; we now mention that.*

Technical corrections: - Polarstereographic maps: Please insert a few more latitude lines, perhaps at the boarders for averages that you use: 70N and 75N.
*We inserted every 5° a latitude line.*

- Figures using a blue to red color scale: The lightest colors are impossible to separate in printed figures. You need to improve this color scale. At the same time, consider to use a white color surrounding zero so that values at this separation (zero values) do not come out in color.
*We added a white surrounding around zero and adapted the color scales for many figures.*

**The following minor changes were adapted following the referees suggestions:**
o P2, L17: Remove "where some (. . .) are labelled"
o P2,L20: Replace "until" with "before"
o P2, L22: Remove "(cruise ships)"
o P2, L28: Remove "and can (. . .) (Vali, 1985)" It is of no relevance here and only distracts the reader.
o P2, L32: Make part of the previous paragraph.
o P3, L1: Perhaps mention why smaller droplets increase the cooling effect of clouds?
o P3, L31: Suggest new paragraph before "How Arctic clouds (. . .)"
o P4, L22: Suggest to remove: "HAM2 (. . .) modes." and move "To link (. . .) implemented (. . .)" to the end of the next paragraph, after "(. . .) sedimentation".
o P6, L13: Suggested rewrite: "(..), we used an inventory described in (. . .)"
o P6,L19: Remove "more equations can be found therein"
o P8,L15: Remove "in addition". It is a bit confusing
o P8, L19: Please replace the word "exploit"
o P8, L21: Consider removing the sentence "We processed (. . .)". Too detailed information in my opinion.
o P9, L10: Suggest to remove "(non- (. . .) control)".
o P11, L3: suggest to remove "(eg. SW radiation, temperature)". It is not necessary.
o P11,L3: Rewrite to "change considerably"

o P11,L9: Rewrite to "deviate considerably"
o P11, L17: Rewrite "in the vicinity". Suggest "Over the arctic ocean"
o P13,L3: Please insert the "(. . .) increases in the future (. . .)"
o P13,L8: "averaging over cloudy and non-cloudy conditions". This is a bit confusing.
Are you writing about allsky conditions? Also, please replace increases with increased.
o P16,L10: cooling effect vs warming effect. Please specify the actual numbers here.
o P16,L14: Consider replacing disentangle by "distinguish between"
o P22,L7-8: Please insert "(not shown)" after "late summer" and after "significant for
OC".
o P22,L12: Please remove "using the hypsometric equation". Redundant.
o P22,14: Please move your reference to figure 9b to directly after "800 hpa" to avoid
confusion.
o P25,L21: How much does the optical thickness change. You should strengthen your
discussion by describing the actual model results.
o P25,L27: "(. . .) under clear-sky conditions." Insert "(not shown)" here.
o P27,L4: please consider changing the numbers here to (-2 to -20 Wm-2)
o P27,L5: please insert "(not shown)" after "correlated with ship emissions".
o P27,L16: Please insert "very": "(. . .),ship emissions lead to a significant, but very
weak (. . .)"
o P27,L20: Suggest to add "in limited regions" or something similar after "(. . .) and lead
to significant net cooling"
o P32,L16: Insert "a very small" in front of "local warming".
- All figures: Move (a), (b) etc above figures.
- Figure 7: Suggest to use different color scales for positive and negative values in (a)
and (d).

---

## Author Comment (AC2) · 11 May 2018

**Response to Referee 2**

We thank the referee very much for his/her suggestions and comments, especially those concerning the impact of Arctic shipping. They helped to improve the paper considerably.

Note that we conducted new simulations because the ship emissions were shifted by two weeks in the old simulations. This especially affected the impact of transit ship emissions in the Arctic in late summer 2050. Furthermore, we increased the number of simulated years from 10 to 20 for better statistics, as suggested by referee 3. In some cases, the results have thus changed; as an example, the SW CRE now increases significantly with additional tenfold Arctic ship emissions in 2050.

Referee's comments in blue, our replies in *grey and italic*.

Page 1: cite Melia et al. 2016? (Melia, N., K. Haines, and E. Hawkins (2016), Sea ice decline and 21st century trans-Arctic shipping routes, Geophys. Res. Lett., 43, 9720–9728, doi:10.1002/2016GL069315.)
*We added this interesting reference, which fits nicely into the introduction.*

Page 3, line 18: "re-emission" – suggest rephrasing.
*Changed to:*
*"Aerosol scattering of shortwave (SW) radiation tends to cool the atmosphere, whereas absorption of SW and longwave (LW) radiation tend to warm it (Boucher et al., 2013)."*

Page 4, line 8-15: I think the motivation and objective for this study needs a couple of additional lines, e.g., to summarize what the bulk of the literature described above show about the importance of combining all the processes and what is new/unique about the present study.
*We agree with the referee that more literature concerning the impact of ship emissions should have been mentioned and that we did not mention explicitly enough what is new in this study. We thus rewrote and extended the introduction (p. 4, l.23 to the end of the p. 4 in the new document).*

Page 4, line 11: I don't see that this goal is sufficiently addressed in the paper. The model is run with fixed SSTs and no quantification of temperature responses. As far as I can tell, Fig.1 shows arrows only from temperature changes to radiative changes. If you want to maintain this as a main objective, you need to come back to it later in the manuscript in a better way. However, I think that disentangling the aerosol-radiation-cloud interactions is a sufficient objective in itself.
*We agree with the referee. Since both SST and SIC are fixed, talking about temperature feedbacks is misleading. We can only refer to the impact on temperature by looking at the different radiative forcings. Therefore, we delete "Our goal is to draw conclusions about how changes in radiative forcings and radiative effects may feed back on temperature;"*

Fig. 1: I like the figure, but find the colors a bit confusing. E.g., use of blue from less sea ice to more aerosols. Perhaps use red for increases and blue for decreases? Or add colors.
*We changed the figure following the referee's suggestions (in line with the comment of referee 1).*

Page 5, line 8: "simplistic treatment": please specify.
*Changed to:*
*"This ensures that the global CDNC is not unrealistically low due to missing aerosol species in the model such as nitrate or due to the simplistic model description of organics (no explicit treatment of secondary organic aerosols; neglection of marine organics)."*

Page 5, line 12: changing CDNC - does that affect the global radiative balance?
*We used a retuned model version. This is now explicitly mentioned in the text.*
*Changed to:*

*"Thus, we decided to use 10 cm⁻³ as a lower threshold for the CDNC everywhere and retuned this new model version."*

*Only partly. It is a compromise between accounting for the low aerosol concentrations in the Arctic and missing aerosol sources in the model. We now included observational data of Arctic CCN/CDNC.*
*Changed to:*
*"In the standard ECHAM6-HAM2 setup, a minimum CDNC of 40 cm⁻³ is implemented. This ensures that the global CDNC is not unrealistically low due to missing aerosol species in the model such as nitrate or due to the simplistic model description of organics (no explicit treatment of secondary organic aerosols; neglection of marine organics). Without a lower threshold for CDNC, the model might underestimate the CDNC also in the Arctic, where organic aerosol particles are emitted from the sea surface microlayer (Hawkins and Russell, 2010; Bigg et al., 2004; Leck and Bigg, 2005; Chang et al., 2011). However, since the Arctic is a remote environment with low aerosol concentrations, observations show that the value 40 cm⁻³ is often undershot in this region: between July 15ᵗʰ and September 23ᵗʰ, Bigg and Leck (2001) measured daily median CCN concentrations between 15 and 50 cm⁻³ at a supersaturation of 0.25%. In July 2014, Leaitch et al. (2016) found a median CDNC of 10 cm⁻³ for low-altitude clouds (cloud top below 200 m) and of 101 cm⁻³ at higher altitudes. In October 2004, McFarquhar et al. (2007) conducted aircraft measurements in single-layer stratus clouds and found averaged cloud droplet number concentrations of 43.6±30.5 cm⁻³. Applying the standard CDNC threshold of 40 cm⁻³ would drastically reduce the influence of changes in the CCN concentration and therefore impede aerosol-cloud interactions. Thus, we decided to use 10 cm⁻³ as a lower threshold for the CDNC everywhere and retuned this model version. The studies by Bigg and Leck (2001) and Leaitch et al. (2016) indicate that values even below this lower threshold can occur. While these measurements are representative for a specific point, our model represents average values over a larger area (1.875° × 1.875°), which should be less variable than a point measurement. Nevertheless, we acknowledge that the threshold of 10 cm⁻³ could still be too high under certain conditions. In the model, this threshold is occasionally hit, e.g. over the central Arctic Ocean or in the subtropics. "*

*We followed the referee's suggestion and combined the sections. Marine organics are not included. We now include the following sentence:*
*"Marine organic aerosol emissions are not considered in this study."*

*Changed to:*
*"In ECHAM6-HAM2, dust and BC particles (also those emitted by ships) can act as INPs in the immersion mode when transferred to the internally mixed mode."*

*The section about the ship emissions is indeed not clearly written, which leads to this misunderstanding. We reformulated the paragraphs about the ship emissions as well as Section 2.2. The difference between **arctic_2050** and **arctic_2050_shipping** is that the first does not include the Arctic ship emissions by Peters et al. (2011), while the latter includes these emissions enhanced by a factor of 10.*

Table 1: would be useful to add references for the emissions as well.

*This is a good suggestion, which certainly helps the reader. We extended Table 1 accordingly.*

Page 8, lines 20-22: I think these two sentences are excessively detailed.
*Sentences are deleted.*

Page 9, line 26: first you justify the increase, then you say it is probably too high?
Consider revising for clarity.
*We rewrote this section (p.8, l.10-31 in the new document).*

Page 9, line 27: is it possible to add a reference?
*When looking for a reference, we found that this sentence is too speculative and therefore we deleted this sentence.*

Page 10, line 14: "naïve stipling approach" is not good language. Does this refer to a standard student's t-test? Please clarify/change.
*The naive stippling approach refers to the following: "with this approach, a significance test is calculated for every gridpoint and all gridboxes are stippled where the p-value is smaller than 5% (for a significance level of $\alpha=0.05$)." The wording "naive stippling" is used in the study by Wilks (2016) and we reuse it in lack of a better expression.*

Page 11, line 1: figures show, not "will show". Consider changing the language.
*We changed this.*

Page 11, line 17: consider providing numbers or showing results in a supplementary material.
*We show now the changes in the Supplementary Material (Supplementary Fig. 4) and in Tables 2 and 3.*

Page 11, line 18: perhaps I misunderstand the language, but isn't the increase in wind speed following the reduced SIC the main reason for the increased DMS and sea salt emissions, and hence for the burden increase? Or are there other mechanisms, related to e.g., scavenging due to lower SIC that dominate the burden change? Please clarify.
*The simulated sea salt emissions are a function of SIC, which acts as a barrier between the ocean and the atmosphere. At SIC=1, no sea salt and DMS is emitted from the ocean. In regions where SIC does not change, both wind speed and SST affect the emissions. We now again explicitly mentioned this in the result section.*
*Changed to:*
*"Over the central Arctic Ocean, the decrease in SIC (Fig. 2) enables emission fluxes of DMS and sea salt, which significantly increase their burdens (Supplementary Fig. 4; Tables 2, 3). As a second-order effect, significant increases in $u_{10}$ (Supplementary Fig. 5) over the central Arctic Ocean in early autumn increase sea salt and DMS emissions. In regions where the SIC does not change, both (insignificant) changes in $u_{10}$ and changes in SST (Supplementary Fig. 6) affect DMS and sea salt emissions, and thus their burdens. For example, the decrease in the sea salt burden over the Bering Strait is due to the decreases in SST (caused by a model bias in the MPI-ESM sea surface temperature compared to AMIP) and $u_{10}$."*

Page 11, line 26: caused by what?
*Since the precipitation increases, also the wet deposition is enhanced, which is the most important removal process for BC and OC in the Arctic in our model. (All BC and OC emissions are the same for the two simulations.) We now mention this in the paper.*
*Added the following sentence:*
*"The smaller BC and OC burdens can be explained by the increase in precipitation, which leads to enhanced wet deposition (the BC and OC emissions are identical between the two simulations)."*

Page 11, line 32: JJA/August – do you consider a different period here? Please clarify.
*Yes, for a fair comparison, we look at the same periods as the other two studies. We rewrote the text to make it clearer (p.12, l.28).*

*"When we compare our results to other studies, we average over the same time and space as they do for a fair comparison."*

Page 12, line 1: at some point it would be good to show/describe in detail the changes in variables such as SIC between 2004 and 2050. Could be added in a supplementary material.
*We added the figures showing SIC to the main paper (Fig. 2).*

Page 12, line 4: absolute emissions in 2004 or absolute emission changes? Please clarify.
*Changed to:*
*"The absolute present-day emissions..."*

Section 3.1.2: are the same general features seen during summer?
*Yes. If the season is not specified, (qualitative) results refer to both late summer and early autumn, as mentioned in the paper. We now provide more quantitative information in the text for the two seasons. Furthermore, we included two new tables (Tables 2 and 3), which also provide more quantitative information for the two seasons.*

Page 16, line 13: again, it would be helpful to have the actual numbers.
*We added the numbers.*

Page 17, line 4: perhaps instead say "a strengthening of the direct aerosol effect" since it is in fact much stronger in 2100?
Page 17, line 5-7: I'm not convinced it makes sense to compare these numbers since the foundation and model experiments are so different. Unless you're able to disentangle effects of experimental differences in more detail, I don't see that this section add much information of value and it could be left out.
*We agree with the referee and take this comparison out of the paper (this corresponds also to the referee's comment above).*

Figure 5: very hard to distinguish statistically significant areas.
*We changed the stippling from points to lines for better visibility.*

Page 17, line 8: please add numbers or relative change.
*We added Supplementary Fig. 11.*

Page 17, line 9: if I follow correctly, these results are still without any changes in anthropogenic aerosol emissions, so a small effect due to changes in BC deposition is to be expected, unless there are large changes in the scavenging. Could be useful to remind the readers of this. In fact, even under RCP8.5, anthropogenic aerosol emissions decline strongly through the century, which could perhaps reduce this forcing altogether.
*This is correct. When discussing changes in the size distribution, we now remind the reader that the emissions are identical ("The smaller BC and OC burdens can be explained by the increase in precipitation, which leads to enhanced wet deposition (the BC and OC emissions are identical between the two simulations).") Moreover, we now mention that the anthropogenic aerosol emissions decline under RCP8.5.*
*Changed to:*
*"Also most prescribed aerosol emissions (excluding DMS terrestrial emissions, biogenic organic carbon emissions, and ship emissions) follow RCP8.5, which decline in most industrial sectors from 2004 to 2050."*

Page 22, line 7: A comparison with previous work using the Peters et al. inventory (without the x10 enhancement) could be useful, e.g., Ødemark et al. 2012; Dalsøren et al. 2013.
Dalsøren, S. B., Samset, B. H., Myhre, G., Corbett, J. J., Minjares, R., Lack, D., and Fuglestvedt, J. S.: Environmental impacts of shipping in 2030 with a particular focus on the Arctic region, Atmos. Chem. Phys., 13, 1941-1955, https://doi.org/10.5194/acp-13-1941-2013, 2013. Ødemark, K., Dalsøren, S. B., Samset, B. H., Berntsen, T. K., Fuglestvedt, J. S., and Myhre, G.: Short-lived climate forcers from current shipping and petroleum activities in the Arctic, Atmos. Chem. Phys., 12, 1979-1993, https://doi.org/10.5194/acp-12-1979-2012, 2012.

*Thank you very much for this suggestion. We now use the study from Dalsøren et al. (2013) for comparing our radiative forcings/effects in Sections 3.2.3 and 3.2.4.*

Page 22, line 8: the maximum changes occur at the same location as the emissions; however, there are statistically significant increases over much larger areas. Should be specified.
*Changed to:*
*"The maximum increases in burden (see Fig. 9b) occur at the same locations as the ship emissions, but significant increases can spread over a large part of the Arctic (see Fig. 9c), as shown for the example of BC."*

Page 22, line 11-17: are these shifts large enough to have notable implications, e.g., for forcing? Possible to discuss to add some context?
*The changes in the size distribution can have an effect on the radiative forcing. However, we find that the radiative forcing by aerosols hardly changes. In general, changes in the number size distribution can not only affect the aerosol radiative forcing, but also the number of CCN.*

Page 25, line 18-19: actual magnitudes would be useful.
*Numbers are now included.*

Section 3.2.3: this section is missing a discussion of and connection to studies of the radiative forcing of shipping, both in the Arctic and overall to global impacts. This is important given that main conclusion of the study concern the negligible impact of shipping aerosol emissions. In particular, a discussion of the impact of shipping found in studies that do include explicit treatment of aerosol-cloud interactions and/or offline radiative transfer calculations could be important.
*We now include a comparison with the work from Dalsøren et al., 2013 (p.33, l.10-25 and p.34, l.10-15).*

Page 32, line 20-25: be careful about the phrasing of this conclusion, as it does not cover other effects of shipping emissions, such as NOx-induced ozone changes and CO2.
*Thank you for this comment. We include now the sentence:*
*"Furthermore, this study does not account for ship-induced changes in greenhouse gases (e.g. $O_3$, $CO_2$), which are also important forcers (Dalsøren et al., 2013; AMAP Assessment, 2015)."*

---

## Author Comment (AC3) · 11 May 2018

**Response to Referee 3**

We thank the referee very much for his/her comments and suggestions, especially those highlighting that the uncertainty from this study must be communicated more clearly. They helped to improve the quality of the paper.

Note that we conducted new simulations because the ship emissions were shifted by two weeks in the old simulations. This especially affected the impact of transit ship emissions in the Arctic in late summer 2050. Furthermore, we increased the number of simulated years from 10 to 20 for better statistics. In some cases, the results have thus changed; as an example, the SW CRE now increases significantly with additional tenfold Arctic ship emissions in 2050.

Referee's comments in blue, our replies in *grey and italic*.

Gilgen et al present a set of sensitivity studies with the atmospheric GCM ECHAM-HAM. The control simulation is driven by conditions approximately representative for present-day, three sets of differing boundary conditions are then computed: (a) increased sea surface temperatures and decreased sea ice cover, (b), in addition, changed aerosol emissions, and (c) in addition, further ship emissions. Each of the simulations is run for a short period of ten years. A large set of results is presented. The study is to a large extent based solely on the results of the one model and thus the hypotheses developed are strongly dependent on the chosen parameterisations. Very little comparison to data (for the control simulation) is presented. In one paragraph, the cloud radiative effects are compared to SHEBA data – from this it seems that the model has a very large bias. I believe it would be necessary for an improved paper to at least show some evidence that the model performs satisfactorily in the Arctic in comparisons to observations, before the results from the sensitivity studies can be considered meaningful.

*We agree with the reviewer that the results of this study are uncertain, especially because they were conducted with only one climate model. Therefore, more information about the performance of the model in the Arctic will help the reader to put the results of this study into context. Thus, we added a comparison of some key variables in the Supplementary Information: AOT; BC and sulphate concentrations at the surface; cloud cover; LWP and IWP; cloud radiative effects (SW, LW, and net) at the surface and at the TOA. The AOT and the IWP are underestimated, whereas all other variables compare well (the bias in the SW surface CRE mentioned in the old paper version was likely caused by the local surface albedo of the observations; see Supplementary Information for more details). Despite the model bias in AOT and IWP, we consider our results meaningful: in our opinion, model biases do not imply that the simulated changes are necessarily wrong. Vice versa, also a model that compares perfectly to present-day observations can make wrong projections for the future. Furthermore, we want to stress that observations in the Arctic are sparse and sometimes also uncertain.*
*Next to the detailed comparison that we provide in the Supplementary Information (the most important results of which are mentioned in the paper), we added the following paragraph to the results section:*

*"Heterogeneous freezing is still an active field of research, and contradictory evidence exists concerning the ability of combustion aerosols to act as INPs (Kanji et al., 2017). Laboratory results suggest that soot starts initiating freezing at temperatures ≤-30°C (Kanji et al., 2017, Fig. 1-7). On the other hand, Thomson et al. (2018) found an increase in INP concentrations in ship tracks at higher temperatures. The increases were small at temperatures around −20°C, moderate at −25°C (≈ 0.5 $L^{-1}$; saturation ratio of 1.22), and quite pronounced at -30°C (≈ 2 $L^{-1}$; saturation ratio of 1.32). The ship plumes were measured near the port of Gothenburg (57.7° N, 11.8° E) in 2013 and 2014, and the meteorology in general represented climate conditions of the late-autumn maritime North. If ship exhaust (not necessarily the BC particles) can indeed induce freezing at higher temperatures than in the laboratory-based BC-parameterisation used in our model, the impact on*

*cloud ice could be larger than in our simulations, especially in early autumn when temperatures are colder."*

At two instances, the results are compared to previously-published results for similar scenarios. It is astonishing how different the results are. A key hypothesis is that sea salt emissions may substantially increase with decreasing sea ice coverage. Fundamentally, this is no surprise, so the question is how large this could be quantitatively. Unfortunately the two other model studies reported are much more different from the model presented here than the change due to sea ice retreat (one model has a factor of 3 more, the other, a factor of 1000 less emission flux in present-day conditions). Also the radiative forcing due to aerosol-radiation interactions is very different between models – the model presented here has a substantially positive forcing, the other model, a negative one. Since such results are easily available from multi-model ensembles (CMIP5 or AEROCOM), it would be easy to put the model the authors use into context, much beyond the two studies cited.

*Our main goal was to compare how future sea salt emissions might change due to sea ice retreat. Therefore, we initally only compared our values with studies that looked specifically at this question. To our knowledge, nearly no CMIP5 model calculates interactively emissions of sea salt, but we agree that other model intercomparison studies can give insight into the spread in sea salt emissions between different models. Therefore, we included results from the study of de Leeuw et al. (2011) and added the following sentences:*

*"Note that at the present state, sea salt emissions are highly uncertain and differ considerably between models: a comparison of 11 chemical transport and global climate models shows that the global annual mass emissions of sea salt lie in the range between 2.2 and 9.9 $\times$ 10$^{12}$ kg yr$^{-1}$ for 9 of the models; 2 models calculate higher emissions of 22 $\times$ 10$^{12}$ kg yr$^{-1}$ and 118 $\times$ 10$^{12}$ kg yr$^{-1}$ (de Leeuw et al., 2011). For comparison, our simulated value (1.3 $\times$ 10$^{12}$ kg yr$^{-1}$) is on the same order of magnitude as most of these models, but lower than in all of them because the parameterisation does not account for the contributions from spume drops (Long et al., 2011). Our simulated absolute increases in sea salt mass emissions might therefore be underestimated."*

*In the conclusions, we highlight again the uncertainty:*
*"Compared with observations, our model has a low bias in AOT and cloud ice, which could impact the absolute changes in the radiative aerosol forcing and the CREs. Furthermore, when we compare our results with other modelling studies that looked at natural aerosol changes with declining sea ice, we find large intermodel differences, e.g. concerning sea salt emissions. This highlights that the results from this study – as from any climate model study projecting the future – are uncertain."*

When it comes to the interpretation of the results, much is left for speculation. If the authors choose to have a pure modelling study, why don't they at least precisely clarify the processes that change? Why not budgets for changes of CCN, INP? A table that lists all relevant numbers (e.g. for the entire region, and split for open ocean and sea ice surfaces) as simulated for the different scenarios would be useful (emission fluxes, CCN, INP, cloud particle concentrations, LWP, IWP).

*We agree that our result's section is somehow speculative. It is not straightforward to understand 'what is going on' in the model, and understanding every single result in detail would be beyond the scope of this study. The large advantage of using a climate model is that it accounts for many processes and their couplings. In this case, the main goal was to look at aerosol-cloud interactions and radiative forcings with a complex aerosol-climate model, which has the drawback that the results stem from a combination of processes that cannot easily be separated.*
*Unfortunately, we do not calculate intermediate variables such as CCN or INP concentrations. The model calculates CDNC and ICNC based on the size and chemical composition of the aerosol particles, temperature, and supersaturation.*
*However, we agree with the referee that a table showing averages over the whole Arctic region, open ocean, sea ice, and regions where sea ice has melted is very interesting. Thus, we included additionally Tables 2 and 3 in the paper, which show averages for sea salt and DMS burdens, AOT,*

*LWP, IWP, cloud cover, CDNC burden, and surface temperature. The results of these tables are discussed in the text.*

Specific comments:
p3 l30 – this is not "generally" true, e.g. not in summer (as the following sentence correctly acknowledges)
*Changed to:*
*"Therefore, the LW absorption of clouds becomes more important and can dominate the total cloud radiative effect depending on the specific time and location."*

p3 l32 – but it is likely a small effect (Pithan and Mauritsen). What is the reference for the following sentence ("generally...")?
*Changed to:*
*"How Arctic clouds and their radiative effects will change in the future is still an open question. Generally, both the SW and the LW cloud radiative effect (CRE) are expected to become stronger when more CCN are available (Mauritsen et al., 2011). However, compared to other temperature feedbacks, the contribution of changes in Arctic clouds might be small (Pithan and Mauritsen, 2014)."*

P5 l12 – it would be good to report already here whether this threshold is hit, and, if so, how often. It would further be good to analyse whether indeed the lack of nitrate or organics is a major problem of this model for the Arctic.
*This threshold is occasionally hit, e.g. in the subtropics and over the Central Arctic Ocean. We now also provide a comparison with observations, as suggested by referee 2.*

*Changed to:*
*"In the standard ECHAM6-HAM2 setup, a minimum CDNC of 40 cm⁻³ is implemented. This ensures that the global CDNC is not unrealistically low due to missing aerosol species in the model such as nitrate or due to the simplistic model description of organics (no explicit treatment of secondary organic aerosols; neglection of marine organics). Without a lower threshold for CDNC, the model might underestimate the CDNC also in the Arctic, where organic aerosol particles are emitted from the surface microlayer (Hawkins and Russell, 2010; Bigg et al., 2004; Leck and Bigg, 2005; Chang et al., 2011). However, since the Arctic is a remote environment with low aerosol concentrations, observations show that the value 40 cm⁻³ is often undershot in this region: between July 15ᵗʰ and September 23ᵗʰ, Bigg and Leck (2001) measured daily median CCN concentrations between 15 and 50 cm⁻³ at a supersaturation of 0.25%. In July 2014, Leaitch et al. (2016) found a median CDNC of 10 cm⁻³ for low-altitude clouds (cloud top below 200 m) and of 101 cm⁻³ at higher altitudes. In October 2004, McFarquhar et al. (2007) conducted aircraft measurements in single-layer stratus clouds and found averaged cloud droplet number concentrations of 43.6±30.5 cm⁻³. Applying the standard CDNC threshold of 40 cm⁻³ would drastically reduce the influence of changes in the CCN concentration and therefore impede aerosol-cloud interactions. Thus, we decided to use 10 cm⁻³ as a lower threshold for the CDNC everywhere and retuned this model version. The studies by Bigg and Leck (2001) and Leaitch et al. (2016) indicate that values even below this lower threshold can occur. While these measurements are representative for a specific point, our model represents average values over a larger area (1.875° × 1.875°), which should be less variable than a point measurement. Nevertheless, we acknowledge that the threshold of 10 cm⁻³ could still be too high under certain conditions. In the model, this threshold is occasionally hit, e.g. over the central Arctic Ocean or in the subtropics. "*

P6 l1 – it would be good to comment on the results of Eckhardt et al. (ACP 2015)
*Thank you for this reference. We included it in the Supplementary Material when comparing simulated and observed SO₄.*

p8 l6 – 10 years seem very little for small forcings
*We increased the number of years to 20.*

p12 top paragraph – what do these discrepancies by a factor of about 3000 imply for

*Thank you for this comment. We added a comparison with de Leeuw et al. (2011) and mention now the uncertainty in the conclusions (see text above).*

P13 l8 increased
*Corrected.*

p13 l22 – i.e. homogeneous freezing of droplets?
*This explanation was wrong. We thought that the increase in CDNC (as well as the increase in radius) increases the contact freezing rate, but this is only important in limited areas far north. The simulated increases in ICNC are due to enhanced convection.*

*Changed to:*
*"The increase of ICNC near the surface is mainly caused by enhanced convection, which leads to small but numerous simulated ice crystals following the temperature-dependent empirical parameterisation of Boudala et al. (2002)."*

P13 l23 – at con-
stant ICNC?
*Changed to:*
*"Between 500 hPa and 200 hPa, the enhanced ice water content is linked to the increase in ice crystal radius, while the ICNC slightly decreases."*

L13 l27 – indeed surface fluxes? Or rather simply moist adiabat changes?
*The explanation in Abe et al., 2016 is the following: "Because of the reduced sea ice, a more extended open ocean area increased the latent and sensible heat fluxes from the ocean to the atmosphere. Along with the seasonal march, the decreased atmospheric temperatures increased the temperature gradient between the air and sea surface in October. Therefore, the fluxes from the ocean to the atmosphere were enhanced in October rather than in September."*
*In our study, most pronounced increases in cloud cover occur where sea ice has melted (Table 3) and where also the changes in latent and sensible heat fluxes are most pronounced (significant changes; not shown in the paper).*

P16 l2 – it would be important to clarify whether this section refers to the radiative forcing by aerosol-radiation interactions only, or to the effective radiative forcing due to aerosol-radiation interactions, or whether it includes aerosol-cloud interactions.
*We calculate the radiative forcing by calling the radiation scheme once with and once without aerosols, i.e. it is the radiative forcing by aerosol-radiation interactions only. We now remind the reader of this in the results section and highlight in the methods that our radiative forcing refers to all aerosols (not only anthropogenic aerosols). Furthermore, we now apply abbreviations from the newest IPCC report in our paper (e.g. "$RF_{ari}$" or "$ERF_{aci}$") for clarification and avoid the term "Twomey effect".*
*We added this sentence:*
*"As mentioned previously, the aerosol radiative forcing refers to the instantaneous effect of all aerosols on radiation ($RF_{ari}$)."*

P16 l4
– what are the absorbing components, and why is the positive forcing so large? P16 l6
*The absorbing components are BC and dust. Since the warming is only prevalent in the presence of clouds, we assume that the scattering of aerosol particles is less important in the presence of clouds and that the warming of aerosol particles might be enhanced by the higher SW radiation reflection by clouds.*
*Changed to:*
*"If the presence of clouds is considered, aerosol particles warm the atmosphere also over Alaska and northeast Siberia (late summer) and over the whole northern Russia (early autumn; shown in Fig. 7e). Part of this warming might be caused by BC and dust aerosols above clouds (Supplementary Fig. 10): the clouds reflect more SW radiation than the snow/ice-free surface and*

*part of the scattered SW radiation can also be absorbed by aerosol particles causing an increase in aerosol absorption as compared to clear-sky conditions (see e.g. Myhre et al. 1998). Moreover, the scattering of aerosol particles could become less important in the presence of clouds, which increases the relative importance of aerosol absorption to extinction.*

– it would be useful to demonstrate this at least in the supplementary material (since the authors write "not shown" it seems hey have the analysis at hand)
*We added Supplementary Fig. 10.*

p17 l5 – how is the coincidence of approximately the same reduction by 0.2 Wm-2 explained? Is the same thing happening in both models?
*Based on the comment of referee 2, we take this comparison out. In contrast to Struthers et al. (2011), we look at all aerosol particles, which makes a comparison difficult.*

P18 l13 – i.e. the effect is twice as large as observed? The authors should report this analysis as a table or similar.
*We provide now a more detailed analysis in the Supplementary Material.*

---

## Author Comment (AC6) · 11 May 2018

**1 Comparison with observations**

To interpret the results from this model study, it is important to know how well the model performs and to be aware of certain biases. However, comparing all variables of interest with observations is beyond the scope of this paper. We restrict the comparison therefore to AOT, BC and $SO_4$ surface concentrations, cloud fraction/cover, LWP, IWP, and CREs at the surface and at the TOA (SW, LW, and net). We only consider data that covers a large part of our period of interest (July to October) and that reflects present-day conditions (between the years 1998 and 2015). The simulated values refer to the ensemble mean of the simulation **arctic_2004**.

Compared to CALIPSO measurements (60° to 82° N, 2006-2011; Sand et al. 2017, Fig. 6), the AOT in our model (0.037 in late summer and 0.033 in early autumn) is underestimated by a factor of 2 to 3. On the other hand, our simulated AOT over the Greenland/Barents Sea (10° W to 40° E, 75° to 82° N) is 0.034 in late summer and agrees very well with MODIS measurements from 2003 to 2011 (Sand et al. 2017, Fig. 5). The underestimation of AOT in our model compared to CALIPSO measurements can have various reasons, e.g. missing local aerosol sources in the model (e.g. marine organics or gas flaring emissions; Hawkins and Russell 2010; Chang et al. 2011; Stohl et al. 2013), an underestimation of aerosol transport from midlatitudes to the Arctic (Bourgeois and Bey 2011), uncertainties in the optical properties and emissions of aerosols (e.g. for BC, see Bond and Bergstrom 2006; Bond et al. 2013), and/or the neglection of spume drops in the sea salt parameterisation by Long et al. (2011).

Next to AOT, we also compare our simulated BC and $SO_4$ concentrations with observations. The impact of future BC and $SO_4$ emissions from Arctic shipping depends on the background concentration, which is mainly determined by long-range transport in the case of BC. We compare our simulated BC concentrations with the recent long-term surface observations at Zeppelin (78.92° N, 11.93° E) and Barrow (71° N, 156.6° W; Sinha et al. 2017). Note that earlier studies report higher BC concentrations at the same stations, since they neglected the effect of other aerosol components on the aerosol light absorption coefficient when deriving mass concentrations of BC (Sinha et al. 2017). From July to October, our monthly averages range from 4.3 to $9.7 \, \mathrm{ng \, m^{-3}}$ at Zeppelin and from 6.1 to $12.3 \, \mathrm{ng \, m^{-3}}$ at Barrow. This is in good agreement with the observations, which lie in the ranges 3 to $9 \, \mathrm{ng \, m^{-3}}$ and 3 to $10 \, \mathrm{ng \, m^{-3}}$ at Zeppelin and Barrow, respectively (depending on the averaging period and the derivation method).

We compare our simulated $SO_4$ surface concentrations with the values from Eckhardt et al. (2015). Averaged over the stations Alert (82.5° N, 62.5° W), Zeppelin, and Barrow and the years 2008 and 2009, they report a summer (July to September) value of $103.2 \, \mathrm{ng \, m^{-3}}$. Our simulated value (averaged over the same locations and months) is approximately twice as large ($214 \, \mathrm{ng \, m^{-3}}$). Concerning the large spread between models (Eckhardt et al. 2015), we consider this to be in reasonable agreement with the observations.

We used data from the SHEBA campaign, which took place from October 1997 to October 1998 in the region 180° W to 130°W, 70° to 80° N, to compare LWP and IWP in the Arctic (Shupe and Uttal 2007). Since only few data is available for October, we restrict the comparison to the months July, August, and September. The LWP measurements are based on microwave radiometer retrievals (Westwater et al. 2001). For the derivation of IWP, we use estimates from two different methods: an empirical technique that relates cloud ice water content to radar reflectivity, and the technique described in Matrosov et al. (2002), which relates ice particle size to radar Doppler velocities. Mixed-phase clouds have retrieval values for only the ice component. The uncertainties are $\approx 25 \, \%$ for LWP and up to a factor of 2 for IWP (Matrosov et al. 2002; Shupe et al. 2004; Shupe et al. 2005).

Individual data points are unrealistically high (several thousands $\mathrm{g \, m^{-2}}$), which could be due to erroneous measurements. As an example, the LWP derived from the microwave radiometer becomes wet under rainy conditions, which results in overestimated brightness temperatures and, thus, overestimated LWP (Matthew Shupe, personal communication). In addition to the

original data, we therefore also show results where values larger than $800\,\mathrm{g\,m^{-2}}$ are considered as NAN in Fig. 1. The value of $800\,\mathrm{g\,m^{-2}}$ is somewhat arbitrary, but comparing the original with the processed data helps to identify cases in which the mean is strongly biased by the outliers.

[Figure]

**Figure 1:** Monthly statistics of (a) LWP and (b)/(c) IWP from the SHEBA campaign for July, August, and September. The IWP in (b) is derived from an empirical technique using radar reflectivity, whereas the IWP in (c) is estimated with a technique based on radar Doppler velocities. The original data by Shupe and Uttal (2007) is shown (Jul, Aug, Sep) as well as data where values larger than $800\,\mathrm{g\,m^{-2}}$ are excluded (Jul_thr, Aug_thr, Sep_thr). Orange lines and green triangles show the observed medians and means of each month, respectively. For comparison, the monthly means from our simulations are shown as red squares. The length of the whiskers is restricted to 1.5 times the interquartile range.

The simulated LWP in ECHAM6-HAM2 (average over the SHEBA region) compares well with the observations (Fig. 1a). The simulated and the observed average are similar in July (ECHAM6-HAM2: $72.7\,\mathrm{g\,m^{-2}}$; SHEBA: $68.0\,\mathrm{g\,m^{-2}}$) and August (ECHAM6-HAM2: $82.5\,\mathrm{g\,m^{-2}}$; SHEBA: $96.3\,\mathrm{g\,m^{-2}}$), whereas ECHAM6-HAM2 underestimates the LWP in September (ECHAM6-HAM2: $63.9\,\mathrm{g\,m^{-2}}$; SHEBA: $92.2\,\mathrm{g\,m^{-2}}$). However, in September, the observed average is quite sensitive to the upper threshold we applied for LWP and should therefore be taken with caution.

While the probability distributions of IWP are different for the two different methods (Figs. 1b,c), the average values compare relatively well when the upper threshold is applied. The observed medians in Fig. 1c are (near) zero because cloud ice has a rather low occurrence frequency, i.e. the measured IWP is often zero. In all three months, ECHAM6-HAM2 severly underestimates the average IWP: $13.3\,\mathrm{g\,m^{-2}}$ instead of 34.1 to $34.3\,\mathrm{g\,m^{-2}}$ (depending on the method) in July, $15.2\,\mathrm{g\,m^{-2}}$ instead of 35.3 to $41.1\,\mathrm{g\,m^{-2}}$ in August, and $15.0\,\mathrm{g\,m^{-2}}$ instead of 50.2 to $57.8\,\mathrm{g\,m^{-2}}$ in September. This underestimation is a global phenomenon of ECHAM6-HAM2 (Lohmann and Neubauer 2018, in review) and previous model versions.

Furthermore, we compare our simulated cloud cover/fraction and surface CREs with values reported from the SHEBA campaign (Intrieri et al. 2002). The cloud fractions shown in Fig. 2a are determined from temporal and spatial averages of lidar and ceilometer measurements. Until mid-August 1998, data is derived from the lidar, after that from the ceilometer. The derived surface SW CRE from observations in Fig. 2b relies on model calculations for the clear-sky using single site albedos (ASFG). In Fig. 2c, the net surface CRE is also shown for calculations with line-averaged albedos (CRREL). The ASFG albedo was computed hourly by the Atmospheric Surface Flux Group radiometers. THE CRREL albedo was obtained by the Cold Regions Research and Engineering Laboratory group once per day around solar noon and includes many different ice types (e.g. melt ponds, open water). While the ASFG albedos are directly linked with the observed fluxes, the CRREL albedos are more representative of the SHEBA ice camp area.

The simulated cloud cover compares relatively well with the cloud fraction observed in the SHEBA campaign (Fig. 2a). We noted that the large values of observed cloud fraction (near 100 %) coincide with the time when the instrument switched from lidar to ceilometer. Data derived from ISCCP, averaged from 1982 to 1999 in summer, is smaller in the regions where the SHEBA campaign took place, namely 0.74 in the Beaufort and 0.79 in the Chucki Sea (Wang and Key 2005).

While the simulated LW CRE is in good agreement with the observations (Fig. 2b), the simulated SW CRE is consistently lower. However, the SW CRE depends strongly on the surface albedo. When using the CRREL instead of the ASFG albedo, the observed SW CRE is considerably stronger. This is evident in Fig. 2c, where the net CRE is shown with both surface albedo estimates. Overall, our simulated values compare reasonably with the estimates of net CRE from Intrieri et al. (2002).

Next to comparing the CREs at the surface, we also compare the simulated CREs at the TOA with recent satellite data from the Clouds and the Earth's Radiant Energy System (CERES; Loeb et al. 2018). The CRE values over the poles are more uncertain than over other parts of the world because clear-sky measurements over snow and ice are challenging (Loeb et al. 2018). However, the advantage of satellite products is that they provide data for a larger region and a longer time period than measurement campaigns such as SHEBA. In Fig. 3, interannual monthly means are shown for the period from June 2005 to July 2015. We averaged the data between a) 60° and 90° N and b) 75° and 90° N. The model compares well with the satellite data. Largest absolute deviations occur in July between 75° and 90° N for SW CRE, where the observed value is $-59.0\,\mathrm{W\,m^{-2}}$ and the simulated value is $-49.6\,\mathrm{W\,m^{-2}}$.

To summarise, ECHAM6-HAM2 has a low bias concerning cloud ice and AOT, whereas the simulated BC and sulphate surface concentrations, the LWP, the cloud cover, and the CREs at the surface and at the TOA compare well with observations.

[Figure]

**Figure 2:** Comparison of ECHAM6-HAM2 (markers) with observations (lines) for a) cloud fraction, b) SW and LW surface cloud radiative effect, and c) net surface cloud radiative effect. All figures are adapted from Intrieri et al. (2002). The shown values for ECHAM6-HAM2 represent July, August, and September (placed at the 15th of each month).

[Figure]

**Figure 3:** Comparison of ECHAM6-HAM2 (markers) with satellite-derived observations (lines) for SW, LW and net CRE at the TOA averaged between a) 60° and 90° N and b) 75° and 90° N.

**2 Additional Figures**

[Figure]

**Figure 4:** Sea salt and DMS burdens in 2004 in (a)/(c) and differences between 2050 and 2004 (i.e. between simulations **arctic_2050_EM2004** and **arctic_2004**) in (b)/(d) in early autumn (Sep/Oct). Hatched areas are significant at the 95% confidence level.

[Figure]

**Figure 5:** Wind speed at 10 m altitude in early autumn (Sep/Oct): a) absolute values in 2004 and b) differences between 2050 and 2004 (i.e. between simulations **arctic_2050_EM2004** and **arctic_2004**). Hatched areas are significant at the 95% confidence level.

[Figure]

**Figure 6:** Surface temperature in early autumn (Sep/Oct): a) absolute values in 2004 and b) differences between 2050 and 2004 (i.e. between simulations **arctic_2050_EM2004** and **arctic_2004**). Hatched areas are significant at the 95% confidence level. Note that the SST is prescribed in the simulations and shows no interannual variability.

[Figure]

**Figure 7:** Updraft available for activation in early autumn (Sep/Oct): a) absolute values in 2004 and b) differences between 2050 and 2004 (i.e. between simulations **arctic_2050_EM2004** and **arctic_2004**). Hatched areas are significant at the 95% confidence level.

[Figure]

**Figure 8:** Surface precipitation rate in early autumn (Sep/Oct): a) absolute values in 2004 and b) differences between 2050 and 2004 (i.e. between simulations **arctic_2050_EM2004** and **arctic_2004**). Hatched areas are significant at the 95% confidence level.

[Figure]

**Figure 9:** Convective surface precipitation rate in early autumn (Sep/Oct): a) absolute values in 2004 and b) differences between 2050 and 2004 (i.e. between simulations **arctic_2050_EM2004** and **arctic_2004**). Hatched areas are significant at the 95% confidence level.

[Figure]

**Figure 10:** Total (a)/(b) BC and (c)/(d) dust concentrations in 2004. Figures on the left are averaged over late summer (Jul/Aug), figures on the right over early autumn (Sep/Oct).

[Figure]

**Figure 11:** Radiative forcing of deposited BC in early autumn (Sep/Oct): a) absolute values in 2004 and b) differences between 2050 and 2004 (i.e. between simulations **arctic_2050_EM2004** and **arctic_2004**). Hatched areas are significant at the 95% confidence level. Note that the scale is logarithmic.

[Figure]

**Figure 12:** The impact of additional future ship emissions (**arctic_2050_shipping** versus **arctic_2050**) on the total BC concentration in b) late summer and d) early autumn. In (a)/(c), the reference without additional ship emissions is shown (**arctic_2050**). Hatched areas are significant at the 95% confidence level.

[Figure]

**Figure 13:** The impact of additional future ship emissions (**arctic_2050_shipping** versus **arctic_2050**) on: b) the vertically integrated condensation rate of sulphate on aerosol particles, d) the vertically integrated nucleation rate of sulphate, and f) the number of aerosol particles in the nucleation mode in early autumn (Sep/Oct). In (a)/(c)/(e), the reference without additional ship emissions is shown (**arctic_2050**). Hatched areas are significant at the 95% confidence level.

[Figure]

**Figure 14:** In (b), the impact of additional future ship emissions (**arctic_2050_shipping** versus **arctic_2050**) on the aerosol radiative forcing is shown under clear-sky conditions in early autumn (Sep/Oct). In (a), the reference without additional ship emissions is shown (**arctic_2050**). Hatched areas are significant at the 95% confidence level.

**References**

Bond, T. C., S. J. Doherty, D. W. Fahey, P. M. Forster, T. Berntsen, B. J. DeAngelo, M. G. Flanner, S. Ghan, B. Kärcher, D. Koch, S. Kinne, Y. Kondo, P. K. Quinn, M. C. Sarofim, M. G. Schultz, M. Schulz, C. Venkataraman, H. Zhang, S. Zhang, N. Bellouin, S. K. Guttikunda, P. K. Hopke, M. Z. Jacobson, J. W. Kaiser, Z. Klimont, U. Lohmann, J. P. Schwarz, D. Shindell, T. Storelvmo, S. G. Warren, and C. S. Zender (2013). "Bounding the role of black carbon in the climate system: A scientific assessment". In: *Geophy. Res. Lett.-Atmos.* 118.11, pp. 5380–5552. ISSN: 2169-8996. DOI: `10.1002/jgrd.50171`.

Bond, T. C. and R. W. Bergstrom (2006). "Light Absorption by Carbonaceous Particles: An Investigative Review". In: *Aerosol Science and Technology* 40.1, pp. 27–67. DOI: `10.1080/02786820500421521`.

Bourgeois, Q. and I. Bey (2011). "Pollution transport efficiency toward the Arctic: Sensitivity to aerosol scavenging and source regions". In: *J. Geophys. Res.* 116.D8. ISSN: 0148-0227. DOI: `10.1029/2010jd015096`.

Chang, R. Y.-W., C. Leck, M. Graus, M. Müller, J. Paatero, J. F. Burkhart, A. Stohl, L. H. Orr, K. Hayden, S.-M. Li, A. Hansel, M. Tjernström, W. R. Leaitch, and J. P. D. Abbatt (2011). "Aerosol composition and sources in the central Arctic Ocean during ASCOS". In: *Atmos. Chem. Phys.* 11.20, pp. 10619–10636. DOI: `10.5194/acp-11-10619-2011`.

Eckhardt, S., B. Quennehen, D. J. L. Olivié, T. K. Berntsen, R. Cherian, J. H. Christensen, W. Collins, S. Crepinsek, N. Daskalakis, M. Flanner, A. Herber, C. Heyes, Ø. Hodnebrog, L. Huang, M. Kanakidou, Z. Klimont, J. Langner, K. S. Law, M. T. Lund, R. Mahmood, A. Massling, S. Myriokefalitakis, I. E. Nielsen, J. K. Nøjgaard, J. Quaas, P. K. Quinn, J.-C. Raut, S. T. Rumbold, M. Schulz, S. Sharma, R. B. Skeie, H. Skov, T. Uttal, K. von Salzen, and A. Stohl (2015). "Current model capabilities for simulating black carbon and sulfate concentrations in the Arctic atmosphere: a multi-model evaluation using a comprehensive measurement data set". In: *Atmos. Chem. Phys.* 15.16, pp. 9413–9433. DOI: `10.5194/acp-15-9413-2015`.

Hawkins, L. N. and L. M. Russell (2010). "Polysaccharides, Proteins, and Phytoplankton Fragments: Four Chemically Distinct Types of Marine Primary Organic Aerosol Classified by Single Particle Spectromicroscopy". In: *Advances in Meteorology* 2010.ID 612132. DOI: `10.1155/2010/612132`.

Intrieri, J. M., C. Fairall, M. Shupe, P. Persson, A. E.L., P. Guest, and R. Moritz (2002). "An annual cycle of Arctic surface cloud forcing at SHEBA". In: *J. Geophys. Res.* 107.C10. ISSN: 0148-0227. DOI: `10.1029/2000jc000439`.

Loeb, N. G., D. R. Doelling, H. Wang, W. Su, C. Nguyen, J. G. Corbett, L. Liang, C. Mitrescu, F. G. Rose, and S. Kato (2018). "Clouds and the Earth's Radiant Energy System (CERES) Energy Balanced and Filled (EBAF) Top-of-Atmosphere (TOA) Edition-4.0 Data Product". In: *Journal of Climate* 31.2, pp. 895–918. DOI: `10.1175/JCLI-D-17-0208.1`.

Lohmann, U. and D. Neubauer (2018). "The importance of mixed-phase clouds for climate sensitivity in the global aerosol-climate model ECHAM6-HAM2". In: *Atmos. Chem. Phys. Disc.* 2018, pp. 1–32. DOI: `10.5194/acp-2018-97`.

Long, M. S., W. C. Keene, D. J. Kieber, D. J. Erickson, and H. Maring (2011). "A sea-state based source function for size- and composition-resolved marine aerosol production". In: *Atmos. Chem. Phys.* 11.3, pp. 1203–1216. DOI: `10.5194/acp-11-1203-2011`.

Matrosov, S. Y., A. V. Korolev, and A. J. Heymsfield (2002). "Profiling Cloud Ice Mass and Particle Characteristic Size from Doppler Radar Measurements". In: *Journal of Atmospheric and Oceanic Technology* 19.7, pp. 1003–1018. DOI: `10.1175/1520-0426(2002)019<1003:PCIMAP>2.0.CO;2`.

Sand, M., B. H. Samset, Y. Balkanski, S. Bauer, N. Bellouin, T. K. Berntsen, H. Bian, M. Chin, T. Diehl, R. Easter, S. J. Ghan, T. Iversen, A. Kirkevåg, J.-F. Lamarque, G. Lin, X. Liu,

G. Luo, G. Myhre, T. V. Noije, J. E. Penner, M. Schulz, Ø. Seland, R. B. Skeie, P. Stier, T. Takemura, K. Tsigaridis, F. Yu, K. Zhang, and H. Zhang (2017). "Aerosols at the poles: an AeroCom Phase II multi-model evaluation". In: *Atmos. Chem. Phys.* 17.19, pp. 12197–12218. DOI: 10.5194/acp-17-12197-2017.

Shupe, M. and T. Uttal (2007). *ETL Radar-based Cloud Microphysics Retrievals. Version 1.0. UCAR/NCAR - Earth Observing Laboratory.* https://doi.org/10.5065/D63R0R7P. DOI: doi.org/10.5065/D63R0R7P.

Shupe, M. D., P. Kollias, S. Y. Matrosov, and T. L. Schneider (2004). "Deriving Mixed-Phase Cloud Properties from Doppler Radar Spectra". In: *Journal of Atmospheric and Oceanic Technology* 21.4, pp. 660–670. DOI: 10.1175/1520-0426(2004)021<0660:DMCPFD>2.0.CO; 2.

Shupe, M. D., T. Uttal, and S. Y. Matrosov (2005). "Arctic Cloud Microphysics Retrievals from Surface-Based Remote Sensors at SHEBA". In: *Journal of Applied Meteorology* 44.10, pp. 1544–1562. DOI: 10.1175/JAM2297.1.

Sinha, P. R., Y. Kondo, M. Koike, J. A. Ogren, A. Jefferson, T. E. Barrett, R. J. Sheesley, S. Ohata, N. Moteki, H. Coe, D. Liu, M. Irwin, P. Tunved, P. K. Quinn, and Y. Zhao (2017). "Evaluation of ground-based black carbon measurements by filter-based photometers at two Arctic sites". In: *Geophy. Res. Lett.-Atmos.* 122.6, pp. 3544–3572. ISSN: 2169-8996. DOI: 10.1002/2016JD025843.

Stohl, A., Z. Klimont, S. Eckhardt, K. Kupiainen, V. P. Shevchenko, V. M. Kopeikin, and A. N. Novigatsky (2013). "Black carbon in the Arctic: the underestimated role of gas flaring and residential combustion emissions". In: *Atmos. Chem. Phys.* 13.17, pp. 8833–8855. DOI: 10.5194/acp-13-8833-2013.

Wang, X. and J. R. Key (2005). "Arctic Surface, Cloud, and Radiation Properties Based on the AVHRR Polar Pathfinder Dataset. Part I: Spatial and Temporal Characteristics". In: *Journal of Climate* 18.14, pp. 2558–2574. DOI: 10.1175/JCLI3438.1.

Westwater, E. R., Y. Han, M. D. Shupe, and S. Y. Matrosov (2001). "Analysis of integrated cloud liquid and precipitable water vapor retrievals from microwave radiometers during the Surface Heat Budget of the Arctic Ocean project". In: *Geophy. Res. Lett.-Atmos.* 106.D23, pp. 32019–32030. ISSN: 2156-2202. DOI: 10.1029/2000JD000055.

---

## Author Response (AR2)

*The authors decided to only rather superficially address my comments. They invested some effort in adding comparison to data as Supplementary material, but the discussion and interpretation of their results remains unaffected. Similarly, in terms of comparison to other models, only the global mean sea salt emissions are put into context of one previously-published model study. The fact that their model is an outlier is not taken into further consideration. Both results are used to declare their results as "uncertain". To respond to my proposition that the acting processes should be investigated in detail, the authors added two tables with mean numbers, without discussing these further.*

*Below I provide some more comments on the responses to my main concerns. In conclusion, I believe there is little use for another review. Even if I am quite disappointed by the replies, and indeed by the study itself, I believe only little improvement can be expected if one insisted on proper replies to the suggestions.*

*Observations: The authors provide now some comparison of their model to observational data. With respect to the aerosol, the results seem awkward. A low bias of 200 – 300% is quoted with respect to aerosol optical depth. In turn, in comparison to surface data, a factor of 2 overestimation is reported. This might imply problems with vertical and/or horizontal transport in the model for the Arctic, which puts the aerosol results simulated by the model into question.*

We thank the reviewer for his/her comments. We agree that problems with the transport/the vertical distribution of aerosols likely contribute to the underestimation in AOT. However, there are also other reasons why the AOT in the Arctic could be underestimated (e.g. missing aerosol sources). The following changes in the manuscript were made:

- We slightly changed the following sentence and shifted it from the Supplementary Material to the main paper: "The underestimation of AOT in our model is probably a combination of several causes, including e.g. missing local aerosol sources in the model (e.g. marine organics or gas flaring emissions; Hawkins and Russell 2010, Chang et al. 2011, Stohl et al. 2013), an underestimation of aerosol transport from midlatitudes to the Arctic (Bourgeois and Bey, 2011), uncertainties in the optical properties and emissions of aerosols (e.g. for BC, see Bond and Bergström 2006, Bond et al. 2013), and the neglection of spume drops in the sea salt parameterisation by Long et al. (2011)."

- We added the following sentence to the paper: "In general, it is very likely that our model underestimates the total aerosol concentrations in the Arctic."

- In the Section about the aerosol radiative forcings (difference between 2004 and 2050), we added the following text: "The simulated AOT has a low bias in the Arctic, which can affect these estimates of the aerosol radiative effect. Depending on whether the aerosol absorption or the scattering is underestimated, the aerosol radiative effect is either under- or overestimated. It is also possible that both effects cancel each other."

- In the conclusions, we added the following sentence: "Considering that our model probabily underestimates the background aerosol concentrations in the Arctic, the simulated impact of the (tenfold) ship emissions could be overestimated."

*A side note: "recent long-term surface observations" is a funny formulation. The ice water path bias is correctly discussed, and with regard to liquid clouds as well as the radiation balance, I also concur with the author's conclusion that the model performs almost satisfactorily.*

We changed the wording to: "with the recently published long-term surface observations"

*It is peculiar opinion that "model biases do not imply that the simulated changes are necessarily wrong". It certainly does not help build confidence that they are right. Any demonstration that the simulated changes are trustworthy of course would be welcome.*

We agree that it does not help to build confidence, points to problems in the model, and impedes a quantitative evaluation. We rather wanted to point out that a variable with a bias might still react correctly under changing conditions. On the other hand, a specific variable in the model that compares well with observations might react in a wrong way to changing conditions.

*I do not see how the quoted paragraph ("Heterogeneous freezing...") relates to the model evaluation discussion.*

In the quoted paragraph about heterogeneous freezing, we wanted to discuss whether the INP changes that our model predicts due to enhanced shipping (i.e., basically none) are supported by observations. We should have pointed that out more clearly in the reply to you.

*In my opinion, the model evaluation substantially improved, but a meaningful discussion of the results shown in the Supplementary material is missing. The authors merely superficially summarize the result of substantial model deficiencies, without discussing the implications for the interpretation of their simulation results.*

Next to the changes mentioned above concerning the impact of the low AOT/low aerosol concentrations in the Arctic, we added the following sentence to the section "Clouds" (difference between 2004 and 2050): "The absolute changes might be underestimated since our model in general underestimates the ice water content of clouds." The implications of the low sea salt mass emissions are mentioned below.

*Model context: The context of the model used by the authors with other model simulations also now improved a bit, but is still not very advanced. It is strange that the authors are not aware of the CMIP5 model capabilities. E.g. Ekman (JGR 2014) list half of the models as ones with interactive aerosol.*

This is correct, about half of the CMIP5 models have interactive aerosols, while a quarter of them provide sea salt emissions. The wording "nearly no CMIP5 model" in the reply was wrong.

*The fact that the model the authors use is even in the global mean lower than any of the other models does not add confidence to the simulations. But why do the authors not look at Arctic data alone? After all, this is the region of interest.*

As the reviewer suggests, we now restrict our comparison to the Arctic and the period of interest (late summer/early autumn). We compare our model with all CMIP5 models that provide sea salt emissions. The text was changed to:

"When we compare our simulated mass emissions in the Arctic (60° to 90° N) from July to October with the 11 CMIP5 models that provide sea salt mass emission fluxes, our model shows the lowest sea salt emissions: we arrive at a value of $5.9 \cdot 10^{-3} \mathrm{\mu g\,m^{-2}\,s^{-1}}$ under present-day conditions, while the CMIP5 models emit $\approx 4 \cdot 10^{-2} \mathrm{\mu g\,m^{-2}\,s^{-1}}$ (GISS-E2-H, GISS-E2-R, MIROC-ESM, MIROC-ESM-CHEM), 7 to $9 \cdot 10^{-2} \mathrm{\mu g\,m^{-2}\,s^{-1}}$ (MIROC5, NorESM1-M, NorESM-ME), and 1 to $2 \cdot 10^{-1} \mathrm{\mu g\,m^{-2}\,s^{-1}}$ (GFDL-CM3, MIROC4h, MRI-CGCM3, MRI-ESM1)."

The parameterisation by Long et al. (2011) is the new standard sea salt parameterisation of ECHAM-HAM since it compares satisfactorily with sea salt concentration measurements at stations with values below $100\,\mathrm{\mu g\,m^{-3}}$; higher concentrations point to local influences, which cannot be captured by ECHAM-HAM (Tegen et al., in preparation*). The lower mass emissions compared to other parameterisations (e.g. Guelle, 2001 and Gong et al., 2003) are due to less supermicron aerosol particles, which contribute little to the sea salt particle number concentrations and have a short atmospheric lifetime. Overall, we think that the smaller sea salt mass

emissions should not induce a low bias in CCN since the number concentrations in the accumulation mode are not lower than for the other parameterisations. Therefore, we added to following text: "Our simulated absolute increases in sea salt mass emissions are therefore likely underestimated because our parameterisation does not account for the contributions from spume drops (Long et al., 2011) and thus results in small emission fluxes of large (i.e. supermicron) aerosol particles. However, these large aerosol particles have a comparatively low impact on climate due to their low number concentrations. Since the total sea salt number emissions of the parameterisation by Long et al. (2011) are not generally lower than in other parameterisations, we do not expect that our simulated impact on CCN and radiation is completely different compared with other sea salt parameterisations. To confirm this, we conducted an additional simulation similar to arctic_2004, but with the old standard sea salt parameterisation of ECHAM-HAM (i.e. the parameterisation by Guelle et al., 2001). This parameterisation results in considerably higher sea salt mass emissions than the parameterisation by Long et al. (2011) ($9.8 \cdot 10^{-2} \mu g \, m^{-2} \, s^{-1}$ averaged from 60° to 90° N and from July to October). Nevertheless, the resulting AOT and CDNC are quite comparable: using the parameterisation by Guelle et al. (2001), the AOT is somewhat higher in the Arctic than with the parameterisation by Long et al. (2011) (0.039 compared to 0.034; averaged from July to October), while the CDNC burden is slightly lower ($4.8 \cdot 10^{10} m^{-2}$ compared to $5.0 \cdot 10^{10} m^{-2}$; in-cloud values)."

*Minimum CDNC: It is necessary that the authors quantify the "occasionally hit" (not necessary for the subtropics, but for the Arctic. What is the percentage of cloudy-sky situations in which the CDNC actually should be lower?*

We thank the reviewer for pointing this out again. We agree that a quantification is very helpful to interpret the results. For the diagnostics, we consider (next to the cloud occurrence frequency) how large the cloud cover in a gridbox is. Since mixed-phase clouds occur frequently in the Arctic, cloud cover represents both the liquid and the ice phase and is therefore alone not a good indicator for cloud water. Furthermore, the cloud mass is generally more important for cloud radiative effects than cloud volume. For these two reasons, we diagnose how often the threshold is hit referring to cloud mass. On the global scale, the threshold is hit 2.9%. In the Arctic, this threshold is higher with 15%. In late summer and early autumn, the threshold is hit slightly less in the Arctic with 11%. For the simulation 2050_EF2004, the value is similar (10%). In the manuscript, we replaced the previous sentence with: "In the Arctic, this threshold is hit 11% (weighted with liquid water content) averaged from July to October under present-day conditions. Without a lower threshold for CDNC, the underestimation of the Arctic aerosol concentrations in our model (see also Section 3) would locally lead to unrealistically high precipitation formation rates and a too strong effect of increased aerosol emissions on clouds." We acknowledge that it would be better if the model did not need such a minimum threshold. In the conclusions, we thus added the following text: "Missing aerosol sources such as nitrate and SOA most likely contribute the simulated underestimation of AOT. In future work, nitrate as well as a state-of-the-art SOA scheme will therefore be incorporated in our model."

*Tegen, I., U. Lohmann, D. Neubauer, C. Siegenthaler-Le Drian, S. Ferrachat, I. Bey, T. Stanelle, P. Stier, N. Schutgens, D. Watson-Parris, H. Schmidt, S. Rast, M. Schultz, S. Schroeder, H. Kokkola, S. Barthel, and B. Heinold. The aerosol-climate model ECHAM6.3-HAM2.3: Aerosol evaluation. To be submitted to Geosci. Model Dev. Discuss. 2018 (in preparation)

To summarise, we made the following changes:

- We now discuss the implications of the deficiencies of our model in more detail.

- As suggested by the reviewer, we now restrict our comparison of simulated sea salt emissions to the Arctic.

- To estimate the effect of the low sea salt mass emissions in our standard parameterisation, we conducted a simulation with another sea salt parameterisation and discuss the most important outcome.

- We now provide quantitative information about how often the CDNC treshold is hit.

[revised manuscript text omitted]